# A midbrain GABAergic circuit constrains wakefulness in a mouse model of stress

Shuancheng Ren [1,2,10] ✉, Cai Zhang[1,10], Faguo Yue[1,3,10], Jinxiang Tang[3], Wei Zhang[1], Yue Zheng[4], Yuanyuan Fang[5], Na Wang[1,6], Zhenbo Song[1], Zehui Zhang[7], Xiaolong Zhang[1], Han Qin [8], Yaling Wang[1], Jianxia Xia[1], Chenggang Jiang[9], Chao He [1] ✉, Fenlan Luo [1] ✉ & Zhian Hu [1,8] ✉

Enhancement of wakefulness is a prerequisite for adaptive behaviors to cope with acute stress, but hyperarousal is associated with impaired behavioral performance. Although the neural circuitries promoting wakefulness in acute stress conditions have been extensively identified, less is known about the circuit mechanisms constraining wakefulness to prevent hyperarousal. Here, we found that chemogenetic or optogenetic activation of GAD2-positive GABAergic neurons in the midbrain dorsal raphe nucleus (DRN$^{GAD2}$) decreased wakefulness, while inhibition or ablation of these neurons produced an increase in wakefulness along with hyperactivity. Surprisingly, DRN$^{GAD2}$ neurons were paradoxically wakefulness-active and were further activated by acute stress. Bidirectional manipulations revealed that DRN$^{GAD2}$ neurons constrained the increase of wakefulness and arousal level in a mouse model of stress. Circuit-specific investigations demonstrated that DRN$^{GAD2}$ neurons constrained wakefulness via inhibition of the wakefulness-promoting paraventricular thalamus. Therefore, the present study identified a wakefulness-constraining role DRN$^{GAD2}$ neurons in acute stress conditions.

Animals shift their wakefulness/sleep states according to the ever-changing internal state and external environment. When facing with allostatic challenges, such as in acute stress conditions or encountering emotionally salient stimuli, animals have the built-in ability to enhance wakefulness for favoring the behavioral demands[1,2]. Rapid transitions from sleep to wakefulness and elevations of arousal level facilitate animals to quickly detect the threating cues and make proper adaptive behaviors[1,3–5]. On the other hand, enhanced wakefulness in the limit can lead to distractibility and hamper behaviors. It has been suggested that inappropriate high arousal in response to acute stress is associated with declined cognitive performance[6,7]. Moreover, the abnormal hyperarousal is recognized as a core symptom of multiple neurological disorders, including stress-induced insomnia[8–10], mania phase of bipolar disorder[11], posttraumatic stress disorder[12], and attention deficit and hyperactivity disorder[13]. Wakefulness is thus tightly controlled in acute stress situations to avoid hyperarousal. However, the neural circuitries involved in constraining wakefulness in acute stress conditions are largely unknown.

Neurons synthesizing and releasing γ-aminobutyric acid (GABA) play prominent roles in the regulation of wakefulness/sleep. GABAergic

[1]Department of Physiology, College of Basic Medical Sciences, Army Medical University, Chongqing 400038, China. [2]No. 953 Army Hospital, Shigatse, Tibet Autonomous Region 857000, China. [3]Sleep and Psychology Center, Bishan Hospital of Chongqing Medical University, Chongqing 402760, China. [4]Department of Anesthesiology, Southwest Hospital, Army Medical University, Chongqing 400038, China. [5]Department of Anesthesiology, Zhongnan Hospital, Wuhan University, Wuhan 430071, China. [6]College of Bioengineering, Chongqing University, Chongqing 400044, China. [7]Department of Physiology, College of Basic Medical Sciences, Jilin University, Changchun 130021, China. [8]Chongqing Institute for Brain and Intelligence, Guangyang Bay Laboratory, Chongqing 400064, China. [9]Psychology Department, Women and Children's Hospital of Chongqing Medical University, Chongqing Health Center for Women and Children, Chongqing 401147, China. [10]These authors contributed equally: Shuancheng Ren, Cai Zhang, Faguo Yue. ✉e-mail: shuan0808@163.com; hechaochongqing@163.com; fenlanluo@163.com; zhianhu@aliyun.com

neurons promoting non-rapid eye movement (NREM) sleep have been identified in multiple brain areas, including the preoptic area[14,15], the basal forebrain[16], and the substantia nigra pars reticulata[17]. These GABAergic neurons are NREM sleep-active and suppress wakefulness-promoting systems to initiate and maintain NREM sleep[18,19]. In addition, populations of GABAergic neurons in the lateral hypothalamus (LH)[20,21], the basal forebrain[16,22,23], and the bed nucleus of the stria terminalis[24] are more active during wakefulness. Activation of these wakefulness-active GABAergic neurons induces fast transitions from sleep to wakefulness. Intriguingly, recent studies reported populations of GABAergic neurons with mismatched physiological activity in wakefulness/sleep cycles and behavioral outcomes of functional manipulations. For example, GABAergic neurons in the ventral tegmental area (VTA) are wakefulness-active[25,26], but lesioning of these neurons produced a dramatic increase of wakefulness[27,28]. In addition, GABAergic neurons in the locus coeruleus (LC) display either correlated or anticorrelated activity with arousal and shape the arousal responses to acoustic stimuli[29]. These results highlighted a previously underappreciated complexity of GABAergic neurons in wakefulness control.

The midbrain dorsal raphe nucleus (DRN) has long been considered as a principal part of wakefulness/sleep-regulation system, containing neuronal populations releasing serotonin (5HT), dopamine (DA), GABA, and glutamate[30,31]. GABAergic neurons are the second major cell population in the DRN. DRN GABAergic neurons send wide projections to brain regions involved in wakefulness/sleep regulation[32,33], including the LH, the preoptic area, the VTA, and midline thalamus, and thus are engaged in the control a variety of state-dependent behaviors[33–36]. GABAergic neurons are heterogeneous and subtypes of GABAergic neurons with different functions in wakefulness/sleep regulation have been reported[18]. For example, vesicular GABA transporter (Vgat)-positive GABAergic neurons in the VTA are wakefulness-active but NREM sleep-promoting[25], whereas glutamic acid decarboxylase 1 (GAD1)-positive VTA GABAergic neurons are NREM sleep-active[37]. As for the DRN GABAergic neurons, Vgat-positive DRN (DRN$^{Vgat}$) neurons have been shown to be wakefulness-active and wakefulness-promoting[38,39]. Activation of DRN$^{Vgat}$ neurons induces sleep to wakefulness transitions[38,39]. Glutamic acid decarboxylase 2 (GAD2)-positive neurons in the DRN (DRN$^{GAD2}$) are activated by arousing stressful conditions, such as social defeat and stress hormone corticosterone[40,41], also suggesting a potential role of these neurons in wakefulness control. However, direct evidence linking DRN$^{GAD2}$ neurons with wakefulness regulation, particularly in acute stress conditions, is still lacking.

In the present study, using cell type-specific manipulations and in vivo neural activity recording, we found that activation of DRN$^{GAD2}$ neurons decreased wakefulness but these neurons were paradoxical wakefulness-active. Combining with behavior paradigms, we revealed that DRN$^{GAD2}$ neurons were activated by acute stress and constrained wakefulness via inhibition of thalamic wakefulness-promoting neurons to prevent hyperarousal. Overall, our results delineated a wakefulness-constraining role of DRN$^{GAD2}$ neurons and advanced the mechanisms of finely regulation of wakefulness in acute stress conditions.

## Results
### Selective activation of DRN$^{GAD2}$ neurons decreases wakefulness
First, we examined whether DRN$^{GAD2}$ neurons were directly involved in the regulation of wakefulness. We used chemogenetic method to activate DRN$^{GAD2}$ while monitoring wakefulness/sleep states by electroencephalogram (EEG)/electromyogram (EMG) recording. Cre-dependent adeno-associated virus (AAV) encoding excitatory DREADDs (AAV-EF1α-DIO-hM3Dq-mCherry) was injected into the DRN of GAD2-Cre mice (Fig. 1a), which exclusively expressed in the DRN with few ectopic expressions in adjacent brain regions (Fig. 1b, Supplementary Fig. 1a). Fluorescence in situ hybridization (FISH) validated the specificity and efficiency of hM3Dq-mCherry expression in the DRN of GAD2-Cre mice (Fig. 1c, d). At the beginning of dark phase

(zeitgeber time (ZT) 12), a specific ligand for hM3Dq, clozapine-N-oxide (CNO, 1 mg/kg), or saline as a control, was intraperitoneally injected in a random order (Fig. 1e). Compared with saline injection, CNO-induced activation of DRN$^{GAD2}$ neurons significantly decreased wakefulness ($-21.52 \pm 2.97\%$) with complementary increased NREM sleep ($22.28 \pm 3.13\%$) during the first 3 h (ZT12–15) post injection (Fig. 1f, Supplementary Fig. 1b–d). Moreover, we found that chemogenetic activation of DRN$^{GAD2}$ neurons during the light phase (ZT0–3) also decreased wakefulness and increased NREM sleep (Supplementary Fig. 1e–i). However, CNO itself had no obvious effects on wakefulness/sleep of GAD2-Cre mice with AAV-EF1α-DIO-mCherry expression in the DRN (Supplementary Fig. 1j, k).

To investigate the role of DRN$^{GAD2}$ neurons in regulating the transitions between different wakefulness/sleep states, we injected AAV encoding channelrhodopsin-2 fused with mCherry (AAV-EF1α-DIO-ChR2-mCherry) into the DRN of GAD2-Cre mice for optogenetic activation (Fig. 1g). This injection resulted in specific expression of ChR2-mCherry in GAD2-positive neurons in the DRN (Fig. 1h–j, Supplementary Fig. 2a). Optogenetic stimulation (473 nm, 20 Hz, 120 s per trial) was randomly applied every 15–20 min at ZT 4–9 during the light phase (Fig. 1k). We quantified the effects of optogenetic activation of DRN$^{GAD2}$ neurons on wakefulness/sleep by calculating the percentage of each brain state around optogenetic stimulation (Fig. 1l). We found that optogenetic activation of DRN$^{GAD2}$ neurons significantly increased the probability (120 s before or during optogenetic stimulation) of NREM sleep while decreased wakefulness, without significantly affecting REM sleep (Fig. 1l, m, Supplementary Movie. 1). In DRN$^{GAD2}$-mCherry mice, laser stimulation during the light phase had no obvious effects on wakefulness/sleep states (Supplementary Fig. 2b–d, Supplementary Movie 1). Furthermore, optogenetic activation of DRN$^{GAD2}$ neurons during the dark phase induced significant increase in NREM sleep and decrease in wakefulness, too (Supplementary Fig. 2e–j). We also examined the role of DRN$^{GAD2}$ neurons in behavior control when exploring an arousing novel environment (Supplementary Fig. 3a). DRN$^{GAD2}$-ChR2 mice were introduced into an open field chamber and the locomotor activities before and during 20 Hz laser stimulation were analyzed. Compared with the baseline condition, optogenetic activation of DRN$^{GAD2}$ neurons decreased the traveled distance (Supplementary Fig. 3b, c). These data together demonstrated that activation of DRN$^{GAD2}$ neurons decreased wakefulness by inducing wakefulness to NREM sleep transitions.

### Inhibition or ablation of DRN$^{GAD2}$ neurons increases wakefulness
We next examined the necessity of DRN$^{GAD2}$ neurons for wakefulness/sleep. Cre-dependent AAV encoding inhibitory DREADD (AAV-EF1α-DIO-hM4Di-mCherry) was injected into the DRN of GAD2-Cre mice (Fig. 2a, b, Supplementary Fig. 4a). The cell-type specificity of hM4Di-mCherry expression was validated by FISH (Fig. 2c, d). Saline or CNO (2 mg/kg or 5 mg/kg) was intraperitoneally injected at the beginning of light phase (ZT 0) (Fig. 2e, Supplementary Fig. 4e). Compared with saline controls, we found that CNO-induced inhibition of DRN$^{GAD2}$ neurons resulted in a significant increase of wakefulness (2 mg/kg, $16.63 \pm 2.84\%$) with corresponding decrease of NREM sleep (2 mg/kg, $-15.05 \pm 2.52\%$) during the first 3 h of light phase (ZT0–ZT3), without obvious effects on REM sleep (Fig. 2f, Supplementary Fig. 4b–d, f–i). In addition, CNO injection at the beginning of dark phase (ZT12) increased wakefulness in comparison with saline injection, with corresponding decrease of NREM and REM sleep (Supplementary Fig. 4j–n). Moreover, we examined the effects of chemogenetic inhibition of DRN$^{GAD2}$ neurons on locomotor activity (Supplementary Fig. 3d). We found that chemogenetic inhibition of DRN$^{GAD2}$ neurons significantly increased the traveled distance of mice in the open field test (Supplementary Fig. 3e, f). These results indicated that DRN$^{GAD2}$ neurons were essential for wakefulness/sleep regulation.

To further confirm the essential role of DRN$^{GAD2}$ neurons for wakefulness/sleep, we also injected AAV encoding diphtheria toxin

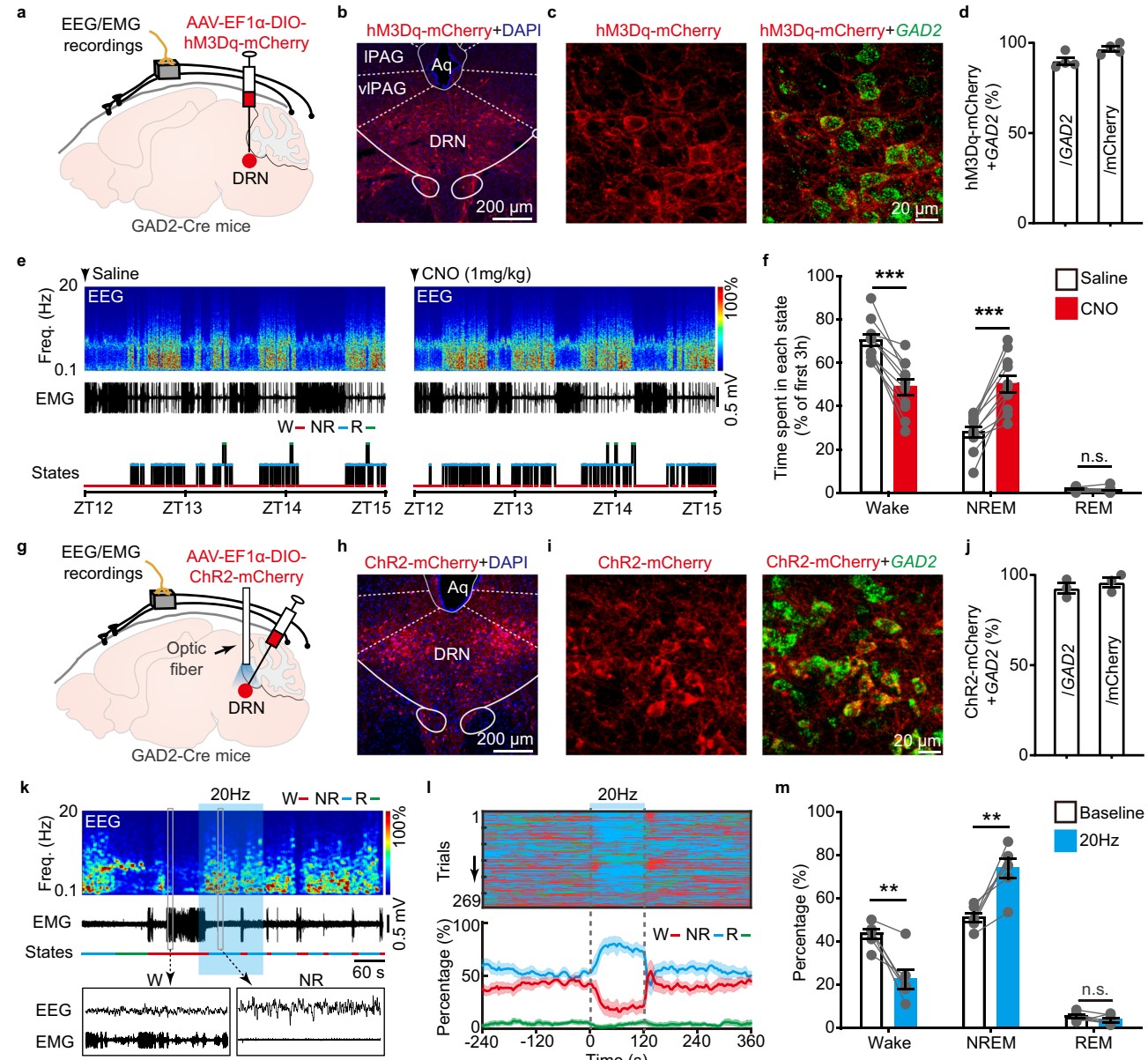

**Fig. 1 | Chemogenetic or optogenetic activation of DRN^GAD2 neurons decreases wakefulness. a** Schematic of viral injection and EEG/EMG recordings. AAV-EF1α-DIO-hM3Dq-mCherry was injected into the DRN of GAD2-Cre mice for chemogenetic activation. **b, c** Representative images showing the expression of hM3Dq-mCherry in the DRN (**b**) and the colocalization of hM3Dq-mCherry with *GAD2* mRNA (**c**). lPAG, lateral periaqueductal gray; vlPAG, ventrolateral periaqueductal gray; Aq, aqueduct. **d** Quantitative analysis of the efficiency and specificity of hM3Dq-mCherry to label DRN^GAD2 neurons. $n = 4$ mice. **e** EEG power spectrogram, EMG traces, and hypnograms from a DRN^GAD2-hM3Dq mouse during 3 h post saline (left) or CNO (1 mg/kg, right) injection. Freq., frequency; W, wake; NR: NREM; R: REM. **f** Time spent in each state during the first 3 h after saline or CNO injection. $n = 11$ mice, two tailed paired $t$ test, wake: $t_{10} = 7.248$, $P = 2.76 \times 10^{-5}$; NREM: $t_{10} = 7.128$, $P = 3.19 \times 10^{-5}$; REM: $t_{10} = 1.703$, $P = 0.119$. **g** Schematic of viral injection and EEG/EMG recordings. AAV-EF1α-DIO-ChR2-mCherry was injected into the DRN

of GAD2-Cre mice for optogenetic activation. **h, i** Example images showing the expression of ChR2-mCherry in the DRN (**h**) and the colocalization of ChR2-mCherry with *GAD2* mRNA (**i**). **j** Quantitative analysis of the efficiency and specificity of ChR2-mCherry to label DRN^GAD2 neurons. $n = 3$ mice. **k** EEG power spectrogram, EMG trace, and hypnogram showing optogenetic activation of DRN^GAD2 neurons from a DRN^GAD2-ChR2 mouse during the light phase. Laser stimulation is indicated by blue stripe. **l** Top, brain states of recorded trials from DRN^GAD2-ChR2 mice. Bottom, percentage of wake, NREM, or REM sleep around 20 Hz stimulation of DRN^GAD2-ChR2 mice. Shading represents ±SEM. **m** Quantification of time spent in each state 120 s before and during optogenetic stimulation of DRN^GAD2 neurons. $n = 6$ mice, two tailed paired $t$ test, wake: $t_5 = 4.775$, $P = 0.00499$; NREM: $t_5 = 5.243$, $P = 0.00335$; REM: $t_5 = 1.425$, $P = 0.214$. **P < 0.01, ***P < 0.001, n.s., not significant. Data (**d, f, j, m**) are presented as mean ± SEM.

subunit A (DTA, AAV-EF1α-DIO-DTA) to selectively ablate DRN^GAD2 neurons (Fig. 2g). DTA injection effectively induced a deletion of DRN^GAD2 neurons, which was identified by the significant decrease of *GAD2*-positive neurons in the DRN (Fig. 2h, Supplementary Fig. 5a–d). EEG/EMG recordings were performed at least 4 weeks after DTA injections. Compared with mCherry mice, we found that ablation of DRN^GAD2 neurons resulted in an increase of time spent in wakefulness,

particularly during the dark phase of mice (Fig. 2i, j). Ablation of DRN^GAD2 neurons reduced dark phase NREM sleep (Fig. 2k). No obvious changes of REM sleep were observed during the dark phase except a slight but significant increase during the light phase (Fig. 2l). Analysis of wakefulness/sleep architecture revealed that ablation of DRN^GAD2 neurons increased the probability and duration of long-bout wake (>1 min) during the dark phase (Supplementary Fig. 5e–g). Meanwhile,

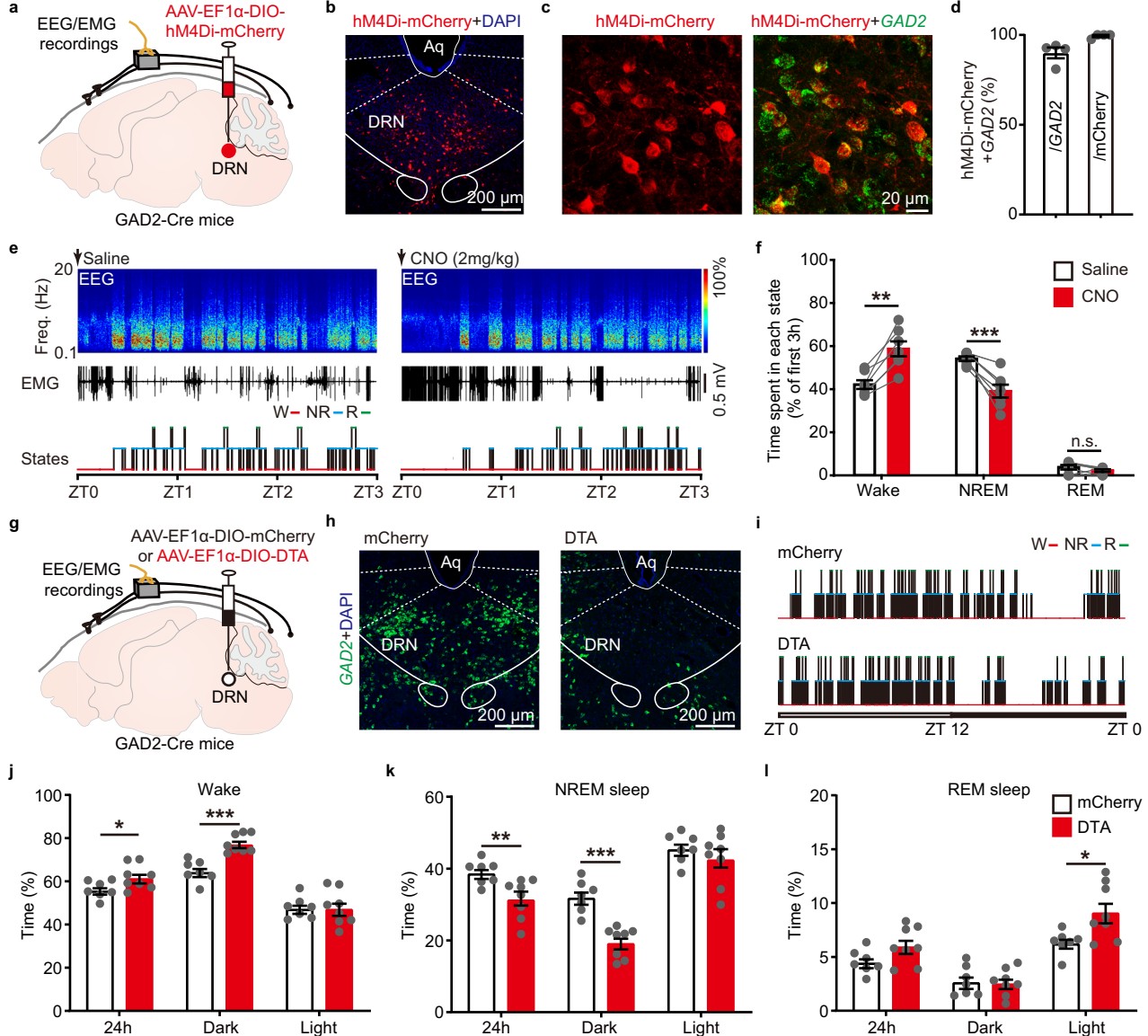

**Fig. 2 | Inhibition or ablation of DRN^GAD2 neurons increases wakefulness.**
**a** Schematic of viral injection and EEG/EMG recordings. AAV-EF1α-DIO-hM4Di-mCherry was injected into the DRN of GAD2-Cre mice for chemogenetic inhibition. **b**, **c** Representative images showing the expression of hM4Di-mCherry in the DRN (**b**) and the colocalization of hM4Di-mCherry with *GAD2* mRNA (**c**). Aq, aqueduct. **d** Quantitative analysis of the efficiency and specificity of hM4Di-mCherry to label DRN^GAD2 neurons. $n = 4$ mice. **e** EEG power spectrogram, EMG traces, and hypnograms from a DRN^GAD2-hM4Di mouse during 3 h post saline (left) or CNO injection (2 mg/kg, right). Freq., frequency; W, wake; NR: NREM; R: REM. **f** Time spent in each state during the first 3 h after saline or CNO injection. $n = 7$ mice, two tailed paired $t$ test for wake and NREM, wake: $t_7 = 5.85$, $P = 0.0011$; NREM: $t_7 = 5.971$, $P = 0.00099$. For REM, Wilcoxon signed rank test, $W = 11$, $P = 0.313$. **g** Schematic of viral injection

and EEG/EMG recordings. AAV-EF1α-DIO-DTA or AAV-EF1α-DIO-mCherry was injected into the DRN of GAD2-Cre mice. **h** Images showing the expression of *GAD2* mRNA in GAD2-Cre mice injected with mCherry or DTA. **i** Examples of hypnograms from a mCherry mouse or DTA mouse across 24 h light/dark cycle. **j**–**l** Quantitative analysis of percentage of time spent in wakefulness (**j**), NREM sleep (**k**) and REM sleep (**l**) during 24 h, dark, and light phase. $n = 7$ for mCherry mice and $n = 8$ for DTA mice, two tailed unpaired $t$ test. For wakefulness in (**j**), 24 h: $t_{13} = 2.221$, $P = 0.0447$; dark: $t_{13} = 5.685$, $P = 7.48 \times 10^{-5}$; light: $t_{13} = 0.25$, $P = 0.807$. For NREM sleep in (**k**), 24 h: $t_{13} = 3.031$, $P = 0.00965$; dark: $t_{13} = 5.662$, $P = 7.77 \times 10^{-5}$; light: $t_{13} = 0.914$, $P = 0.377$. For REM sleep in (**l**), 24 h: $t_{13} = 1.992$, $P = 0.0678$; dark: $t_{13} = 0.175$, $P = 0.864$; light: $t_{13} = 2.715$, $P = 0.0177$. *$P < 0.05$, **$P < 0.01$, ***$P < 0.001$, n.s., not significant. Data (**d**, **f**, **j**–**l**) are presented as mean ± SEM.

ablation of DRN^GAD2 neurons increased the power of theta (6–10 Hz) activity whereas decreased delta (0.5–4 Hz) activity of wakefulness, indicating a more consolidated wakefulness state (Supplementary Fig. 5h, i). Moreover, ablation of DRN^GAD2 neurons increased the traveled distance of mice in the open field (Supplementary Fig. 5j–l), indicating hyperarousal-like behaviors[27,28].

Together, the above findings suggested that DRN^GAD2 neurons were tightly involved in the regulation of wakefulness/sleep. Moreover, results from functional manipulation experiments implied a possible NREM sleep-promoting role of DRN^GAD2 neurons.

## Populational activities of DRN^GAD2 neurons are wakefulness-active

Based on the chemogenetic, optogenetic, and genetic ablation results, we might have expected DRN^GAD2 neurons to be NREM sleep-active. To test this, we first mapped the *Fos* (a marker of neuronal activity) expression pattern of DRN^GAD2 neurons during the diurnal wakefulness/sleep cycles of mice. Wild type (WT) mice were perfused at light phase ZT6, dark phase ZT18, and after sleep deprivation (SD) from ZT0 to ZT6. FISH results for *Fos* and *GAD2* revealed that mice perfused at different time points had biased *Fos* expression patterns

(Supplementary Fig. 6a). We found that mice perfused after 6 h SD had a markedly higher number of *GAD2*-positive neurons expressing *Fos* than that of mice perfused at phase ZT 6 (Supplementary Fig. 6b), indicating that DRN[GAD2] neurons were more active during prolonged wakefulness period.

To directly monitor the activity profiles of DRN[GAD2] neurons across spontaneous wakefulness/sleep cycles, we applied fiber photometry to measure the temporal dynamics of populational calcium (Ca[2+]) activities of DRN[GAD2] neurons in freely behaving mice. AAV encoding Ca[2+] indicator jCaMP7b (AAV-EF1α-DIO-jGCaMP7b) was injected into the DRN of GAD2-Cre mice and an optic fiber was implanted to collect the jGCaMP7b fluorescent signals (Fig. 3a, b, Supplementary Fig. 6c). FISH results showed that most of jGCaMP7b-labeled DRN neurons were *GAD2*-positive (Fig. 3c, d). Populational Ca[2+] activities of DRN[GAD2] neurons exhibited strong state-dependent variations (Fig. 3e). Compared with NREM sleep and REM sleep states, DRN[GAD2] neurons displayed significantly higher Ca[2+] activities in wakefulness state (Fig. 3f). In addition, Ca[2+] activities were higher during long-bout (>1 min) wakefulness than short-bout (<1 min) wakefulness state (Fig. 3g). We next examined the activities of DRN[GAD2] neurons during different wakefulness/sleep state transitions. Ca[2+] activities of DRN[GAD2] neurons gradually decreased when mice falling into NREM sleep (Fig. 3h). Conversely, DRN[GAD2] neurons activities significantly increased when mice woke up from either NREM sleep or REM sleep (Fig. 3i, k). There was no obvious fluorescence change of DRN[GAD2] neurons activities during NREM sleep to REM sleep transitions (Fig. 3j). These data indicated that the populational Ca[2+] activities of DRN[GAD2] neurons were wakefulness-active.

We proceeded to determine whether activities of DRN[GAD2] neurons were altered at different arousal level. We first examined the correlations between Ca[2+] activities of DRN[GAD2] neurons and specific EEG power bands. During wakefulness, Ca[2+] signals of DRN[GAD2] neurons showed a weak positive correlation with the EEG theta activity and negative correlation with EEG delta activity (Supplementary Fig. 6d–f, $R_{delta} = -0.07$, $R_{theta} = 0.12$). Ca[2+] activities of DRN[GAD2] neurons were negatively correlated with both delta and theta power during NREM sleep (Supplementary Fig. 6g–i, $R_{delta} = -0.18$, $R_{theta} = -0.26$). No significant correlations were observed between Ca[2+] activities of DRN[GAD2] neurons and EEG power of REM sleep (Supplementary Fig. 6j–l). Furthermore, we recorded pupil diameter as an index for characterizing the arousal level, which tracks the fast changes in brain state[4]. After depth accommodation to head-fixed setups with controlled ambient luminance (~60 lx), we recorded pupil size fluctuations in awake mice while measuring the Ca[2+] activities of DRN[GAD2] neurons (Fig. 3l). Simultaneous Ca[2+] and pupil recordings revealed a correlation between spontaneous Ca[2+] activities of DRN[GAD2] neurons and pupil diameter changes (Fig. 3m). We extracted all pupil dilation events (1509 trials from 5 mice) and analyzed Ca[2+] activities of DRN[GAD2] neurons around pupil dilations (Fig. 3n). We found that the Ca[2+] activities of DRN[GAD2] neurons significantly increased after pupil dilation and positively correlated with pupil size (Fig. 3o, p). These findings demonstrated that the activities of DRN[GAD2] neurons were wakefulness-active and displayed arousal state-dependent alterations.

## DRN[GAD2] neurons constrain wakefulness in acute stress conditions

Chemogenetic or optogenetic activation of DRN[GAD2] neurons increased NREM sleep, but it was unlikely that wakefulness-active DRN[GAD2] neurons drove wakefulness to NREM sleep transitions, thus against a NREM sleep-promoting role of these neurons. Rather, DRN[GAD2] neurons might suppress wakefulness-promoting neurons following sleep to wakefulness transitions or in arousing conditions, to tightly control wakefulness state. Indeed, ablation or inhibition of DRN[GAD2] neurons induced hyperarousal-like behaviors, including increased wakefulness (Fig. 2), hyperactivity (Supplementary Figs. 3d to f and 5j–l), and

hyperalgesia to mechanical stimuli[42]. Thus, we hypothesized that DRN[GAD2] neurons may constrain wakefulness in acute stress conditions to prevent a state of hyperarousal.

To test this hypothesis, we used an acute restraint stress model to induce arousing acute stress condition[43]. DRN[GAD2]-mCherry mice were exposed to acute restraint for 10 min and then perfused for c-Fos staining (Fig. 4a). Quantification of c-Fos revealed a significantly stronger activation of mCherry-labeled-DRN[GAD2] neurons in restraint group than that in control group (Fig. 4b, c, control: 7.86 ± 1.16%; restraint: 67.02 ± 5.22%). These results reinforced the previous notion that stress induces activation of DRN[GAD2] neurons[40] and further implied that DRN[GAD2] neurons may play a role in acute stress conditions. To examine whether DRN[GAD2] neurons were involved in constraining wakefulness in acute stress conditions, we subjected DRN[GAD2]-DTA mice or DRN[GAD2]-mCherry mice to acute restraint for 10 min. Wakefulness/sleep states were immediately recorded after mice releasing from the restraint tube (Fig. 4d). Acute restraint typically causes delayed sleep onset latency, increased wakefulness or even insomnia[9,10]. Compared with mCherry controls, DRN[GAD2]-DTA mice exhibited a more pronounced delay of NREM sleep onset latency (Fig. 4e). In addition, DRN[GAD2]-DTA mice spent more time in wakefulness with concomitant less time in NREM and REM sleep (Fig. 4f). These results demonstrated that mice without DRN[GAD2] neurons had more enhanced increase of wakefulness in acute stress conditions.

We further asked whether chemogenetic activation of stress-activated DRN[GAD2] neurons could constrain the acute stress-induced increase of wakefulness. We specifically labeled restraint stress activated-DRN[GAD2] neurons using c-Fos-based activity tagging method, also known as TetTagging, according to previous published method[44]. Cre recombinase-dependent tagging vectors (AAV-cFos-tTA and AAV-TRE-DIO-hM3Dq-mCherry) were injected into the DRN of GAD2-Cre mice. To drive a c-Fos-dependent hM3Dq-mCherry expression in DRN[GAD2] neurons, mice were experienced a session of 10 min restraint stress when doxycycline (Dox) was removed from the diet (Fig. 4g). Post experiments histological analysis showed that 74.21% TRE-hM3Dq-mCherry neurons were c-Fos positive (Fig. 4h, i). In undisturbed physiological conditions, CNO (1 mg/kg) injection at the beginning of dark phase significantly decreased wakefulness while increased NREM sleep when compared with saline injection (Supplementary Fig. 7). In acute stress conditions, we found that activation of c-Fos-tagged DRN[GAD2] neurons significantly shortened the NREM sleep latency following restraint stress (Fig. 4j). More importantly, activation of c-Fos-tagged DRN[GAD2] neurons decreased wakefulness while increased NREM sleep in acute stress conditions (Fig. 4k). These results suggested that DRN[GAD2] neurons were activated in response to acute stress and could constrain the increased wakefulness in acute stress conditions.

## DRN[GAD2] neurons control arousal responses to stressful stimuli

Besides inducing sleep to wakefulness alterations and marked increase of wakefulness, acute stress was also associated with urgent elevation of arousal level[4]. Previous reports have shown that DRN GABAergics neurons are activated in arousing stressful conditions[35,36,45], raising a possibility that the activities of DRN[GAD2] neurons may shape arousal responses to the stressful stimuli. To address this proposal, we first recorded the populational Ca[2+] activities of DRN[GAD2] neurons in awake head-fixed mice while delivering a series of mild tail shock as arousing stressful stimuli (Fig. 5a). We found that tail shock current-dependently increased pupil size (Fig. 5b, d, Supplementary Fig. 8a), indicating elevations of arousal level. Moreover, tail shock induced-enlargement of pupil size was consistently associated with increased Ca[2+] activities of DRN[GAD2] neurons (Fig. 5c, d, Supplementary Fig. 8b), suggesting potential functional connections between these two events in stress conditions.

To investigate whether the activities of DRN[GAD2] neurons shaped pupillary responses in stress conditions, we used optogenetics to

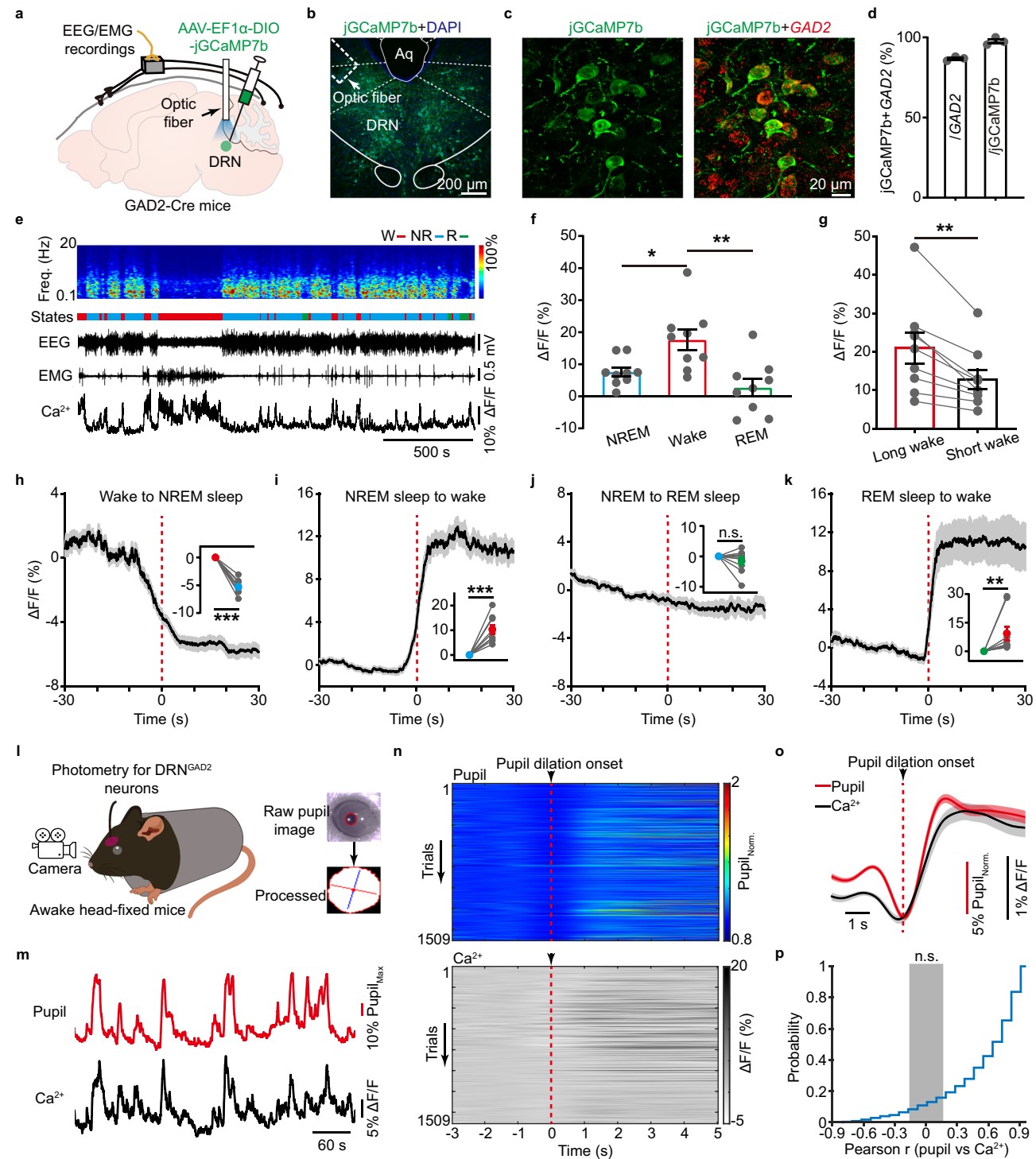

bidirectionally modulate DRN^GAD2 neurons during tail shock-induced changes of pupil size. AAV encoding inhibitory opsin (AAV-EF1α-DIO-GtACR-GFP) was injected into the DRN of GAD2-Cre mice, which specifically transfected of DRN^GAD2 neurons (Fig. 5e, Supplementary Fig. 9a–d). Functional expression of GtACR-GFP was further verified using whole-cell patch-clamp recording (Supplementary Fig. 9e–g). Laser stimulation was randomly delivered to silence the activity of DRN^GAD2 neurons when mice receiving tail shocks (0.1 mA, 1 s). Compared with light off conditions, optogenetic inhibition of DRN^GAD2 neurons led to a more prominent increase of pupil size (Fig. 5f, g, Supplementary Movie 2), indicating enhanced arousal responses. Laser stimulation itself had no obvious effects on tail shock-induced increase

of pupil size (Supplementary Fig. 10a, b). In contrast, we found that optogenetic activation of DRN^GAD2 neurons significantly attenuated the increase of pupil size induced by tail shock (0.2 mA, 1 s) (Fig. 5h–j). Furthermore, we compared the effects of manipulating DRN^GAD2 neurons on pupillary responses in undisturbed physiological conditions and in acute stress conditions. In undisturbed conditions, optogenetic activation of DRN^GAD2 neurons significantly decreased pupil size (Supplementary Fig. 10c, d). Optogenetic activation-induced pupil size change was negatively correlated with baseline pupil size, as activating DRN^GAD2 neurons in periods with large baseline pupil size produced more obvious pupil constriction (Supplementary Fig. 10e). Notably, activation of DRN^GAD2 neurons-induced pupil size decrease was larger in tail shock

**Fig. 3 | DRN$^{GAD2}$ neurons display wakefulness-dependent populational activities. a** Schematic of fiber photometry and simultaneous EEG/EMG recordings. AAV-EF1α-DIO-jGCaMP7b was injected into the DRN of GAD2-Cre mice. **b, c** Representative images showing the expression of jGCaMP7b in the DRN (**b**) and the colocalization of jGCaMP7b with *GAD2* mRNA (**c**). Aq, aqueduct. **d** Quantitative analysis of the efficiency and specificity of jGCaMP7b to label DRN$^{GAD2}$ neurons. $n = 3$ mice. **e** Representative EEG spectrogram, EEG/EMG traces, color-coded brain states and Ca$^{2+}$ fluorescence trace across different wakefulness/sleep states. Freq., frequency; W, wake; NR: NREM; R: REM. **f** Quantification of ΔF/F during wake, NREM sleep and REM sleep. $n = 9$ mice, one way ANOVA followed by Tukey post hoc test, $F_{2,24} = 8.38$, wake vs NREM: $P = 0.032$, wake vs REM: $P = 0.002$. **g** Comparison of ΔF/F during long- and short-bout wakefulness. $n = 9$ mice, two tailed paired $t$ test, $t_8 = 4.687$, $P = 0.00157$. **h–k** Ca$^{2+}$ activities of DRN$^{GAD2}$ neurons during transitions from wake to NREM sleep (**h**), NREM sleep to wake (**i**), NREM to REM sleep (**j**), and

REM sleep to wake (**k**). $n = 9$ mice, two tailed paired $t$ test (**h–j**). For (**h**), $t_8 = 12.075$, $P = 2.04 \times 10^{-6}$; for (**i**), $t_8 = 5.865$, $P = 3.76 \times 10^{-4}$; for (**j**), $t_8 = 1.273$, $P = 0.251$. For **k**, Wilcoxon signed rank test, W = 45, $P = 0.004$. Shading represents ±SEM. **l** Schematic experimental setup for simultaneous photometry and pupil recording in awake head-fixed mice. **m** Representative pupil (top) and Ca$^{2+}$ (bottom) traces during spontaneous pupil size recording. Pupil$_{Max}$, max pupil size. **n** Heatmaps showing all pupil dilation bouts (top) and corresponding Ca$^{2+}$ signals (bottom) extracted from 6 recorded mice. The vertical red lines indicate the onset of pupil dilation. Pupil$_{Norm}$, normalized pupil size. **o** Average traces of pupil size (red) and Ca$^{2+}$ signals (dark) aligned to onset of pupil dilation. Shading represents ±SEM. **p** Cumulative probability distribution of all the Pearson correlation coefficients of Ca$^{2+}$ signals of DRN$^{GAD2}$ neurons with pupil size. The gray area indicates the non-significant Pearson correlation coefficients. *$P < 0.05$, **$P < 0.01$, ***$P < 0.001$, n.s., not significant. Data (**d, f–k**) are presented as mean ± SEM.

conditions than that in baseline conditions (Supplementary Fig. 10f). In conjunction with the positive correlation between the activities of DRN$^{GAD2}$ neurons and arousal level, these functional manipulation results together indicated that DRN$^{GAD2}$ neurons sculpted the physiological arousal responses to acute stress and acted as a negative regulator of acute stress-induced elevation of arousal.

## DRN$^{GAD2}$ neurons constrain wakefulness by inhibiting the wakefulness-promoting paraventricular thalamus (PVT)

To search for the downstream wakefulness-promoting targets through which DRN$^{GAD2}$ neurons constrain wakefulness, we mapped the projection patterns of DRN$^{GAD2}$ neurons by delivering AAV-EF1α-DIO-EGFP into the DRN of GAD2-Cre mice (Fig. 6a, Supplementary Fig. 11a). As reported previously[32], DRN$^{GAD2}$ neurons send widely distributed projections to multiple brain regions associated with wakefulness/sleep regulation, including the nucleus accumbens, the basal forebrain, the preoptic area, the midline thalamus, the LH, the VTA, the parabrachial nucleus, and the LC (Supplementary Fig. 11). As we have shown that the PVT in the midline thalamus plays an important role in wakefulness control in previous studies[10,46], we thus examined whether DRN$^{GAD2}$ neurons could inhibit the PVT to tightly control wakefulness. We found that the density of axonal terminals of DRN$^{GAD2}$ neurons was gradually increased along the anterior to posterior axis of the PVT (Fig. 6b, c), consistent with the notion that the anterior and posterior parts of PVT have opposite role in arousal control[47]. With the help of recombinant rabies virus (RV), we performed monosynaptic retrograde tracing to further confirm the anatomical connections between DRN$^{GAD2}$ neurons and the PVT (Fig. 6d, e). We found a robust population of RV-labeled neurons in the DRN. To characterize the molecular identities of these neurons, we performed FISH or immunohistochemistry experiments. We found that most of RV-labeled DRN neurons were *GAD2*-positive (Fig. 6f, g) and only a minor proportion of RV-labeled DRN neurons were tryptophan hydroxylase 2 (Tph2, a marker of 5HT neurons)- or tyrosine hydroxylase (TH, a marker of DA neurons)-positive (Fig. 6g and Supplementary Fig. 12a–d), suggesting a preferential innervation of the PVT by DRN$^{GAD2}$ neurons. ChR2-associated circuit mapping was used to investigate the functional connections between DRN$^{GAD2}$ neurons and the PVT (Fig. 6h, i). We found that optogenetic activation of projection terminals of DRN$^{GAD2}$ neurons in the PVT evoked postsynaptic currents (Fig. 6j). Latency of light-evoked postsynaptic current was $5.15 \pm 0.56$ ms and average amplitude was $155.6 \pm 58.83$ pA (Fig. 6k). Following the isolation of light-evoked postsynaptic currents using blockers for sodium channels (tetrodotoxin, TTX) and potassium channels (4-aminopyridine, 4AP), we found that postsynaptic currents were blocked by a selective GABA$_A$ receptor antagonist bicuculine (Fig. 6j, l). In addition, bicuculine also abolished the postsynaptic currents induced by prolonged 10 Hz or 20 Hz stimulation in the presence of TTX and 4AP (Fig. 6m). These data demonstrated that DRN$^{GAD2}$ neurons formed monosynaptic inhibitory connections with the PVT.

We next investigated the functional role of DRN$^{GAD2}$-PVT circuit in wakefulness regulation. We used an intersectional strategy to chemogenetically activate PVT-projecting DRN$^{GAD2}$ neurons. An axon terminal-transducing AAV encoding Cre-dependent Flp recombinase (Retro-AAV-FLEx$^{loxP}$-FLP) was injected into the PVT and AAV expressing Flp-dependent hM3Dq (AAV-FLEx$^{FRT}$-hM3Dq-GFP) was injected into the DRN of GAD2-Cre mice, respectively (Fig. 7a, b). Chemogenetic activation of PVT-projecting DRN$^{GAD2}$ neurons at the beginning of dark phase induced a significant decrease of wakefulness (Fig. 7c, d, Supplementary Fig. 12e–g), indicating that inhibition of wake-promoting PVT neurons could partially mediate the effects of DRN$^{GAD2}$ neurons on wakefulness. We next investigated the effects of chemogenetic activation of PVT-projecting DRN$^{GAD2}$ neurons on wakefulness in acute stress conditions (Fig. 7e). Compared with saline injection, CNO injection significantly shortened the latency to NREM sleep, reduced wakefulness time while increased NREM sleep time (Fig. 7f, g). To examine whether DRN$^{GAD2}$ neurons control arousal responses to tail shock via inhibition of the PVT, we used fiber photometry to quantify the changes of PVT neurons to tail shock while activation of PVT-projecting DRN$^{GAD2}$ neurons (Supplementary Fig. 12h, i). We observed that tail shock reliably evoked activation of PVT neurons, whereas chemogenetic activation of PVT-projecting DRN$^{GAD2}$ neurons significantly decreased the excitatory effects of tail shock on PVT neurons (Supplementary Fig. 12j). Finally, we optogenetically activated the DRN$^{GAD2}$-PVT circuit when delivering tail shock (Fig. 7h). We found that activation of the DRN$^{GAD2}$-PVT circuit significantly decreased the tail shock-induced enlargement of pupil size (Fig. 7i–k, Supplementary Fig. 12k–m). Taken together, these results demonstrated that DRN$^{GAD2}$ neurons constrained wakefulness via inhibition of the wakefulness-promoting PVT neurons.

## Discussion

Combining chemogenetic and optogenetic manipulation, fiber photometry recording, anatomical and functional circuit mapping, and behavioral paradigms, we identified a population of GAD2-positivie neurons in the DRN for wakefulness control in physiological and acute stress conditions. Functional manipulation of DRN$^{GAD2}$ neurons suggested a possible NREM sleep-promoting role of these neurons. However, fiber photometry recording revealed wakefulness state-dependent activities of DRN$^{GAD2}$ neurons. We further demonstrated that DRN$^{GAD2}$ neurons were activated in conditions with stressful stimuli and acted as a brake to constrain wakefulness time and arousal level. Finally, we uncovered the circuit mechanism through which DRN$^{GAD2}$ neurons constrain wakefulness and demonstrated that the DRN$^{GAD2}$-PVT circuit controlled wakefulness and shaped arousal responses in acute stress conditions.

As a major part of the ascending reticular activating system, the DRN has long been implicated in the regulation of wakefulness/sleep. Early researches using large lesion or acute ablation methods have yielded controversial results about the role of DRN in wakefulness/

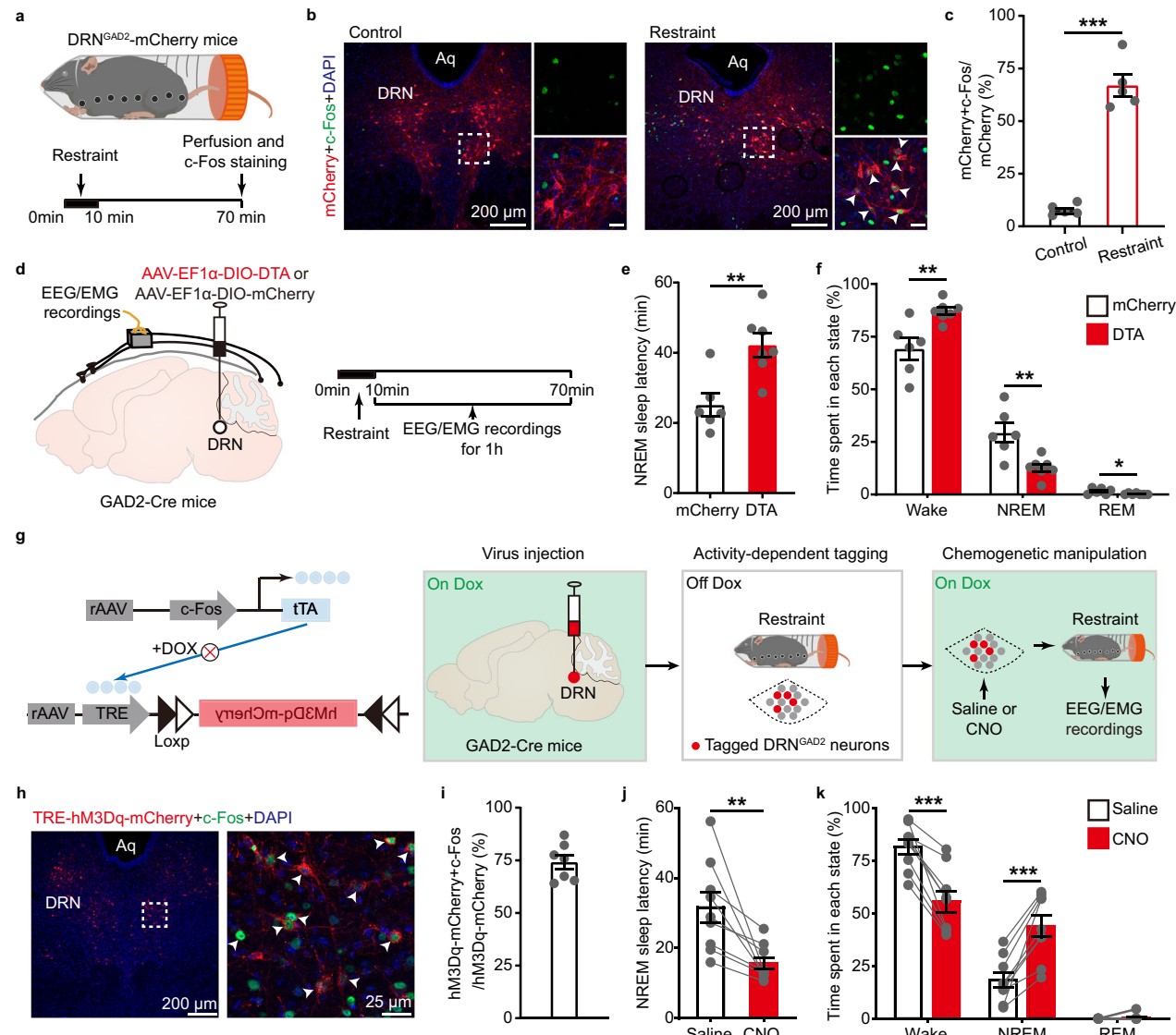

**Fig. 4 | DRN^GAD2 neurons constrain wakefulness in acute stress condition.**
**a** Schematic of a 10 min restraint stress session following c-Fos staining in DRN^GAD2-mCherry mice. **b** Representative images showing c-Fos expression in DRN^GAD2-mCherry mice from control (left) or restraint group (right). Zoomed images are enlarged views of the dashed areas. Arrowheads indicate mCherry and c-Fos co-labeled neurons. Scale bars in the right panels are 25 μm. Aq, aqueduct. **c** Number of mCherry and c-Fos co-labeled neurons in control and restraint group. $n = 5$ mice for each group, two tailed unpaired $t$ test, $t_8 = 11.056$, $P = 3.99 \times 10^{-6}$. **d** Schematic of a 10 min restraint stress session and the following EEG/EMG recordings in DRN^GAD2-mCherry or DRN^GAD2-DTA mice. **e** NREM sleep onset latency following a 10 min restraint session. $n = 6$ for mCherry mice and $n = 7$ for DTA mice, two tailed unpaired $t$ test, $t_{11} = 3.638$, $P = 0.0039$. **f** Time spent in wake, NREM sleep and REM sleep. $n = 6$ for mCherry mice and $n = 7$ for DTA mice, two tailed unpaired $t$ test,

wake: $t_{11} = 3.554$, $P = 0.00452$, NREM: $t_{11} = 3.561$, $P = 0.00446$; REM: $t_{11} = 2.374$, $P = 0.0369$. **g** Schematic of experimental protocol for c-Fos-based activity tagging method to label restraint stress-activated DRN^GAD2 neurons with TRE-hM3Dq-mCherry. **h, i** Representative images (**h**) and quantification (**i**) of the specificity of TRE-hM3Dq-mCherry to label restraint stress-activated DRN^GAD2 neurons. Arrowheads indicate TRE-hM3Dq-mCherry and c-Fos co-labeled neurons. $n = 7$ mice in (**i**). **j** NREM sleep onset latency following a 10 min restraint session. $n = 9$ mice, Wilcoxon signed rank test, W = 45, $P = 0.004$. **k** Time spent in wake, NREM sleep and REM sleep post saline or CNO injections in restraint stress conditions. $n = 9$ mice, two tailed paired $t$ test, wake: $t_8 = 6.708$, $P = 1.51 \times 10^{-4}$; NREM: $t_8 = 6.618$, $P = 1.66 \times 10^{-4}$. *$P < 0.05$, **$P < 0.01$, ***$P < 0.001$. Data (**c, e, f, i–k**) are presented as mean ± SEM.

sleep[2,48,49]. Recent cell-type based neuronal activity recording and functional manipulation studies revealed a functional diversity of DRN neurons in wakefulness/sleep. DRN 5-HT neurons have been proved to promote sleep in zebrafish and mice. Pharmacological inhibition or ablation of DRN 5-HT neurons reduces sleep, while tonic optogenetic activation increases sleep[50]. DRN DA neurons are wakefulness-active and optogenetic activation of these neurons potently promotes wakefulness. In addition, DRN DA neurons are further activated by salient stimuli and represent critical modulators of behavioral arousal[51]. Here we found that chemogenetic or optogenetic activation of DRN^GAD2 neurons increased NREM sleep, whereas chemogenetic

inhibition or genetic ablation increased wakefulness, implying a NREM sleep-promoting role of these neurons. Surprisingly, the spontaneous activity of DRN^GAD2 neurons positively correlated with wakefulness, suggesting that these neurons may function more than regulating physiological wakefulness. We further showed that DRN^GAD2 neurons were activated by acute stress and constrained the heightened wakefulness in acute stress conditions. These results uncovered a role of DRN^GAD2 neurons in wakefulness/sleep, highlighting the importance of DRN in brain states control.

Most neuronal ensembles engaged in wakefulness/sleep regulation show consistent spontaneous activity and physiological function. For

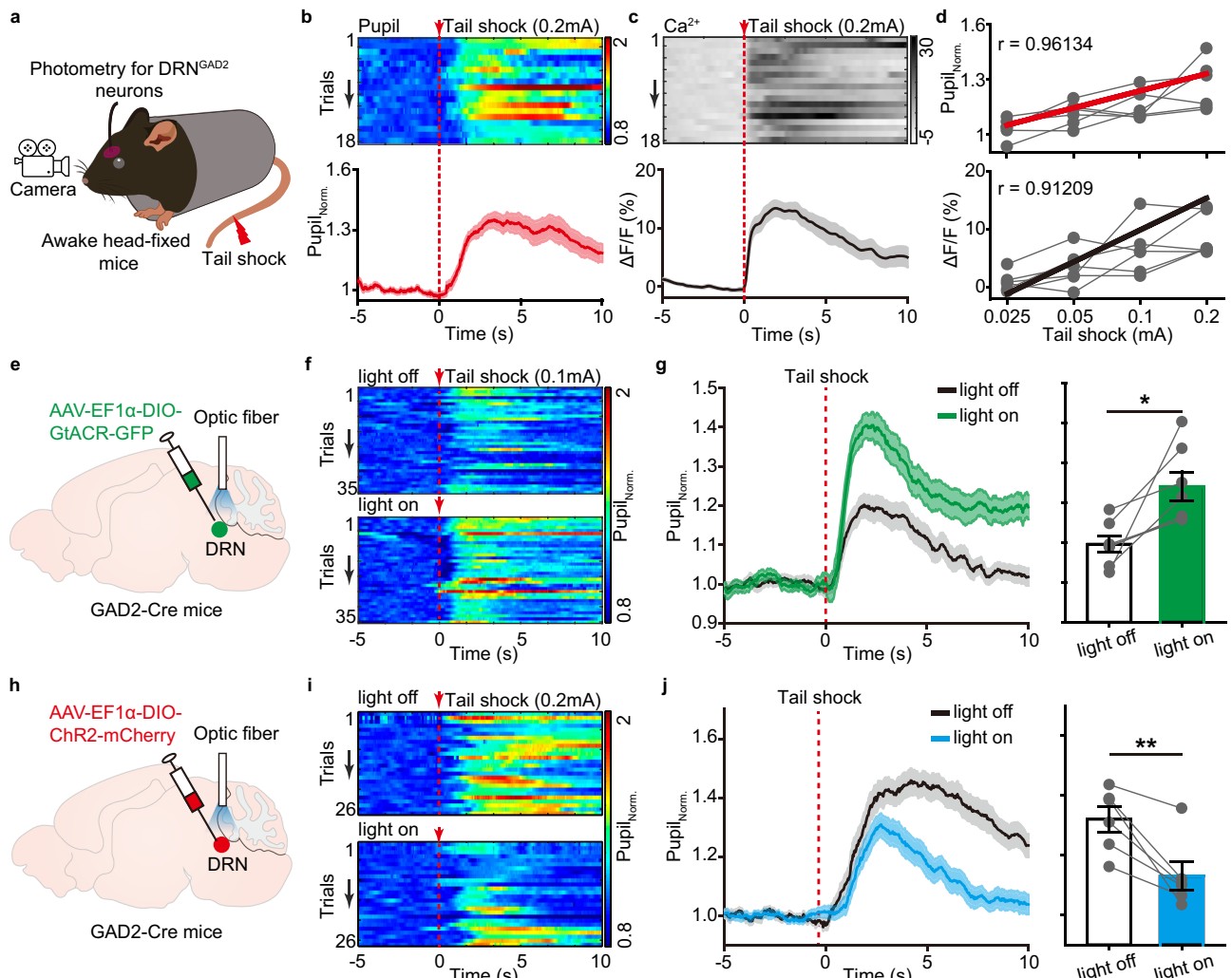

**Fig. 5 | DRN^GAD2 neurons control the arousal responses to tail shock. a** Schematic of experimental setup for photometry recording and delivery of tail shock in awake head-fixed mice. **b, c** Heatmaps (top) and average traces (bottom) of pupil size (**b**) or Ca²⁺ activities of DRN^GAD2 neurons (**c**) aligned to the onset of tail shock (vertical red lines). Pupil_Norm., normalized pupil size. **d** The effects of different amplitude of tail shock on pupil size (top) and Ca²⁺ activities of DRN^GAD2 neurons (bottom). *n* = 6 mice. **e** Schematic of viral injection for optogenetic inhibition of DRN^GAD2 neurons. AAV-EF1α-DIO-GtACR-GFP was injected into the DRN of GAD2-Cre mice. **f** Heatmaps showing tail shock induced-changes of pupil size in DRN^GAD2-GtACR mice. **g** Tail shock with simultaneous inhibition of DRN^GAD2 neurons induces more pronounced

increase of pupil size. *n* = 7 mice, Wilcoxon signed rank test, *W* = 28, *P* = 0.016. Shading in line graph represents ±SEM. **h** Schematic of viral injection for optogenetic activation of DRN^GAD2 neurons. AAV-EF1α-DIO-ChR2-mCherry was injected into the DRN of GAD2-Cre mice. **i** Heatmaps showing the effects of optogenetic stimulation on tail shock induced-changes of pupil size in DRN^GAD2-ChR2 mice. **j** Optogenetic activation of DRN^GAD2 neurons attenuates tail shock-induced increase of pupil size. *n* = 6 mice, two tailed unpaired *t* test, *t₅* = 4.391, *P* = 0.00708. Shading in line graph represents ±SEM. *\*P < 0.05, \*\*P < 0.01.* Data (bar graphs in **g**, **j**) are presented as mean ± SEM.

example, hypocretin neurons are most active during wakefulness and activation of hypocretin neurons increases wakefulness[52,53]. However, recent studies reported several neuronal populations with mismatched spontaneous activity and functional outcomes. Vgat-positive GABAergic neurons in the VTA (VTA^Vgat) are selective wakefulness-active whereas chemogenetic activation of these neurons induces sleep[25]. VTA^Vgat neurons do not promote physiological NREM sleep, but instead preventing hyperarousal via inhibition of VTA DA neurons and hypocretin neurons[25,28]. Chronically lesioning of VTA^Vgat neurons leads to mania-like state with hyperarousal and a decreased need for sleep[25,28]. Moreover, though DRN 5-HT neurons promote sleep in zebrafish and mice, these neurons exhibit the highest activity during the wakefulness state[50]. Mice with genetic ablation of DRN 5-HT neurons display reduced homeostatic sleep recovery, suggesting that DRN 5-HT neurons are part of the sleep homeostasis system and the high activity during wakefulness generates sleep pressure[50]. We here found that DRN^GAD2 neurons were wakefulness-active but inhibition or genetic

ablation of DRN^GAD2 neurons paradoxically increased wakefulness. Particularly, we found that ablation of DRN^GAD2 neurons increased wakefulness during the dark phase but not during the light phase. Considering that compensated mechanisms may occur during DTA-induced lesion period, the genetic ablation results highlighted an essential role of DRN^GAD2 neurons in controlling wakefulness in the dark phase, during which mice were more active and thus tight control over behavioral states was more needed. Indeed, we found that ablation of DRN^GAD2 neurons enhanced the increased wakefulness in acute stress conditions, whereas activation of stress-activated DRN^GAD2 neurons significantly constrained acute stress-induced increase of wakefulness. Thus, the heterogeneity of wakefulness/sleep system enables a tight control over brain states at different conditions or different timescales.

A rapid transition from sleep to wakefulness or an enhancement of arousal is a prerequisite for coping with acute stress situations[1,3]. Elevations of arousal to a certain level are beneficial for efficiently transforming information and making adapting behavioral responses[4].

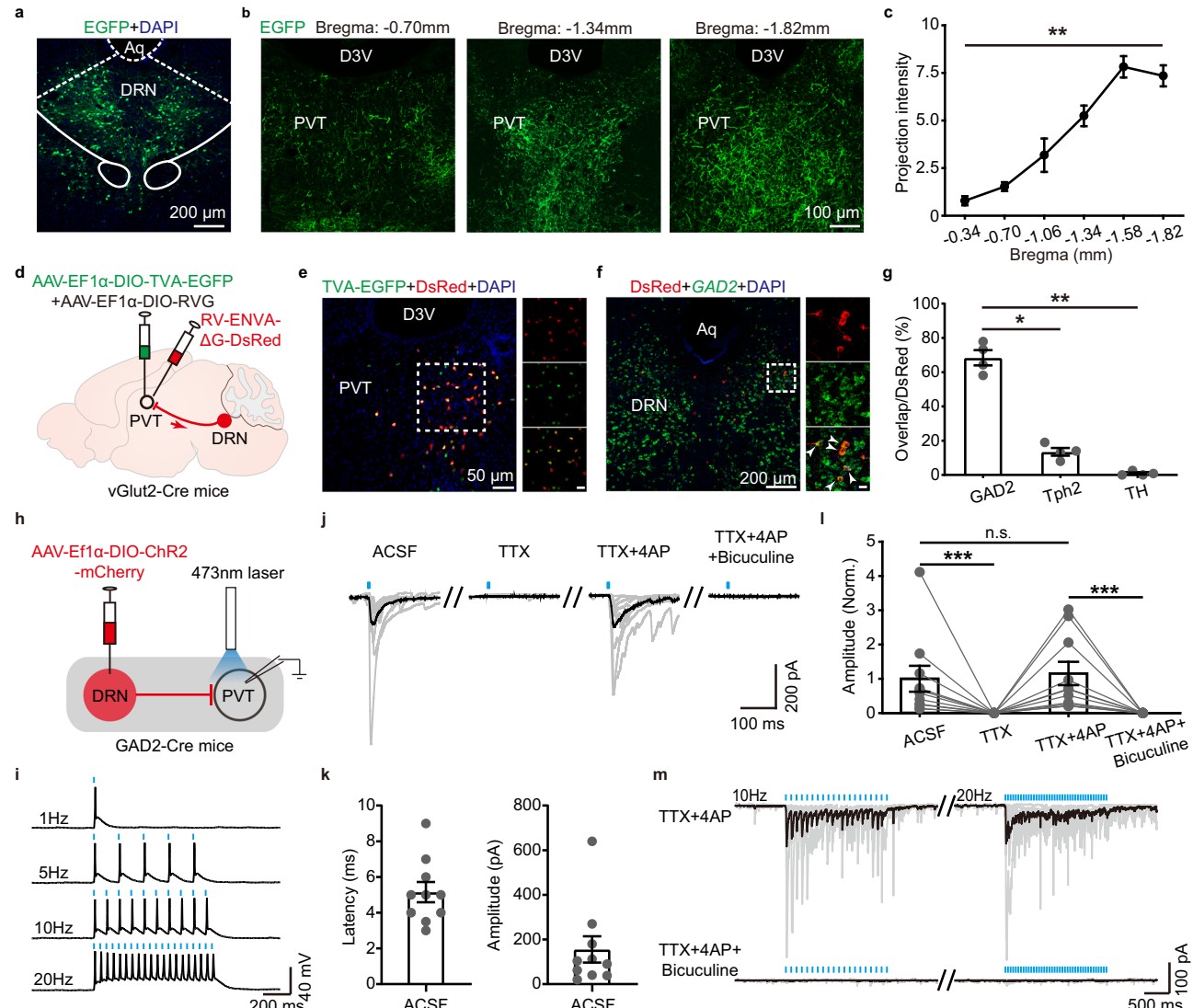

**Fig. 6 | DRN^GAD2 neurons send monosynaptic inhibitory inputs to PVT neurons.** **a** AAV-EF1α-DIO-EGFP expression in the DRN. Aq, aqueduct. **b** Representative images showing the distribution of axon fibers of DRN^GAD2 neurons in the PVT. D3V, dorsal third ventricle. **c** Quantification of relative projection density of DRN^GAD2 neurons along rostral to caudal axis of the PVT. *n* = 4 mice, Kruskal-Wallis one way ANOVA on ranks, *H* = 20.46, *P* = 0.001. **d** Schematic of viral injection for retrograde tracing the presynaptic inputs of PVT glutamatergic neurons. **e** Representative images showing the location of start cells in the PVT. Scale bar in the right panel is 20 μm. **f** Images showing the RV-dsRed-labeled DRN neurons and their co-localization with *GAD2*. Arrowheads indicate dsRed and *GAD2* co-labeled neurons. Scale bar in the right panels is 20 μm. **g** Quantification the percentage of *GAD2*-, Tph2-, or TH-positive DRN neurons projecting to PVT glutamatergic neurons. *n* = 4 mice, Kruskal-Wallis one way ANOVA on ranks following Student-Newman-Keuls multiple comparisons test, *H* = 9.881, GAD2 vs Tph2: *P* = 0.021; GAD2 vs TH: *P* = 0.005. **h** Schematic diagram of whole-cell recording in the PVT. AAV-DIO-EF1α-

ChR2-mCherry was injected into the DRN of GAD2-Cre mice. Recordings were made from PVT neurons with simultaneous optogenetic stimulation of terminals of DRN^GAD2 neurons. **i** Light-evoked action potentials of DRN^GAD2 neurons verify the expression efficacy of ChR2-mCherry. **j** Light-evoked postsynaptic currents in recorded PVT neurons. The dark lines indicate average traces and gray lines indicate responses of individual cells. *n* = 10 cells. ACSF artificial cerebrospinal fluid, TTX tetrodotoxin, 4-AP 4-Aminopyridine. **k** Latency (left) and amplitude (right) of light-evoked postsynaptic currents in ASCF conditions. *n* = 10 cells. **l** Summary of the amplitude of light-evoked currents normalized to that in ACSF. *n* = 10 cells, Friedman repeated measures ANOVA on ranks following Student-Newman-Keuls multiple comparisons test, *H* = 33.515, ACSF vs TTX: *P* = 5.4 × 10^−5; TTX + 4AP vs TTX + 4AP+ bicuculine: *P* = 5.4 × 10^−5; TTX vs TTX + 4AP: *P* = 0.9. Norm., normalized. **m** Bicuculine blocks the 10 Hz and 20 Hz light pulses-induced postsynaptic currents. *n* = 6 cells. **P* < 0.05, ****P* < 0.001, n.s., not significant. Data (**c**, **g**, **k**, **l**) are presented as mean ± SEM.

Neural circuitries promoting wakefulness or arousal in response to acute stress have been identified, including hypocretin neurons[9], corticotropin-releasing factor-expressing neurons in the paraventricular nucleus of the hypothalamus[9], glutamatergic/calretinin neurons in the PVT[10,54], and glutamatergic neurons in the median preoptic nucleus[55]. However, hyperarousal exceeding the limited information processing capacity could eventually induces a consequent loss of efficiency[6,56]. In this context, it is highly suggestive of a regulatory process, where the activation of de-arousal system counteracts an increase in arousal. In fact, we currently know little about the

circuitries downregulating the heightened arousal in acute stress situations. Here we found that DRN^GAD2 neurons were active during wakefulness and further activated in acute stress conditions. Our results were consistent with previous observations that social defeat- and corticosterone-induced stress increased c-Fos expression in DRN^GAD1/2 neurons[40,41]. In addition, activation of DRN^GAD2 neurons decreased acute restraint stress-induced increase of wakefulness and tail shock-induced elevation of arousal level. Mice with ablation or inhibition of DRN^GAD2 neurons displayed exaggerated arousal responses in acute stress conditions, indicating a role of DRN^GAD2 neurons in

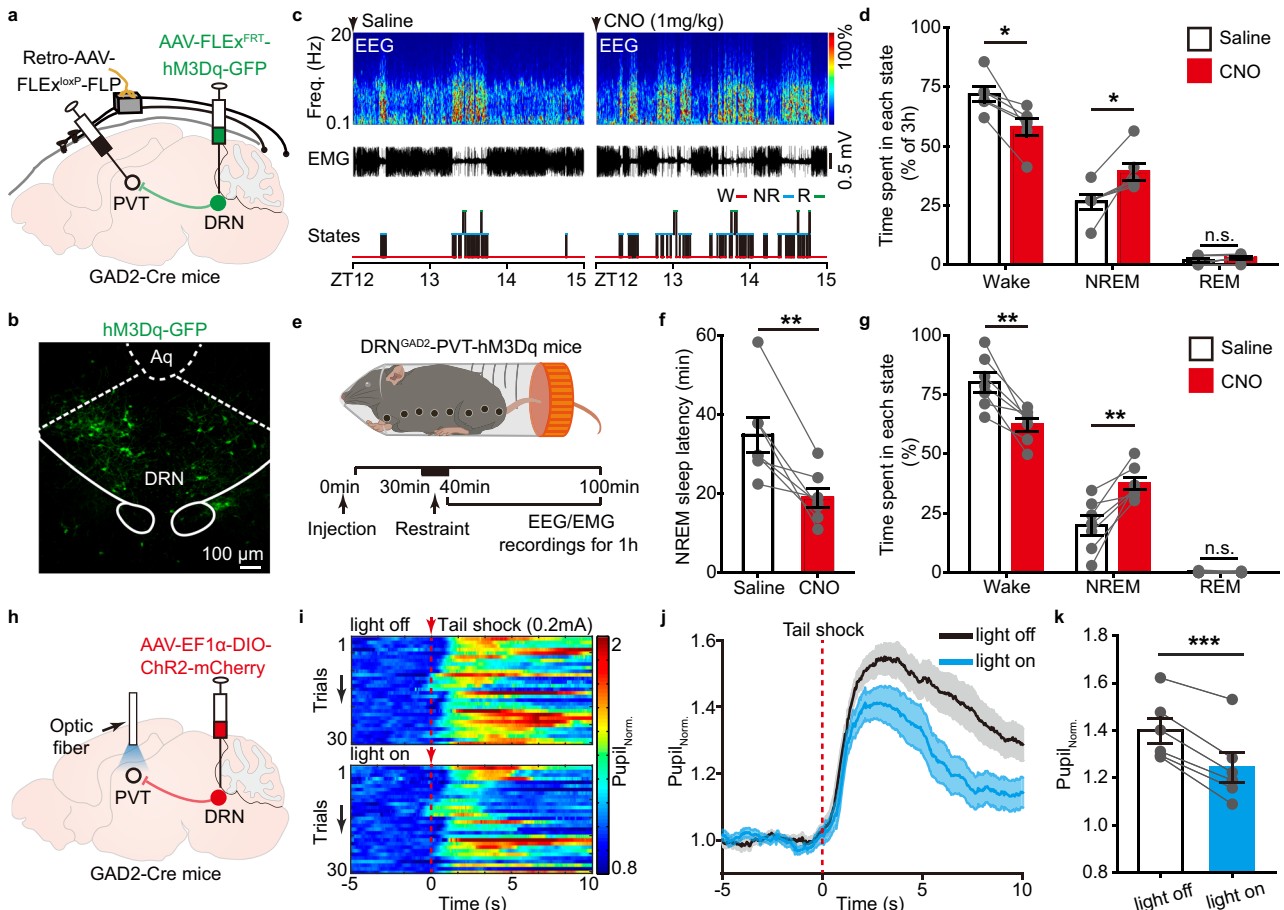

**Fig. 7 | The DRN$^{GAD2}$-PVT circuit constrains wakefulness in acute stress condition. a** Schematic of chemogenetic activation of PVT-projecting DRN$^{GAD2}$ neurons and EEG/EMG recordings. Retro-AAV-FLEx$^{loxP}$-FLP was injected into the PVT and AAV-FLEx$^{FRT}$-hM3Dq-GFP was injected into the DRN of GAD2-Cre mice, respectively. **b** Representative image showing the expression of hM3Dq-GFP in the DRN. Aq, aqueduct. **c** EEG power spectrogram, EMG traces, and hypnograms from a DRN$^{GAD2}$-PVT-hM3Dq mouse during 3 h post saline or CNO (1 mg/kg) injection. Freq., frequency; W, wake; NR: NREM; R: REM. **d**, Chemogenetic activation of PVT-projecting DRN$^{GAD2}$ neurons decreases wakefulness. $n = 6$ mice, two tailed paired $t$ test, wake: $t_5 = 3.701$, $P = 0.014$; NREM: $t_5 = 3.48$, $P = 0.017$; REM: $t_5 = 2.123$, $P = 0.0872$. **e** Schematic of a 10 min restraint session and following EEG/EMG recordings in DRN$^{GAD2}$-PVT-hM3Dq mice. **f** NREM sleep onset latency following a 10 min restraint

session. $n = 7$ mice, two tailed paired $t$ test, $t_6 = 4.2$, $P = 0.00569$. **g** Time spent in wake, NREM and REM sleep of DRN$^{GAD2}$-PVT-hM3Dq mice following restraint. $n = 7$ mice, two tailed paired $t$ test, wake: $t_6 = 4.555$, $P = 0.00387$; NREM: $t_6 = 4.653$, $P = 0.00349$. **h** Schematic of optogenetic activation of DRN$^{GAD2}$-PVT circuit. AAV-EF1α-DIO-ChR2-mCherry was injected into the DRN of GAD2-Cre mice and an optic fiber was placed above the PVT. **i** Heatmaps showing the effects of optogenetic stimulation on tail shock-induced changes of pupil size. **j**, **k** Optogenetic activation of DRN$^{GAD2}$-PVT circuit attenuates tail shock-induced increase of pupil size. $n = 6$ mice, two tailed paired $t$ test, $t_5 = 8.63$, $P = 3.45 \times 10^{-4}$. Shading in **j** represents ±SEM. *$P < 0.05$, **$P < 0.01$, ***$P < 0.001$, n.s., not significant. Data (**d**, **f**, **g**, **k**) are presented as mean ± SEM.

control arousal responses. Considering that the de-arousal process is also preparatory behaviors of sleep[18,56], thus it is reasonable that our artificial activation of DRN$^{GAD2}$ neurons induced NREM sleep. However, as we did not record the single-neuron spiking activity of DRN$^{GAD2}$ neurons in physiological or stress conditions, the parameters for functional manipulation (optogenetic simulation at 20 Hz for example) may not fully mimic the spontaneous activity of DRN$^{GAD2}$ neurons. Thus, the role of DRN$^{GAD2}$ neurons in wakefulness regulation, particular in stress conditions, need to be further investigated using activity pattern-based methods.

It's worth noting that two recent published studies reported DRN$^{Vgat}$ neurons to be wakefulness-active and wakefulness-promoting. Gazea et al. showed that DRN$^{Vgat}$ neurons are more active during wakefulness and stimulation of these neurons promotes wakefulness through reciprocal connection with GABAergic neurons in the LH[38]. In addition, Cai et al. consistently found that DRN$^{Vgat}$ neurons promote wakefulness, which is mediated via direct inhibition of VTA GABAergic neurons[39]. The present study found that DRN$^{GAD2}$ neurons were also wakefulness-active, but their activation decreased wakefulness.

Though both DRN$^{GAD2}$ neurons and DRN$^{Vgat}$ neurons were wakefulness-active, these neurons exhibited biased activity pattern during sleep. DRN$^{Vgat}$ neurons are more active during REM sleep than that during NREM sleep[38,39] and slowly increase their activity during transitions from NREM sleep to REM sleep[39], whereas no obvious activity changes of DRN$^{GAD2}$ neurons had been observed during NREM sleep and REM sleep in present study. The difference of activity pattern and functional outcomes of DRN$^{GAD2}$ neurons and DRN$^{Vgat}$ neurons suggested that these two DRN neuronal populations have different roles in wakefulness/sleep regulation. To explore the possible reasons responsible for these differences of DRN$^{GAD2}$ and DRN$^{Vgat}$ neurons, we first characterized the specificity of targeting DRN$^{GAD2}$ neurons using GAD2-Cre mice. *GAD2* have been reported to be expressed in a subset DRN 5HT neurons[57–59]. Our FISH results showed that *GAD2* was indeed expressed in a 17.25% of DRN 5HT neurons and only in 3.24% of DRN DA neurons, but the expression level of *GAD2* in 5HT neurons was less than that in non-5HT neurons (Supplementary Fig. 13). However, targeting DRN$^{GAD2}$ neurons by injection of AAV-EF1α-DIO-EGFP into GAD2-Cre mice revealed rarely EGFP-labeled DRN$^{GAD2}$ neurons colocalizing with 5HT

neurons (0.43%) or DA neurons (0.27%) (Supplementary Fig. 14), which was in concert with published results using viral vectors to label DRN$^{GAD2}$ neurons in GAD2-Cre mice[60]. Though it is not known why the above two strategies resulted in different co-labeling ratio between GAD2 and Tph2, these results at least suggested that targeting DRN$^{GAD2}$ neurons using GAD2-Cre mice could specifically label an independent population of DRN neurons.

We next explored the expression pattern of *Slc32a1*(*Vgat*), *GAD1* and *GAD2* in the DRN using FISH. Results showed that *GAD2*-positive neurons represented the largest proportion (95.58 ± 1.32%) of DRN GABAergic neurons (Supplementary Fig. 15a, b). Almost all (99.52%) *Vgat*-positive neurons expressed *GAD2* whereas 79.85% of *GAD2*-positive neurons expressed *Vgat*. Notably, there was a small population (10.71% of all DRN GABAergic neurons) of *GAD2*-positive neurons without *Vgat* expression (Supplementary Fig. 15c). Actually, these results revealed an underestimated complexity of DRN GABAergic neurons, suggesting that targeting DRN GABAergic neurons using Vgat-Cre or GAD2-Cre mice may result in labelling different populations of GABAergic neurons. In addition, the different output effector targets of these two GABAergic populations should also be taken into consideration when regarding their functional difference. DRN$^{Vgat}$ neurons promoted wakefulness through direct inhibition of NREM-sleep promoting VTA$^{Vgat}$ neurons or indirectly disinhibition of wake-promoting neurons via LH GABAergic neurons[38,39]. Here, we found that DRN$^{GAD2}$ neurons constrained wakefulness through direct inhibition of wakefulness-promoting PVT neurons. Considering that VTA- or LH-projecting DRN$^{Vgat}$ neurons are proved to be non-overlapping[39], thus future studies are needed to investigate the role of molecular defined- or projection specific-subtypes of DRN GABAergic neurons in wakefulness regulation. Moreover, since parvalbumin (PV)-positive GABAergic neurons have been demonstrated to be wakefulness-active and play an opposite role as GAD2 neurons in substantia nigra pars reticulata[17], we also searched the expression of PV-positive neurons in the DRN. However, PV-positive neurons were rarely observed in the DRN (Supplementary Fig. 16).

In conclusion, our study elucidated the DRN$^{GAD2}$ neurons constrained wakefulness in a mouse model of stress. These neurons might be a possible target for the therapeutic intervention in treating hyperarousal-related neuropsychiatric disorders.

## Methods

### Animals

All experimental protocols were in accordance with Army Medical University Guide for the Care and Use of Laboratory Animals. In all experiments, adult (8–12 weeks, 22–26 g) male or female GAD2-IRES-Cre mice (The Jackson Laboratory, stock number: 010802), vGlut2-IRES-Cre mice (The Jackson Laboratory, stock number: 028863), and WT C57BL/6 mice were used. Mice were housed at controlled environmental temperature (22 ± 1 °C), humidity (-50%), and 12 h light/12 h dark cycle (light on between 6 am (ZT0) and 6 pm). Mice were group (2–5)-housed with *ad libitum* access to food and water.

### Stereotaxic injection of AAV

Eight- to 12-week-old mice were anaesthetized with isoflurane (2% for induction and 1–1.5% for maintenance). Surgery procedures were performed according to published studies[10,46]. After anesthesia, mice were placed on a stereotaxic apparatus (RWD Life Technology, China) and eyes were protected with ophthalmic ointment. A heating pad was used to maintain the body temperature throughout the surgery. After shaving the skin on the head, a longitudinal incision along the midline was made to get access to the skull. Connective tissue in the skull surface was cleaned with cotton swabs. For viral injection, small craniotomy holes (-1 mm diameter) were drilled above the targeted brain regions. Injections were performed using a micro glass pipette (tip diameter, -20 μm) filled with mineral oil in a Nanoject III (Drummond,

USA). A volume of 100 nl virus was injected and the injection rate was adjusted to 20 nl/min. After injection, the pipette was left in situ for 5 min to allow diffusion and slowly withdrawn. To chemogenetic or optogenetic activation of DRN$^{GAD2}$ neurons, AAV-EF1α-DIO-hM3Dq-mCherry (BrainVTA Technology, China), AAV-EF1α-DIO-ChR2-mCherry (Obio Technology, China) or AAV-EF1α-DIO-mCherry (BrainVTA Technology, China) was injected into the DRN (bregma, antero-posterior (AP) = − 4.55 mm; medio-lateral (ML) = 0.00 mm; dorsal-ventral (DV) = −2.22 mm). To selectively ablate DRN$^{GAD2}$ neurons, AAV-EF1α-DIO-DTA (BrainVTA Technology, China) or AAV-EF1α-DIO-mCherry was injected into the DRN. To chemogenetic inhibition of DRN$^{GAD2}$ neurons, AAV-EF1α-DIO-hM4Di-mCherry (Obio Technology, China) or AAV-EF1α-DIO-mCherry was injected into the DRN. For photometry recording, AAV-EF1α-DIO-jGCaMP7b (BrainVTA Technology, China) was injected into the DRN. To selectively label and chemogenetically activate acute restraint stress-activated DRN$^{GAD2}$ neurons using c-Fos-based TetTagging method[44], a mixture (1:1) of AAV-cFos-tTA and AAV-TRE-DIO-hM3Dq-mCherry (BrainVTA Technology, China) was injected into the DRN of GAD2-Cre mice. To optogenetic inhibition of DRN$^{GAD2}$ neurons, AAV-EF1α-DIO-GtACR-GFP (Taitool Bioscience, China) was injected into the DRN. To map the axon projections of DRN$^{GAD2}$ neurons, AAV-EF1α-DIO-EGFP (BrainVTA Technology, China) was injected into the DRN. To selectively chemogenetic activate the PVT-projecting DRN$^{GAD2}$ neurons, Retro-AAV-FLEx$^{loxP}$-FLP (Taitool Bioscience, China) was injected into the PVT and AAV-FLEx$^{FRT}$-hM3Dq-GFP (BrainVTA Technology, China) was injected into the DRN of GAD2-Cre mice. To chemogenetic activate the PVT-projecting DRN$^{GAD2}$ neurons with simultaneous photometry recording of PVT neurons, a mixture of AAV-CaMKIIα-GCaMP6f (BrainVTA Technology, China) and Retro-AAV-FLEX$^{loxP}$-FLP was injected into the PVT, and AAV-FLEx$^{FRT}$-hM3Dq-GFP was injected into the DRN of GAD2-Cre mice. To investigate the monosynaptic inputs of PVT glutamatergic neurons, AAV-EF1α-DIO-TVA-EGFP (BrainVTA Technology, China) and AAV-EF1α-DIO-RVG (BrainVTA Technology, China) was injected into the PVT of vGlut2-Cre mice. RV-ENVA-DsRed (BrainVTA Technology, China) was injected 3 weeks later and mice were perfused 1 week after injection. After recovery from anesthesia, mice were left for 3-4 weeks before behavioral experiments or perfused for histology.

### Surgery

For EEG/EMG recordings, stainless steel screws attached with wires were used to collect EEG/EMG signals. Screws were implanted on the top of frontal cortex (bregma, AP = +1.50 mm; ML = ± 1.50 mm) and parietal cortex (bregma, AP = −3.0 mm; ML = +2.0 mm). Two bar-ended steel wires were sutured into the trapezius muscle of the neck to record EMG signals. All EEG/EMG electrodes were previously soldered to a micro-pin connector. Connectors, screws, and steel wires were all first secured to the skull with a thin layer of C&B Metabond (Parkell, USA). After that, common dental cement was applied to stabilize the implantation. Mice were allowed a minimum of 7 days to recover in the home cage before recordings.

For optogenetic activation or inhibition of DRN$^{GAD2}$ neurons, optic fibers (NA: 0.37, 200 μm diameter, Inper Technology, China) were bilaterally implanted above the DRN (AP = −4.55 mm; ML = ± 0.80 mm with a 10° angle; DV = −2.10 mm). For fiber photometry recording of Ca$^{2+}$ activities of DRN$^{GAD2}$ neurons, an optic fiber was advanced above the DRN (AP = −4.55 mm; ML = +0.80 mm with a 10° angle; DV = −2.10 mm). For optogenetic activation of axon terminals of DRN$^{GAD2}$ neurons in the PVT, an optic fiber was implanted above the PVT (AP = −1.20 mm; ML = 0 mm; DV = −2.35 mm).

### Polysomnographic recordings and analysis

Sleep recordings were performed in sound-proofed recording cages to which mice had been habituated for at least 3 days. Flexible recording cables were connected to the micro-pin connectors previously

implanted on the mice head. A slip ring was used to ensure mice to freely move. Cortical EEG signals and EMG signals of neck muscles were amplified and band-pass filtered (EEG: 0.3–30 Hz, EMG: 10–1000 Hz). EEG/EMG signals were digitized at a sampling rate of 512 Hz and recorded using Vital Recorder software (Kissei Comtec, Japan). To determine the sleep/wake states of mice, EEG/EMG signals were off-line processed with SleepSign software (Kissei Comtec, Japan) by 4-s epochs according to established criteria[46]. Wakefulness was classified as desynchronized, low-amplitude EEG rhythms and high EMG activity. NREM sleep encompassed with synchronized, high amplitude and low frequency (0.5–4 Hz, delta) EEG activity and lower EMG activity compared with wakefulness without phasic bursts. REM sleep contained pronounced theta (4–10 Hz) rhythm with nearly no EMG activity (muscle atonia). The automatically defined brain states were checked by experimenters who were blinded to the experimental manipulations and corrected if necessary. To analyze the EEG spectral activities, raw EEG signals were calculated with fast Fourier transforms (FFTs) in consecutive 4-s epochs in the frequency range of 0–30 Hz. Epochs containing high amplitude movement artifacts with excluded for spectral analysis. Normalized EEG power was calculated as ratio of power in specific range over the sum of power in the entire analysis range.

## Chemogenetic and optogenetic manipulations

For chemogenetic experiments, recorded mice were habituated to experimenter handling for 10–15 min each day for at least 3 consecutive days. After habituation, saline or CNO (Tocris, 1396/10) was injected intraperitoneally at a random order into corresponding mice expressing hM3Dq, hM4Di, or mCherry in the target brain regions. For DRN$^{GAD2}$-hM3Dq-mCherry mice, saline or CNO (1 mg/kg) was injected at the beginning of dark phase (ZT12) or light phase (ZT0). For DRN$^{GAD2}$-hM4Di-mCherry mice, saline or CNO (2 mg/kg, 5 mg/kg) was injected at ZT0 or ZT12. DRN$^{GAD2}$-mCherry mice received saline or CNO (5 mg/kg) injection at ZT12. For DRN$^{GAD2}$-TRE-hM3Dq-mCherry mice, saline or CNO (1 mg/kg) was injected at ZT12. For DRN$^{GAD2}$-PVT-hM3Dq-GFP mice, saline or CNO (1 mg/kg) was injected at ZT12.

For optogenetic activation of DRN$^{GAD2}$ neurons with EEG/EMG recording, mice implanted with optic fibers were connected to a 473 nm laser generator (Fiblaser Technology, China). The laser pulses were controlled by Master-8 (Jerusalem, Israel). Each laser trial consisted of a 20 Hz pulse train (10 ms per pulse) lasting for 120 s. The laser intensity at the tip of optic fibers was adjusted to 10 mW according to previous published studies using optogenetic method to activate DRN neurons[38,51]. Laser pulses were delivered at ZT4 to ZT9 during the light phase or ZT14 to ZT16 during the dark phase of mice with 15–20 min inter-trial interval, regardless of which wakefulness/sleep state the mice were in. For optogenetic activation of DRN$^{GAD2}$ neurons or DRN$^{GAD2}$-PVT pathway in tail shock experiments, 20 Hz (10 ms per pulse, 10 mW) laser pulses were delivered for 15 s (5 s before and 10 s after the onset of tail shock). For optogenetic activation of DRN$^{GAD2}$ neurons in the open field chamber, 20 Hz (10 ms per pulse, 10 mW) laser pulses were delivered for 3 min. For optogenetic inhibition experiment, mice with GtACR expression were connected to a 473 nm laser generator. Constant light (10 mW) was delivered for 10 s with the onset of tail shock.

## Fiber photometry recording and analysis

After recovery from the surgery, mice implanted with optic fibers in the DRN or PVT were connected to the experimental setups. Ca$^{2+}$ signals were collected using Fiber Photometry System (Inper Technology, China). A 488 nm excitation laser was used to excite jGCaMP fluorescence signals and the intensity of laser at the tip of fiber was adjusted to ~40 μW. A 410 nm excitation laser was served as control for movement artifacts and fluorescence bleaching. Data acquisition was conducted at 40 Hz. Ca$^{2+}$ signals were analyzed offline in synchronization with EEG/EMG, pupil size recording videos, or the onset of tail shock.

Ca$^{2+}$ signals were analyzed according to published methods[10,39]. The Ca$^{2+}$ signals were first processed by baseline correction and motion correction using Inper Data Process (Inper Technology, China). The 410 nm trace was subtracted from 488 nm trace using least-squares regression. The Polynomial Fitted correction was used to minimize the bleaching effect caused by long-term recording. For the analysis of Ca$^{2+}$ activity of DRN$^{GAD2}$ neurons across wakefulness/sleep cycles, fiber photometry and EEG/EMG data were collected between ZT2 and ZT12. Each mice recorded 3-4 sessions and each session lasted for 0.5–1 h. Sessions consist vigilance states of wakefulness, NREM, and REM sleep were selected for further analysis. EEG/EMG and fiber photometry data were time aligned according to the simultaneous TTL markers. Wakefulness/sleep states were classified based on EEG/EMG and Ca$^{2+}$ signals in different states (wake, NREM sleep, REM sleep) and during the state transitions were analyzed. Changes of Ca$^{2+}$ signals were calculated as $\Delta F/F = (F - F_0)/F_0$. $F_0$ was the average value of fluorescence signal from a stable NREM sleep (>30 s) from each recording session per mouse. F was the average value of fluorescence signal of a specific wakefulness/sleep state. We also calculated and compared the Ca$^{2+}$ signals of DRN$^{GAD2}$ neurons in long (>1 min) and short (<1 min) wake. In addition, the correlations between Ca$^{2+}$ signals of DRN$^{GAD2}$ neurons and EEG power activity in delta or theta band during 1 s epochs of each state was analyzed.

## Pupil size recording and analysis

Pupil size was recorded in awake head-fixed mice. A custom-made head bar was firmly glued to the skull. After recover from the surgery, mice were acclimated to the head-fixation setups via the implanted head bar according to published work[61]. The body of mice was inserted into an acrylic tube which installed on a kinematic base. The head of mice was extended out of the tube with its front paws to grip to the edge of the tube. Mice were habituated to the head-fixation setups 0.5–1 h per day for 3–5 days. After mice were fully acclimated to head-fixation procedure, a CMOS camera equipped with infrared LED array was used to capture the image of either the left or right eye depending on the experimental setup. The ambient illumination was controlled at illuminance of ~60 lx to keep the pupil constricted between the two eyelids. The image acquisition rate was 30 Hz. The size of pupil was analyzed offline using an open-source software Bonsai (http://bonsai-rx.org/)[62]. The black pupil on a gray iris background were extracted using a threshold adjusted for each session. The diameter of pupil size was calculated from binarized pupil image fitted with a least square fit of ellipse. For each session, a few images were dropped due to eye blinking or excessive micro-saccades. The corresponding pupil diameter was estimated based on the average of five images before the eye blink.

To analyze correlation between Ca$^{2+}$ signals of DRN$^{GAD2}$ neurons and pupil size, we first isolated spontaneous pupil dilations according to previous published methods[29,63]. Ca$^{2+}$ signals and pupil data were resampled to 20 Hz and pupil data was then low-pass filtered to 0.5 Hz using a second-order Butterworth filter. The pupil data was normalized to the maximum diameter and presented in a range between 0 and 1. Candidate pupil dilation events were identified by the sign of the calculated pupil derivative. Only pupil derivative >0 and the changes in pupil diameter >10% were used for subsequent analysis. Candidate pupil dilation events shorter than 1 s duration from onset to offset were removed from further analysis. The pupil dilation onset time points were then used to extract corresponding Ca$^{2+}$ data from each aligned time series data. Ca$^{2+}$ and pupil data 3 s before and 5 s after pupil dilation were analyzed. To evaluate the general association between Ca$^{2+}$ signals and pupil size, we first down-sampled the sampling frequency of pupil size to that of Ca$^{2+}$. We then computed the Pearson's linear correlation coefficients between pupil size and Ca$^{2+}$ by using a sliding window with a window width of 60 s and no overlapping. We used histogram to show the distribution of the R values for Pearson's correlation. The correlation was significant if $P < 0.05$.

## Behavioral protocols

**Open field behavior.** Mice were placed into an open field chamber (60 cm × 60 cm × 60 cm) and video was recorded (25 Hz). For optogenetic experiment, mice were allowed to explore a 9 min session, which consist of 3 min baseline +3 min light on +3 min light off. For genetic ablation or chemogenetic inhibition of DRN[GAD2] neurons experiment, DTA, hM4Di or mCherry mice were allowed to explore the chamber for 10 min. The traveled distance of mice was analyzed using custom-written MATLAB script. For chemogenetic inhibition experiments, injection of saline or CNO was separated at least five to seven days. Mice were introduced to open field chamber 30 min after injection.

**Acute restraint.** The acute restraint experiment was performed by a trained experimenter at ZT12. Mice were habituated to experimenter's handing for 3 to 5 days. Mice were gently placed into a 50 mL centrifuge tube with holes opening on the top. The state of mice was closely monitored in the restraint session to make sure proper ventilation. After ending of a 10 min restraint session, mice were placed back into their home cages. For chemogenetic experiments, mice received saline or CNO injection 30 min before restraint. For c-Fos immunostaining experiment, DRN[GAD2]-mCherry mice were perfused 60 min later. For EEG/EMG recording, mice were immediately connected to the recording chamber once released from the tube.

**Tail shock.** The tail shock experiment was performed during the light on phase (ZT3-ZT9). Mice were habituated to the head-fixation setups and experimenter's handing for 3 to 5 days. Following habituation to the head fixation apparatus, mice received presentations of unsignaled tail shock. For optogenetic activation of DRN[GAD2] neurons experiment, the current amplitude of tail shock was 0.2 mA. 20 Hz stimulation was delivered 5 s before the onset of tail shock and lasted for 15 s. For optogenetic inhibition of DRN[GAD2] neurons experiment, the current amplitude of tail shock was 0.1 mA. Continuous stimulation was simultaneously delivered with onset of tail shock and lasted for 10 s. Each mouse in the optogenetic experiment received a total of 8 to 10 tail shocks, half of the shocks in the presence of laser illumination.

## Patch-clamp recording

AAV-EF1α-DIO-ChR2-mCherry (100 nl) or AAV-EF1α-DIO-GtACR-GFP was injected into the DRN of GAD2-Cre mice and recording was made 4-6 weeks after injection. Slice containing targeted brain regions was prepared according to published work[46]. Briefly, mice were deeply euthanized with isoflurane and sacrificed. Brains were removed carefully following decapitation and submerged into ice-cold oxygenated (95% $O_2$ and 5% $CO_2$) artificial cerebrospinal fluid (ACSF; in mM,124 NaCl, 3 KCl, 26 NaHCO₃, 2 MgCl₂, 2 CaCl₂, 10 Glucose, adjusted to pH 7.2–7.4). Brains were blocked, and an oscillating tissue slicer was used to cut 300μm-thick coronal sections containing the DRN or PVT. Sections were recovered at 32–34 °C for 15 min in oxygenated N-methyl-D-glucamine (NMDG)-incubation solution (in mM, 110 NMDG, 110 HCl, 2.5 KCl, 1.2 NaH₂PO₄, 25 NaHCO₃, 25 Glucose, 10 MgSO₄, and 0.5 CaCl₂). Sections were then transferred to oxygenated ACSF and incubated for 1 h at room temperature (22–24 °C). During recording sessions, sections were transferred to and submerged in a recoding chamber, where oxygenated ACSF was continuously perfused.

Whole-cell patch-clamp recordings were obtained from cell bodies of DRN neurons or PVT neurons. GtACR-GFP- or ChR2-mCherry-expressing DRN neurons were identified and visualized with an upright microscope (Olympus, Japan) equipped with differential contrast optics, a 40× water immersion objective, and an infrared video imaging camera. The resistance of the patch pipette was 3–5 MΩ. For verify the functional expression of GtACR-GFP or ChR2-mCherry in DRN[GAD2] neurons. The pipette was filled with an internal solution (in mM, 125 potassium gluconate, 20 KCl, 10 HEPES, 1 EGTA, 2 MgCl₂·6H₂O, 4 ATP,

adjusted to pH 7.2–7.4 with 1 KOH). Neurons were held at a membrane potential of -60mV at least 5 min for stabilization. Neurons were excluded if the series resistance exceeded 25MΩ or increased by >15% during the recording. To activate GtACR-GFP, neurons were held at −45 mV to elicit action potentials and 473 nm laser pulse lasting 30 s were delivered via an optical fiber placed above the recorded neuron. To activate ChR2-mCherry, neurons were held at −60 mV and 5 ms 473 nm laser pulses were delivered. For characterization of DRN[GAD2] neurons to PVT connections, the pipette was filled with an internal solution (in mM, 135 CsCl, 10 HEPES, 2 MgCl₂·6H₂O, 0.1 EGTA, 2 Na₂-ATP, 0.2 Na₂-GTP). Postsynaptic currents were evoked by delivering 5 ms 473 nm laser pulses to illuminate the PVT. The laser intensity was adjusted to 5 mW. Tetrodotoxin (TTX, 1 μM, MLC, MBZ10175), 4-aminopyridine (4AP, 2 mM, Tocris, 0940), and bicuculine (10 μM, MCE, HY-N0219) were bath applied. Data were obtained using EPC 10 amplifier with PatchMater (HEKA Elektronik, Lambrecht/Pfalz, Germany) and analyzed with Pulse/Pulsefit v.8.74.

## Histology

Mice were deeply anesthetized with vaporized isoflurane and transcardially perfused with 50 mL 0.1 M phosphate buffer saline (PBS), followed by 4% paraformaldehyde (PFA) in PBS. Brains were carefully extracted and post-fixed with 4% PFA at 4 °C overnight. Brains were then stored in 30% sucrose by volume in PBS solution until brains were saturated for at least 24 h. After embedding and freezing, coronal brain slices were cut using a freezing microtome (CM1950, Leica).

For immunohistology, 30 μm brain slices were washed three times (10 min each time) in PBS and then transferred into blocking solution (Beyotime, P0260) and incubated at 37 °C for 30 min. Slices were subsequently incubated with primary antibodies (TH, 1:1000, Abcam, ab112; Tph2: 1:500, Abcam ab184505; parvalbumin, 1:500, Abcam, Ab181086) dissolved in dilution solution overnight at 4 °C. Afterwards, slices were washed in PBS and incubated with a species-specific secondary antibody (Alexa Flour 488 donkey anti-rabbit lgG 1:800, A21207, Invitrogen, USA) at room temperature for 2 h. After washing with PBS, slices were mounted on glass microscope slides, dried and cover-slipped with a DAPI-containing mounting media (F6057, Sigma, USA).

For FISH experiments, brain slices used for FISH experiment was sectioned into 16μm and collected onto Superfrost Plus Slides (Daigger Scientific). Sections were dried at 60 °C for 1 h and subsequently transferred to −80 °C freezer until use. FISH was performed using the RNAscope Multiplex Fluorescent Assays V2 (Advanced Cell Diagnostics, 323100). To validate the cell-type specify of injected virus infection, RNA-Protein Co-Detection Ancillary Kit (Advanced Cell Diagnostics, 323180) was used. The experimental procedures were performed according to the manufacturer's instructions. The following probes were used: *GAD2* (Advanced Cell Diagnostics, 439371), *Fos* (Advanced Cell Diagnostics, 506921), *Slc32a1* (Advanced Cell Diagnostics, 319191), *GAD1* (Advanced Cell Diagnostics, 400951).

Images were taken using LSM 800 laser scanning confocal microscope (Carl Zeiss) and further processed by ZEN 2012 software. Cell counting and axonal fiber intensity analysis were performed using ImageJ software (Fiji) by two independent investigators.

## Statistical analysis and reproducibility

Data were presented as mean ± standard error of the mean (SEM). Statistical analyses were performed using SigmaPlot 14.0. GraphPad Prism 8.0, and Matlab 2021a. Shapiro-Wilk normality test was initially performed to determine the appropriateness of the statistical tests used. Two-tailed paired *t* test was used for comparison between two groups quantification of time spent in wakefulness/sleep in chemogenetic and optogenetic experiments, Ca²⁺ activities of DRN[GAD2] neurons during states transitions, pupil recording in response to tail shock, open field with chemogenetic or optogenetic manipulations.

Two-tailed unpaired *t* test was used for comparison between two groups quantification of time spent in wakefulness/sleep in genetic ablation experiments and c-Fos staining experiments. One way ANOVA was used for comparison between groups quantification of $Ca^{2+}$ activities of $DRN^{GAD2}$ neurons in different wakefulness/sleep states, $DRN^{GAD2}$ neurons projection density in the PVT, and RV-DsRed-labeled neurons in the DRN. Two way repeated measure ANOVA was used for comparison between groups quantification of time spent in wakefulness/sleep in time course graphs. If the data didn't pass the Shapiro-Wilk normality test, non-parametric analysis methods, including Wilcoxon signed rank test, Kruskal-Wallis one way ANOVA on ranks and Friedman repeated measures ANOVA on ranks were used. Statistical significance was set at $*P < 0.05$, $**P < 0.01$, and $***P < 0.001$. The images in Figs. 1b, c, h, i; 2b, c, h; 3b, c; 4b, h; 6a, b, e, f; and 7b are representative ones from mice we collected data in individual experiments. The reproducibility of images for behavioral experiments is consistent with the actual number of experimental animals. The repeatability of images for statistical analysis is consistent with the number of animals.

### Reporting summary

Further information on research design is available in the Nature Portfolio Reporting Summary linked to this article.

## Data availability

The complete raw data from EEG/EMG recording, fiber photometry recording, behavioral experiments, and histological experiments are available upon request from the corresponding author. Source data underlying Figs. 1–7 and Supplementary Figs. are available as a Source Data file and shared in https://doi.org/10.6084/m9.figshare.24779127. Source data are provided with this paper.

## Code availability

Custom codes for locomotor activity analysis in open field test are available on GitHub.

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

## Acknowledgements

We thank all the members from Zhian Hu's lab for critical comments on the paper. This work was supported by the National Natural Science Foundation of China (31921003) to Z.H., National Major Project of China Science and Technology Innovation 2030 for Brain Science and Brain-Inspired Technology (2022ZD0205600) to S.R. and (2021ZD0203400) to Z.H., and National Natural Science Foundation of China (82371493) to S.R., (32022030) to C.H., and (82071492) to F.Y.

## Author contributions

Z.H., S.R., F.L., and C.H. conceived the project and designed the experiments. S.R., C.Z., and F.Y. performed the EEG/EMG recording, optogenetic and chemogenetic experiments, and fiber photometry recording. S.R., Z.S, Z.Z., and N.W. performed the behavioral experiment. S.R., F.Y. J.T., Y.Z., Y.F., Z.S., W.Z., and Y.W. performed histology and FISH experiments. S.R., N.W., and Z.S. performed patch-clamp recording experiments. S.R., C.Z., C.J., and J.X. analyzed the EEG/EEG data. S.R., X.Z., and H.Q. analyzed the behavioral data. All authors contributed to the data interpretation. S.R., Z.H., C.H., and F.L. wrote the paper wtih the help of all co-authors. All authors read and commented on the paper.

## Competing interests

The authors declare no competing interests.
