## [Peer Review File · Nature Communications]

REVIEWER COMMENTS

Reviewer #1 (Remarks to the Author):

Summary:

Combining chemogenetic and optogenetic manipulation, fiber photometry recording, anatomical and functional circuit mapping, and behavioral paradigms, Ren et al. investigates whether and how DRN GAD2 neurons constrained wakefulness in physiological and allostatic challenging conditions. Overall, it is a highly innovative study. Previous studies showed that DRN Vgat-positive GABAergic neurons strongly promote arousal. Quite different from previous studies, Ren et al's study demonstrated that another group of DRN GABAergic neurons, GAD2-positive GABAergic neurons, promote sleep. More interestingly, these sleep-promoting neurons were highly-active during wakefulness in mice, which is quite different from most currently-known sleep-promoting neurons showing high-activity during sleep. Authors' further results showed that stressful stimuli significantly increased the activity of sleep-promoting DRN GAD2 neurons, and activation of DRN GAD2 neurons promotes sleep in acute stress condition. The authors proposed that DRN GAD2 neurons regulate sleep by constraining wake-promoting paraventricular thalamus in physiological and allostatic challenging conditions. These findings not only further elucidate the regulatory effects of DRN GABAergic neurons on sleep-wake behavior, but also innovates the sleep-regulating theory, which is of great interest to the sleep field. However, there are some problems in the experimental design (eg. lack of control experiments) which weaken the soundness of conclusions in this study. The comments below aim to help to improve the soundness of the conclusions and make the conclusions more convincing.

Major comments:

1. Considering that the GAD2 neurons in the PAG have been shown to promote sleep (Weber, F. et al. 2018. doi:10.1038/s41467-017-02765-w), and line 121 mentions that there are few ectopic expressions in adjacent PAG in Figure 1b, the authors should provide more anatomical details for Figure 1 b (e.g. a profile of more brain sections showing the detailed expression extent of AAV-ef1 α -DIO-hM3D-mCherry). Similar to Figure 1b, please provides detailed expression extent of AAV-ef1 α -DIO-hM4D-mCherry in Figure 2b.
2. Fig. 1d only showed 3 hours (ZT12-14) sleep-wake behavior of mice after chemogenetic activation of DRN GAD2 neurons. The effective time of CNO is usually longer than three hours. To better understand the time-effect relationship of chemogenetic activation of DRN GAD2 neurons on sleep-wake behavior, please provide a 6-hour or 12-hour time-course curve graph. Additionally, please mark the zeitgeber time (ZT) in y-axis of Fig. 1d for easier understanding. Similar to Figure 1d, please add a time-course curve graph of chemogenetic inhibition of DRN GAD2 neurons on sleep-wake behavior in Figure 2.
3. In the experiment of optogenetic activation of DRN neurons (Fig. 1j and 1k), the control (no light) of optogenetic experiment is not enough. To exclude the influence of blue light on experimental result, authors should provide the results of mCherry-control group or WT-control group.

4. Moreover, could the authors provide a schematic heatmap that shows the result of each trail in Fig. 1j? In addition, have the authors tested the effect of long-term (for example, one or three hours) photogenetic activation of DRN GAD2 neurons on sleep behavior ?
5. Figure 2e showed that chemogenetics inhibition of DRN GAD2 neurons during light periods significantly increased wakefulness. But Figure 2i showed that genetic ablation of DRN GAD2 neurons increased wakefulness only in dark period, but not in light period. Please discuss the possible causes of the discrepancy between chemogenetics inhibition and genetic ablation results in the discussion section. In addition, have the authors tested the effect of chemogenetically inhibiting DRN GAD2 neurons in dark period (ZT12:00)?
6. The authors tried to illustrate that DRN GAD2 neurons constrain wakefulness in both physiological and allostatic challenging conditions. However, the results of figure 4 could not ideally support the conclusion“DRN GAD2 neurons constrain wakefulness in acute stress condition”, because the possibility exists that the reduced wakefulness is simply due to the physiological regulation by DRN GAD2 neurons, but not their regulation in allostatic challenging conditions. In order to exclude this possibility, authors should provide relevant evidences or provide a full explanation in discussion section.
7. In Figure 5d to 5i, in order to rule out the influence of yellow light and blue light on pupil size, authors should provide the results of mCherry-control group or WT-control group.
8. The base value of pupil size in light off condition varies largely in Figure 5f and Figure 5i (1.2 vs. 1.5), authors need to control experimental conditions more strictly (such as illuminance of the experimental environment) or provide relevant explanations.
9. Similar to Figure 1d, please add a 6-hour or 12-hour time-course curve graph of chemogenetic activation of DRN-PVT pathway on sleep-wake behavior in Figure 6.
10. In order to illustrate the control of sleep-wakefulness behavior by DRN-PVT pathway under allostatic challenging, authors had better add experiments to showed the effect of chemogenetic activation of DRN-PVT pathway in acute stress condition like figure 4g-4i.

Minor comments:

1. To accurately characterize the transduction selectivity of the virus, the authors had better provide a quantification of the percent of GAD2 neurons in all neurons transduced by AAV-EF1 α -DIO-hM3Dq-mCherry or AAV-EF1 α -DIO-hM4Di-mCherry.
2. Have the authors tested the effect of chemogenetically activating DRN GAD2 neurons in ZT 00:00 (light period)?
3. To give the reader a more direct understanding of the sleep-promoting effect of optogenetic activation of DRN GAD2 neurons, the authors should provide typical video files of experiments of Figure 1j.
4. Please provide more details on the method of fiber photometry data analysis. Specifically, how is the value of $\Delta F/F$ for each sample determined in Figure 3e and 3f ?
5. Line 246“DTA-induced genetic ablation of DRN GAD2 neurons or mCherry controls to acute restraint for 1h.”It is not consistent with the labeling in Figure 4d.

6. Line 863, "Data were analyzed by two tailed unpaired t test (c, e, g) and two tailed paired test (h, i). Data are presented as mean \pm s.e.m." Please deleting "g".
7. Please check the labeling of "GABAergic neurons" in figure 5a.
8. In Figure 5d, the authors had better employ whole-cell patch-clamp recordings to verify the optogenetic inhibition effect of AAV-ef1 α -DIO-GtACR-GFP.
9. Line 878 "h, i, Optogenetic activation of DRNGAD2 neurons attenuates tail shock induced increase of pupil size. n=6 mice, two tailed paired t test, **P=0.00708). Data were analyzed by two tailed unpaired t test. Data are presented as mean \pm s.e.m." Please check the description of statistical methods.
10. Please add a schematic heatmap to show pupil changes of each trail in Figure 6l, and add a mCherry-control group or WT-control group to rule out the influence of blue light on pupil changes.
11. Please indicate what the abbreviation "W, N, and NR" stands for in the legends of Figures 1,2,3 and 6, for the easier understanding of readers.
12. Line 934 "c, Distance traveled in the open filed over a 10-min period. n=6 for mCherry mice and n=7 for DTA mice". Please check the number of animal samples.
13. It is good that the authors showed the effect of activation and ablation of DRN GAD2 neurons on locomotion. Have the authors tested the effect of chemogenetic inhibition of DRN GAD2 neurons on locomotion?
14. Line 444 "the pipette was left in situ for 5min to allow diffusion and slowly withdrawn. For photometry recording, AAV-EF1 α -DIO-GCaMp7b (BrainVTA Technology, China) was injected into the DRN". Please unify the virus name in Method and figure 3a.
15. Line 517 "The laser intensity at the tip of optic fibers was adjusted to 10 mW". Generally, 3-5 mW is enough for photoactivating neurons. Why authors chosed 10 mW? It needs to be explained briefly.

Suggestion:

Authors can attempt to co-label Vgat with GAD1 (or GAD2) in DRN with fluorescence in situ hybridization. The possibility may exist that GAD1 neurons in the DRN promote wakefulness (these neurons may corelease GABA and glutamate), while GAD2 neurons promote sleep and only release GABA. It is quite possible that, in previous studies, the activation of DRN Vgat neurons induced the simultaneous activation of GAD1 neurons and GAD2 neurons. Since the wake-promoting effects of GAD1 neurons may cover the sleep-promoting effects of GAD2 neurons, the activation of Vgat neurons causes wakefulness, which is the opposite effect of activating GAD2 neurons alone in current study. The authors can discuss this possibility in the discussion section. It can well explain the discrepancy between current and previous results, and may further illustrate the specific mechanisms by which DRN GABA neurons regulate sleep-wake behavior.

Reviewer #2 (Remarks to the Author):

In this manuscript, Ren et al. employed viral techniques, fiber photometry and pupil recordings, and sleep analysis to investigate the functions of DR GAD2 neurons in arousal regulation. They find that while

the activation of the neurons promotes NREM sleep, the neurons are wake-active and that their manipulation can modulate pupil size during stress. While some findings are interesting, several major concerns diminish the impact of the study:

1. The complex role of DR neurons in regulating wakefulness is well-established, and the DR is highly heterogeneous. DR neurons co-express genes traditionally used as markers for different neuronal cell types, such as GAD2 (Huang et al. 2019; Fu et al., 2010; Okaty et al., 2015). Thus, using conventional genetic driver lines like GAD2, as the authors did, will result in labeling a mixed population that may be functionally irrelevant. Indeed, the discrepancy between the effects of chemogenetic/optogenetic manipulations, ablation and the physiological activity patterns of the neurons could be due to the modulation of a heterogeneous population. DR GABAergic neurons (Vgat-expressing) have already been shown to be wake-active and stress-activated, so studying a "non-functionally significant" population does not significantly contribute to the current understanding of dorsal raphe functions in wake regulation.
2. The authors assert that the function of these neurons is to constrain arousal but do not provide adequate justification for this claim. Activating a functionally insignificant subset of neurons may cause opposing effects on arousal. Reducing pupil size is not convincing evidence that the neurons' function is to constrain arousal. The magnitude of response to the manipulation of the neurons is similar in the stress condition (Fig. 4) to the undisturbed conditions (Fig 1 and 2), and there is no evidence in the activity pattern of the neurons to support their claim.
3. The term "allostatic challenges" is misused. Allostasis refers to chronic stress, not the response to an acute stressor, as used in this study.
4. Moreover, the paper lacks proper anatomical characterization of the targeted regions, making it impossible to identify the region targeted or recorded. It is evident from the limited representative images that areas outside the DR were substantially transduced. Similarly, the ablated region has not been characterized. A single 'representative' image is not sufficient to establish colocalization.
5. Lack of physiological relevance of manipulations: The authors used 5 mg/kg CNO in some experiments, but this dose has been demonstrated to affect sleep in mice and should not be used; the optogenetic stimulation of neurons continuously for 2 minutes at 20 Hz is not physiologically relevant.
6. The statistical approach is not rigorous. For example - Extended Data Fig.5d: Fisher LSD post hoc test does not correct for multiple comparisons and should not be employed. In Fig 3n, the authors should clarify whether all instances of pupil dilation during the recording were collected.

Minor comments

7. This sentence is misleading and incorrect: "DRN vesicular GABA transporter (Vgat)-positive neurons (DRNVgat) and glutamic acid decarboxylase 2 (GAD2)-positive neurons (DRNGAD2) may represent two functional subgroups of GABAergic neurons, regarding their biased modulation of locomotor activity^{33,38,39}." Both ref 38 and 39 demonstrated that DRN-GABA neurons are wake-active and wake-promoting. Ref 33 studied the function of the neurons in the context of feeding and is unrelated.

8. Correlation between delta and population activity during wake: an r-value of 0.07 is very low (and thus provides very low explanatory power), and the fact that there is a correlation with theta doesn't mean it is associated with electrocortical arousal. It can be a behavioral-related modulation.

9. At what circadian time was the acute stress done? Did you handle the control mice at the same time – to exclude the possible effects of being awake?

Reviewer #3 (Remarks to the Author):

Ren and Zhang et al. employed a multidisciplinary approach, utilizing various techniques such as opto-/chemogenetics, fiber photometry, polysomnographic recordings, pupil size recording, and circuit tracing, to investigate a novel circuit involved in regulating wakefulness and sleep. Their study specifically focused on the GAD2-positive dorsal raphe nucleus (DRN), which was found to modulate arousal levels in response to allostatic 'stressful' events that can potentially limit hypersomnia. The inclusion of pupil analysis in this study to assess arousal levels within the context of allostasis is commendable. By identifying a new pathway that connects physiological conditions with vigilance states, this research not only broadens our understanding of sleep but also has the potential to impact other fields, as the circuit they discovered may contribute to other emotion-related behaviors.

The manuscript is well-written and provides a thorough description of the methodology employed. The results presented adequately support the authors' claims. However, to further strengthen the study, additional experiments are required.

Firstly, it is essential to provide a statistical quantification of the percentage of GAD2-positive cells that were transduced with DREADD. This information will contribute to a better understanding of the efficiency and specificity of DREADD-mediated transduction.

Similarly, to confirm the specificity and effectiveness of ablation using diphtheria toxin subunit A (DTA), it is necessary to statistically quantify the number of Gad2-positive cells that were deleted in the DRN and adjacent areas. This quantitative analysis will support the reliability of the ablation method.

Additionally, I note discrepancies in the activity patterns of GABA cells reported in various studies. It would be worthwhile to investigate whether there are any PV+ (parvalbumin-positive) neurons within the Gad2 cell population or if they are distinct. Previous studies have suggested that PV+ cells exhibit wake-active properties, while Gad2 or other subtypes are primarily active during sleep. Dissecting PV+ cells and examining their population activity during different brain states would provide valuable information on the spontaneous activity of subsets of GABAergic neurons and their role in sleep-wake regulation across different brain regions.

In terms of minor comments, it is recommended to provide more precise labeling for AAV-ef1 α -DIO-hM3D-mCherry as hM3Dq, as well as for other hM3D and hM4D constructs mentioned in the manuscript (line 117). Furthermore, the description of the stress protocol appears inconsistent. While it is mentioned that the animals were restrained for 10 minutes, line 246 states that mice were subjected to acute restraint for 1 hour. Clarifying this inconsistency is necessary for accurate interpretation.

REVIEWER COMMENTS

Reviewer #1 (Remarks to the Author):

Summary:

Combining chemogenetic and optogenetic manipulation, fiber photometry recording, anatomical and functional circuit mapping, and behavioral paradigms, Ren et al. investigates whether and how DRN GAD2 neurons constrained wakefulness in physiological and allostatic challenging conditions. Overall, it is a highly innovative study. Previous studies showed that DRN Vgat-positive GABAergic neurons strongly promote arousal. Quite different from previous studies, Ren et al's study demonstrated that another group of DRN GABAergic neurons, GAD2-positive GABAergic neurons, promote sleep. More interestingly, these sleep-promoting neurons were highly-active during wakefulness in mice, which is quite different from most currently-known sleep-promoting neurons showing high-activity during sleep. Authors' further results showed that stressful stimuli significantly increased the activity of sleep-promoting DRN GAD2 neurons, and activation of DRN GAD2 neurons promotes sleep in acute stress condition. The authors proposed that DRN GAD2 neurons regulate sleep by constraining wake-promoting paraventricular thalamus in physiological and allostatic challenging conditions. These findings not only further elucidate the regulatory effects of DRN GABAergic neurons on sleep-wake behavior, but also innovates the sleep-regulating theory, which is of great interest to the sleep field. However, there are some problems in the experimental design (eg. lack of control experiments) which weaken the soundness of conclusions in this study. The comments below aim to help to improve the soundness of the conclusions and make the conclusions more convincing.

Major comments:

1. Considering that the GAD2 neurons in the PAG have been shown to promote sleep (Weber, F. et al. 2018. doi:10.1038/s41467-017-02765-w), and line 121 mentions that there are few ectopic expressions in adjacent PAG in Figure 1b, the authors should provide more anatomical details for Figure 1 b (e.g. a profile of more brain sections showing the detailed expression extent of AAV-ef1 α -DIO-hM3D-mCherry). Similar to Figure 1b, please provides detailed expression extent of AAV-ef1 α -DIO-hM4D-mCherry in Figure 2b.

Re: We thank the reviewer's suggestion. We have carefully re-examined the infected area of AAV-ef1 α -DIO-hM3Dq-mCherry and AAV-ef1 α -DIO-hM4Di-mCherry in the DRN and adjacent brain regions. For each mouse, we determined the spread of hM3Dq- and hM4Di-mCherry in 4 consecutive brain sections from -4.36~4.72 mm along the anterior to posterior axis, where most of viral expression was observed. In addition, we drawn the detailed expression pattern of hM3Dq- and hM4Di-mCherry in mice included for behavioral analysis. Results showed that both hM3Dq- and hM4Di-mCherry were exclusively expressed in the DRN, while very few viral construct expression was found in adjacent brain regions, such as ventral lateral PAG (vIPAG) and caudal linear nucleus of the raphe (CLi) (**Fig. R1**, n=17 for DRN^{GAD2}-hM3Dq and n=15 DRN^{GAD2}-hM4Di mice). In reviewer mentioned reference (Weber, F et al., *Nat Commun.*, 2018), Weber, F. et al found that vIPAG GAD2 neurons suppressed REM sleep and consolidated NREM sleep while genetic ablation of these neurons had little effects on wakefulness, which were in contrast to

present findings. Thus, these results suggested that viral construct transfected DRN^{GAD2} neurons mainly contributed to the behavioral effects of chemogenetic manipulation. We have added these results in **Supplementary Fig.1a** and **Supplementary Fig.4a** and described in the text, please see in line 119-122.

Fig. R1 | The infected area of hM3Dq-mCherry and hM4Di-mCherry in the DRN. a, b, Left, images showing coronal sections of DRN from GAD2-Cre mice injected with AAV-ef1 α -DIO-hM3Dq-mCherry (a) or AAV-ef1 α -DIO-hM4Di-mCherry (b). Right, drawings of superimposed hM3Dq-mCherry (a) or hM4Di-mCherry (b) expression in the DRN (n=17 for DRN^{GAD2}-hM3Dq mice and n=15 for DRN^{GAD2}-hM4Di mice).

2. Fig. 1d only showed 3 hours (ZT12-14) sleep-wake behavior of mice after chemogenetic activation of DRN GAD2 neurons. The effective time of CNO is usually longer than three hours. To better understand the time-effect relationship of chemogenetic activation of DRN GAD2 neurons on sleep-wake behavior, please provide a 6-hour or 12-hour time-course curve graph. Additionally, please mark the zeitgeber time (ZT) in y-axis of Fig. 1d for easier understanding. Similar to Figure 1d, please add a time-course curve graph of chemogenetic inhibition of DRN GAD2 neurons on sleep-wake behavior in Figure 2.

Re: Just as reviewer suggested, we have analyzed the effects of chemogenetic activation (related to original Fig.1d, e) or inhibition (related to original Fig.2d, e) of DRN^{GAD2} neurons on wakefulness/sleep in a 6h time course. For both chemogenetic activation and chemogenetic inhibition experiments, the statistically significant effects on wakefulness and NREM sleep were mostly observed within the first 3 hours following CNO injection (**Fig. R2**, n=11 for DRN^{GAD2}-

hM3Dq mice and n=8 for DRN^{GAD2}-hM4Di mice, two way repeated measure ANOVA following Bonferroni's multiple comparisons test). These results have been provided in **Supplementary Fig.1b to d** and **Supplementary Fig.4f to h**, respectively.

In addition, axis of ZT time have been added in **Fig. 1e** and **Fig. 2e** in the revised manuscript.

Fig. R2 | Time course of wakefulness/sleep states following chemogenetic manipulation of DRN^{GAD2} neurons. **a to c**, 6h time course curves showing the hourly time spent in wakefulness (**a**), NREM sleep (**b**), and REM sleep (**c**) of DRN^{GAD2}-hM3Dq mice following saline or CNO injection at the beginning of dark phase. Arrow heads indicate the time point of saline or CNO injection. n=11 mice, wake: $P_{ZT13}<0.001$, $P_{ZT14}<0.001$, $P_{ZT15}<0.001$; NREM: $P_{ZT13}<0.001$, $P_{ZT14}<0.001$, $P_{ZT15}<0.001$; REM: $P_{ZT18}=0.042$. **d to f**, 6h time course curves showing the time spent in wakefulness (**d**), NREM sleep (**e**), and REM sleep (**f**) of DRN^{GAD2}-hM4Di mice following saline or CNO (5mg/kg) injection at the beginning of light phase. n=8 mice, wake: $P_{ZT1}<0.001$, $P_{ZT2}=0.006$, $P_{ZT15}<0.001$; NREM: $P_{ZT1}<0.001$, $P_{ZT2}=0.006$, $P_{ZT3}=0.049$; REM: $P_{ZT18}=0.042$. * $P<0.05$, ** $P<0.01$, *** $P<0.001$. Data were analyzed by two-way repeated measure ANOVA following Bonferroni's multiple comparisons test and presented as mean \pm s.e.m.

3. In the experiment of optogenetic activation of DRN neurons (Fig. 1j and 1k), the control (no light) of optogenetic experiment is not enough. To exclude the influence of blue light on experimental result, authors should provide the results of mCherry-control group or WT-control group.

Re: We agree with the reviewer's suggestion and apologize for not providing data from control mice. We have performed optogenetic experiment on GAD2-Cre mice injected with AAV-ef1 α -DIO-mCherry into the DRN during the dark phase. We found that 473nm laser stimulation had no obvious effects on wakefulness/sleep states of DRN^{GAD2}-mCherry mice during dark phase (**Fig.R3a to c**, n=6 mice, $P_{Wake}=0.516$, $P_{NREM}=0.656$, $P_{REM}=0.983$, two tailed paired *t* test), thus excluding the potential influence of blue light on behavioral states. We have added these results in the **Supplementary Fig.2h to j** and described in the text, please see in line 144-145.

Considering that previous studies using optogenetic method to investigate the potential sleep-promoting role of specific neuronal population were mostly performed during the light phase

(Zhong, P. et al., *Neuron*, 2019; Zhang, Z. et al., *Cell*, 2019), we thus also tested the effects of optogenetic activation of DRN^{GAD2} neurons during the light phase (ZT4-ZT9). We found that optogenetic activation of DRN^{GAD2} neurons during the light phase significantly increased NREM sleep and decreased wakefulness, without obvious effects on REM sleep (**Fig.R3d to f**, n=6 mice, $P_{\text{Wake}}=0.00499$, $P_{\text{NREM}}=0.00335$, $P_{\text{REM}}=0.214$, two tailed paired *t* test). However, 473nm laser stimulation itself had no obvious effect on wakefulness/sleep states of DRN^{GAD2}-mCherry mice during the light phase. (**Fig.R3g to i**, n=6 mice, $P_{\text{Wake}}=0.099$, $P_{\text{NREM}}=0.127$, $P_{\text{REM}}=0.57$, two tailed paired *t* test). Although the reviewer did not suggest to examine the effects of optogenetic activation of DRN^{GAD2} neurons during light phase, we think these results could provide additional knowledge about the role of DRN^{GAD2} neurons in wakefulness/sleep regulation and thus add these results in the revised manuscript. Please see in **Fig.1k to m**, **Supplementary Fig.2b to d**, and in line 138-144.

Fig. R3 | Effects of optogenetic activation of DRN^{GAD2} neurons on wakefulness/sleep. a, d, g, EEG power spectrogram, EMG traces, and hypnograms showing 20 Hz laser stimulation of DRN^{GAD2}-mCherry mice during light (a) or dark phase (g) and DRN^{GAD2}-ChR2 mice during light phase (d). b, e, h, Brain states of all trials (top) and percentage of wake, NREM, or REM sleep (bottom) around 20 Hz stimulation of DRN^{GAD2}-mCherry mice during light (b) or dark phase (h) and DRN^{GAD2}-ChR2 mice during light phase (e). c, f, i, Effects of laser stimulation on wakefulness/sleep of DRN^{GAD2}-mCherry mice (c, i) and DRN^{GAD2}-ChR2 mice (f). n.s., not significant. Data were analyzed by two tailed paired *t* test and presented as mean \pm s.e.m.

4. Moreover, could the authors provide a schematic heatmap that shows the result of each trail in Fig. 1j? In addition, have the authors tested the effect of long-term (for example, one or three hours) photogenetic activation of DRN GAD2 neurons on sleep behavior?

Re: In the revised manuscript, we have added a schematic heatmap including all optogenetic activation trials (n=67 trials from 7 mice) related to original Fig. 1j to show brain states transitions. This heatmap along with Fig. 1j together showed that optogenetic activation of DRN^{GAD2} neurons during the dark phase decreased wakefulness while increased NREM sleep. Please see in **Supplementary Fig.2f**.

As optogenetic method was used to investigate the effects of activating DRN^{GAD2} neurons on different wakefulness/sleep states transitions, we here activated DRN^{GAD2} neurons by 20Hz stimulation lasting 120s but did not test the effects of long-term activation of DRN^{GAD2} neurons. As reviewer suggested, we examined the effects of 3h semi-chronic activation of DRN^{GAD2} neurons on wakefulness and sleep. To avoid the potential heating effects of optical stimulation, we delivered 20Hz laser pulses lasting for 15s with 45s interval in every minute during ZT14-ZT17 (**Fig. R4a**), which was adjusted from previous published study (Yu, X. *et al.*, *Nat Neurosci.*, 2019). We analyzed the time spent in each state 1 day pre- (baseline) and during optogenetic stimulation. Compared with baseline conditions, we found that semi-chronic optogenetic activation of DRN^{GAD2} neurons had no obvious effects on wakefulness and sleep (**Fig. R4b**, $P_{\text{Wake}}=0.432$, $P_{\text{NREM}}=0.513$, $P_{\text{REM}}=0.278$, n=8 mice, two tailed paired *t* test). Considering that chemogenetic method was used to investigate the effects long-term activation of DRN^{GAD2} neurons on wakefulness/sleep, we thus did not add these results in the revised manuscript.

Fig. R4 | Effects of semi-chronic optogenetic activation of DRN^{GAD2} neurons on wakefulness/sleep. **a**, Schematic of semi-chronic optogenetic activation of DRN^{GAD2} neurons during ZT14-ZT17. **b**, Time spent in each state during 3h of optogenetic activation of DRN^{GAD2} neurons. n.s., not significant. Data were analyzed by two tailed paired *t* test and presented as mean \pm s.e.m.

5. Figure 2e showed that chemogenetics inhibition of DRN GAD2 neurons during light periods significantly increased wakefulness. But Figure 2i showed that genetic ablation of DRN GAD2 neurons increased wakefulness only in dark period, but not in light period. Please discuss the possible causes of the discrepancy between chemogenetics inhibition and genetic ablation results in the discussion section. In addition, have the authors tested the

effect of chemogenetically inhibiting DRN GAD2 neurons in dark period (ZT12:00)?

Re: We thank the reviewer for pointing out the discrepancy between chemogenetic inhibition and genetic ablation results during the light phase. We speculated that this discrepancy may come from methodological differences. As chemogenetic and genetic ablation method inhibit neuronal activity over different time scales, chemogenetic method was used in present study to reversibly inhibit the activity of DRN^{GAD2} neurons in hours, whereas genetic ablation method induced chronic lesion of DRN^{GAD2} neurons in weeks (Takata, Y. et al., *J Neurosci*, 2018). Considering that compensated mechanisms may occur during DTA-induced lesion period, the different results from these two methods on wakefulness during light phase suggested that other neuronal populations may compensate the effects of ablating DRN^{GAD2} neurons on wakefulness in this time period. However, as ablation of DRN^{GAD2} neurons particularly increased wakefulness during the dark phase, the genetic ablation results highlighted an essential role of DRN^{GAD2} neurons in controlling wakefulness, at least in the dark phase. In addition, as mice typically spend more time in wakefulness during dark phase, thus more tightly control over behavioral states to avoid hyper-wakefulness is needed in dark phase. The genetic ablation results further suggested a more prominent role of DRN^{GAD2} neurons in constraining wakefulness during dark phase. These effects were consistent with ablating other neuronal populations playing a similar role. For example, ablation of Vgat⁺ GABA neurons in the ventral tegmental area especially increased wakefulness in the dark phase (Yu, X. et al., *Nat Neurosci*, 2019). We have added these discussions in the revised manuscript, please see in line 414-418.

For the effects of chemogenetically inhibiting DRN^{GAD2} neurons in dark period, we did not test this in the present study. In the revised manuscript, AAV-ef1 α -DIO-hM4Di-mcherry was injected into the DRN of GAD2-Cre mice and saline or CNO (2mg/kg) was injected at the beginning of dark phase (ZT12) after 4 weeks of virus expression (**Fig. R5a**). Compared with saline injections, we found that CNO injections led to increased wakefulness and decreased NREM sleep and REM sleep (**Fig. R5b to e**, n=5 mice, $P_{\text{Wake}}=0.0197$, $P_{\text{NREM}}=0.0244$, $P_{\text{REM}}=0.00182$, two tailed paired *t* test). These results suggested that chemogenetic inhibition of DRN^{GAD2} neurons during dark phase also increased wakefulness. The related results have been presented in **Supplementary Fig.4j to n** and described in the revised manuscript, please see inline 168-170.

Fig. R5 | Chemogenetic inhibition DRN^{GAD2} neurons during the dark phase increase wakefulness. **a**, EEG power spectrogram, EMG trances, and hypnograms from a DRN^{GAD2}-hM4Diry mouse during 3h post saline or CNO (2mg/kg) injection at the beginning of dark phase. **b to d**, Time course of wakefulness (**b**), NREM sleep (**c**), and REM sleep(**d**) following injection of saline or CNO at the beginning of dark phase. Arrow heads indicate the time point of saline or CNO injection. **e**, Time spent in each state during the first 3h after saline or CNO injection. * $P < 0.05$, ** $P < 0.01$, n.s., not significant. Data were analyzed by two tailed paired t test and presented as mean \pm s.e.m.

6. The authors tried to illustrate that DRN GAD2 neurons constrain wakefulness in both physiological and allostatic challenging conditions. However, the results of figure 4 could not ideally support the conclusion “DRN GAD2 neurons constrain wakefulness in acute stress condition”, because the possibility exists that the reduced wakefulness is simply due to the physiological regulation by DRN GAD2 neurons, but not their regulation in allostatic challenging conditions. In order to exclude this possibility, authors should provide relevant evidences or provide a full explanation in discussion section

Re: We appreciate the reviewer’s insightful comments. We agree with the reviewer’s concerns that the present conclusion “DRN^{GAD2} neurons constrain wakefulness in stress conditions” cannot exclude a physiological role of DRN^{GAD2} neurons in decreasing wakefulness. To strengthen the role of DRN^{GAD2} neurons in constraining wakefulness in acute stress condition, we performed the following experiments.

We first calculated the proportion of activated DRN^{GAD2} neurons in acute stress conditions by c-Fos staining. DRN^{GAD2} neurons were labeled with mCherry by injecting with AAV-ef1a-DIO-mCherry into the DRN of GAD2-Cre mice. DRN^{GAD2}-mCherry mice were restraint for 10 min or left in home cage for control. We found that acute restraint stress significantly increased c-Fos expression in mCherry-labeled DRN^{GAD2} neurons compared with control conditions (control: $7.86 \pm 1.16\%$; restraint: $67.02 \pm 5.22\%$; $n=5$ mice for each group, $P < 0.001$, two tailed unpaired t test). Considering that about two thirds ($67.02 \pm 5.22\%$) of DRN^{GAD2} neurons were activated in acute restraint stress conditions, chemogenetic activation of DRN^{GAD2} neurons could proportionally mimic the behavioral effects of these neurons in acute stress conditions. Please see **Fig.4a to c** in the revised manuscript.

To specifically investigate the behavioral effects of acute restraint stress-activated DRN^{GAD2} neurons, we then used c-Fos-based activity tagging method, also known as TetTagging. According to previous published method (Yu, X. *et al.*, *Science*, 2022), Cre-recombinase-dependent tagging vectors (AAV-cFos-tTA and AAV-TRE-DIO-hM3Dq-mCherry) were injected into the DRN of GAD2-Cre mice. To drive a c-Fos-dependent hM3Dq-mCherry expression in DRN^{GAD2} neurons, mice were experienced a session of 10 min restraint stress when doxycycline (Dox) was removed from the diet. Post experiments histological analysis showed that 74.21% TRE-hM3Dq-mCherry neurons was c-Fos positive. In undisturbed physiological conditions, CNO (1mg/kg) injection at the beginning of dark phase significantly decreased wakefulness while increased NREM sleep when compared with saline injection ($n=5$ mice, $P_{\text{wake}} < 0.001$, $P_{\text{NREM}} < 0.001$, $P_{\text{REM}} = 0.108$, two tailed paired t test). More importantly, in acute stress conditions, we found that activation of c-Fos-tagged DRN^{GAD2} neurons significantly shortened the NREM sleep latency following restraint stress ($n=9$ mice, $P = 0.004$, two tailed paired t test). In addition, activation of c-Fos-tagged

DRN^{GAD2} neurons decreased wakefulness while increased NREM sleep in acute stress conditions (n=9 mice, $P_{\text{wake}} < 0.001$, $P_{\text{NREM}} < 0.001$, two tailed paired *t* test). Please refer to **Fig.4g to k** and **Supplementary Fig.7** in the revised manuscript.

Taken together, these results suggested that DRN^{GAD2} neurons could constrain wakefulness in stressful challenging conditions. These results have been added in **Fig.4** and **Supplementary Fig.7** and described in the main text, please see in line 249-252, line 265-278.

7. In Figure 5d to 5i, in order to rule out the influence of yellow light and blue light on pupil size, authors should provide the results of mCherry-control group or WT-control group.

Re: We thank the reviewer's suggestion. To rule out the potential effects of laser stimulation on pupil size, we performed control experiments in DRN^{GAD2}-mCherry mice in the revised manuscript. As blue light was used to activate the ChR2-mCherry- or GtACR-GFP-expressing DRN^{GAD2} neurons, we delivered 10s blue light when mice receiving 0.2mA tail shock and recorded the pupil size. We found that blue light had no obvious effects on tail shock induced enlarge of pupil size (**Fig. R6**, n=6 mice, $P=0.991$, two tailed paired *t* test). These results have been added into the **Supplementary Fig.10a, b** and described in the main text, please see line 301-303.

Fig. R6 | Effects of laser stimulation on pupil size in DRN^{GAD2}-mCherry mice. a, Heatmaps showing tail shock induced-changes of pupil size in DRN^{GAD2}-mCherry mice. **b,** Laser stimulation has no obvious on tail shock-induced changes of pupil size. n.s., not significant. Data were analyzed by two tailed paired *t* test and presented as mean \pm s.e.m.

8. The base value of pupil size in light off condition varies largely in Figure 5f and Figure 5i (1.2 vs. 1.5), authors need to control experimental conditions more strictly (such as illuminance of the experimental environment) or provide relevant explanations.

Re: We apologize for not clearly describing the experimental conditions in the manuscript. For pupil recording experiments, the light illumination of recording environments was adjusted to ~60lux and well controlled during the entire recording session. The difference of baseline value of pupil size in Figure 5f and Figure 5i was due to the different amplitude of tail shocks were introduced. For optogenetic activation of DRN^{GAD2} neurons in tail shock experiment, we applied 0.2mA, 1s current. To avoid the potential ceiling effects, we applied 0.1mA 1s current for optogenetic inhibition experiments. We have added the amplitude of tail shock in **Fig5f, i** and described in the corresponding result and method parts in the revised manuscript, please see line300, 303 and line 674-676.

9. Similar to Figure 1d, please add a 6-hour or 12-hour time-course curve graph of chemogenetic activation of DRN-PVT pathway on sleep-wake behavior in Figure 6.

Re: According to reviewer's suggestions, we analyzed the effects of chemogenetic activation of DRN^{GAD2}-PVT pathway on wakefulness/sleep in a 6h time course. The related results have been presented in **Supplementary Fig.12e to g**.

10. In order to illustrate the control of sleep-wakefulness behavior by DRN-PVT pathway under allostatic challenging, authors had better add experiments to showed the effect of chemogenetic activation of DRN-PVT pathway in acute stress condition like figure 4g-4i.

Re: We appreciate the reviewer for this constructive suggestion. In revised manuscript, we investigated the effects of chemogenetic activation of DRN^{GAD2}-PVT pathway in acute stress conditions. We selectively labeled PVT-projecting DRN^{GAD2} neurons with hM3Dq-GFP. Mice were randomly injected with saline or CNO 30min before a 10min session of acute restraint, then EEG/EMG was recorded in the following 1h (**Fig. R7a**). Compared with saline injection, CNO injection significantly shortened the latency to NREM sleep after mice were released from the restraint tube (**Fig. R7b**, n=7 mice, $P=0.00569$, two tailed paired *t* test). Moreover, CNO injection significantly reduced time spent in wakefulness while increased NREM sleep time (**Fig. R7c**, n=7 mice, $P_{\text{Wake}}=0.00387$, $P_{\text{NREM}}=0.00349$, $P_{\text{REM}}=1$, two tailed paired *t* test). These results showed that chemogenetic activation of DRN^{GAD2}-PVT pathway decreased wakefulness in acute stress conditions. The related results have been added in **Fig. 7e to g** and described in the text, please see in line 357-360.

Fig. R7 | Chemogenetic activation of PVT-projecting DRN^{GAD2} neurons decreases wakefulness in acute stress conditions. **a**, Schematic of a 10 min restraint session and following EEG/EMG recording in DRN^{GAD2}-PVT-hM3Dq mice. **b**, NREM sleep onset latency following a 10min restraint session. **c**, Time spent in wake, NREM and REM sleep of DRN^{GAD2}-PVT-hM3Dq mice. ** $P<0.01$, n.s., not significant. Data were analyzed by two tailed paired *t* test and presented as mean \pm s.e.m.

Minor comments:

1. To accurately characterize the transduction selectivity of the virus, the authors had better provide a quantification of the percent of GAD2 neurons in all neurons transduced by AAV-EF1 α -DIO-hM3Dq-mCherry or AAV-EF1 α -DIO-hM4Di-mCherry.

Re: We have characterized the transduction selectivity of AAV-EF1 α -DIO-hM3Dq-mCherry or AAV-EF1 α -DIO-hM4Di-mCherry in DRN^{GAD2} neurons using FISH combined with IHC staining.

For hM3Dq-mCherry, ~96.6% mCherry-positive cells were stained with *GAD2* (n=4 mice). For hM4Di-mCherry, ~99.3% mCherry-positive cells were stained with *GAD2* (n=4 mice). These results suggested that both hM3Dq-mCherry and hM4Di-mCherry selectively transduced DRN^{GAD2} neurons. These results have been added in **Fig. 1d** and **Fig. 2d**.

2. Have the authors tested the effect of chemogenetically activating DRN GAD2 neurons in ZT 00:00 (light period)?

Re: The present study did not test the effects of chemogenetic activation of DRN^{GAD2} neurons during light phase. According to reviewer's suggestions, we tested this point in the revised manuscript. DRN^{GAD2}-hM3Dq mice were injected with saline or CNO (1mg/kg) at the beginning of light period (**Fig. R8a**). We found that chemogenetic activation of DRN^{GAD2} neurons during light phase significantly decreased wakefulness and increased NREM sleep (**Fig. R8b to e**, n=6 mice, $P_{wake}=0.00175$, $P_{wake}<0.001$, two tailed paired *t* test). Additionally, chemogenetic activation of DRN^{GAD2} neurons decreased REM sleep ($P_{REM}=0.004$). These results have been added into the **Supplementary Fig.1e to i**, please see in line 129-131.

Fig. R8 | Chemogenetic activation of DRN^{GAD2} neurons decreases wakefulness during light phase. **a**, Typical examples of EEG power spectrogram, EMG trances, and hypnograms from a DRN^{GAD2}-hM3Dq mouse during 3h post saline (left) or CNO injection at the beginning of light phase. **b to d**, Time course of wakefulness (**b**), NREM sleep (**c**), and REM sleep(**d**) following injection of saline or CNO at ZT0. **(e)** Time spent in each state during the first 3h after saline or CNO injection. * $P<0.05$, ** $P<0.01$. Data were analyzed by two tailed paired *t* test and presented as mean \pm s.e.m.

3. To give the reader a more direct understanding of the sleep-promoting effect of optogenetic activation of DRN GAD2 neurons, the authors should provide typical video files of experiments of Figure 1j.

Re: According to the reviewer's suggestions, we have provided a video file to show the effects of optogenetic activation of DRN^{GAD2} neurons on wakefulness/sleep. EEG, EMG, and behavioral signs showed that optogenetic activation of DRN^{GAD2} neurons induced transitions from wakefulness to NREM sleep. Please see in the **Supplementary Video 1**.

4. Please provide more details on the method of fiber photometry data analysis. Specifically, how is the value of $\Delta F/F$ for each sample determined in Figure 3e and 3f?

Re: We apologize for not providing enough details about calculating the value of $\Delta F/F$ in original Figure 3e and 3f. In the present study, we first aligned the EEG/EMG and fiber photometry data according to the simultaneous TTL markers. Then, we classified the wakefulness/sleep states based on EEG/EMG. The values of $\Delta F/F$ in specific brain states (wake, NREM, REM, long wake, short wake) were exported using the built-in function of Inper Data Process Software (Inper Technology, China). After contacting with the technician of Inper Technology, we knew that the $\Delta F/F$ was calculated as $F(t)-F_0(t)/F_0(t)$, where $F_0(t)$ was taken as the minimum value of $F(t)$ during a time window before a specific time point t (Jia, H. *et al.*, *Nat Protoc.*, 2011). Considering that the $F_0(t)$ was changing with time and was not stable, this method was not suitable for comparing the $\Delta F/F$ value in different states in a recording session. We realized that we unintentionally used a wrong method to determine the $\Delta F/F$ value in different wakefulness/sleep states. To correct this, we defined the F_0 as the average Ca^{2+} signal from a stable NREM sleep (>30s) of each recording session according to published studies (Cai, P. *et al.*, *Sleep*, 2022). F was the average value of fluorescence signal of a specific wakefulness/sleep state (wake, NREM, REM, long wake, short wake). $\Delta F/F$ was calculated as $(F-F_0)/F_0$. We reanalyzed the fiber photometry data in each state and the results have been presented in **Fig. 3f, g**. In addition, we have added the analysis details of fiber photometry data in the method part of revised manuscript, please refer to line 615-620.

5. Line 246“DTA-induced genetic ablation of DRN GAD2 neurons or mCherry controls to acute restraint for 1h.” It is not consistent with the labeling in Figure 4d.

Re: We are sorry for this mistake. Mice were restraint for 10min. We have corrected this in the revised manuscript, please see in line 255-257.

6. Line 863, “Data were analyzed by two tailed unpaired t test (c, e, g) and two tailed paired test (h, i). Data are presented as mean \pm s.e.m.” Please deleting “g”.

Re: Actually, data in pane “g” were analyzed by two tailed unpaired t test.

7. Please check the labeling of “GABAergic neurons” in figure 5a.

Re: We have revised this point, please see in **Fig.5a**.

8. In Figure 5d, the authors had better employ whole-cell patch-clamp recordings to verify the optogenetic inhibition effect of AAV-ef1 α -DIO-GtACR-GFP.

Re: We thank the reviewer’s suggestion. In the revised manuscript, we performed whole-cell patch-clamp recording to verify the functional expression of AAV-ef1 α -DIO-GtACR-GFP in DRN^{GAD2} neurons. FISH results showed that GtACR-GFP specifically expressed in the GAD2 neurons (**Fig. R9a** to **c**, GtACR+GAD2/GAD2=89.07 \pm 4.00%, GtACR+GAD2/GtACR=97.73 \pm 1.16%). In acute DRN brain slices, we recorded the GtACR-GFP-positive cells and delivered blue laser pulses lasting 30s in current clamp mode (**Fig. R9d** and **e**). We found that blue laser pulses abolished the action potential firing of GtACR-GFP-positive neurons, suggesting that activation of GtACR could efficiently inhibit DRN^{GAD2} neurons (**Fig. R9f**, n=6 cells from 3 mice, $P=0.00276$, two tailed paired t test). We have added these results to the

Supplementary Fig.9 and described in the main text, please see in line 297-298.

Fig. R9 | Patch-clamp recording verifies the functional expression of GtACR-GFP in DRN^{GAD2} neurons. **a, b**, Representative images showing the expression of GtACR-GFP in the DRN (**a**) and the colocalization of GtACR-GFP with *GAD2* mRNA (**b**). **c**, Quantification of colocalization of between GtACR-GFP and *GAD2*. **d**, Schematic diagram of experiment to examine the functional expression of GtACR-GFP using whole-cell patch-clamp recording. **e**, Example trace from a recorded GtACR-GFP-positive DRN^{GAD2} neuron. 30s constant blue laser stimulation abolished the spontaneous firing of action potentials of recorded neuron. **f**, Firing rate of action potentials pre and during 30s blue laser stimulation. Data were analyzed by two tailed paired *t* test and presented as mean ± s.e.m.

9. Line 878 “h, i, Optogenetic activation of DRNGAD2 neurons attenuates tail shock induced increase of pupil size. n=6 mice, two tailed paired *t* test, **P=0.00708). Data were analyzed by two tailed unpaired *t* test. Data are presented as mean ± s.e.m.” Please check the description of statistical methods.

Re: We are sorry for this mistake in Line 878. Actually, data in figure 5i were analyzed by two tailed paired *t* test. We have corrected this in the revised manuscript, please see in line 1061-1062.

10. Please add a schematic heatmap to show pupil changes of each trail in Figure 6l, and add a mCherry-control group or WT-control group to rule out the influence of blue light on pupil changes.

Re: According to reviewer’s suggestion, we have added a heatmap related to original Figure 6i to show the effects of optogenetic activation of DRN^{GAD2}-PVT pathway on tail shock induced pupil size changes. Please see in **Fig.7i** in the revised manuscript.

In addition, we tested the effects optogenetic activation of DRN^{GAD2}-PVT pathway on pupil size in DRN^{GAD2}-mCherry mice, just as reviewer suggested. *GAD2*-Cre mice was injected with AAV-EF1α-DIO-ChR2-mCherry and an optic fiber was implanted above the PVT (**Fig. R10a**). 20Hz laser stimulation was randomly delivered during tail shock (**Fig. R10b**). Compared with light off condition, we found that laser stimulation of the PVT in DRN^{GAD2}-mCherry mice had no obvious effects on tail shock induced increase of pupil size (**Fig. R10c**, n=6 mice, *P*=0.998, two tailed

paired t test). We have added these results in **Supplementary Fig.12k to m** and described in the main text, please see in line 367-369.

Fig. R10 | Effects of laser stimulation of DRN^{GAD2}-PVT pathway on pupil size in DRN^{GAD2}-mCherry mice. **a**, Schematic diagram of optogenetic activation of DRN^{GAD2}-PVT pathway in DRN^{GAD2}-mCherry mice. **b**, Heatmaps showing tail shock induced-changes of pupil size in. Pupil_{Norm.}, normalized pupil size. **c**, Laser stimulation has no obvious on tail shock-induced changes of pupil size in DRN^{GAD2}-mCherry mice. n.s., not significant. Data were analyzed by two tailed paired t test and presented as mean \pm s.e.m.

11. Please indicate what the abbreviation “W, N, and NR” stands for in the legends of Figures 1,2,3 and 6, for the easier understanding of readers.

Re: Abbreviation for “W, N, and NR” have been added in the corresponding figure legends in the revised manuscript.

12. Line 934 “c, Distance traveled in the open field over a 10-min period. n=6 for mCherry mice and n=7 for DTA mice”. Please check the number of animal samples.

Re: We have checked and correct this mistake in the revised manuscript, please see in line1238.

13. It is good that the authors showed the effect of activation and ablation of DRN GAD2 neurons on locomotion. Have the authors tested the effect of chemogenetic inhibition of DRN GAD2 neurons on locomotion?

Re: The present study did not test the effects of chemogenetic inhibition of DRN^{GAD2} neurons on locomotion. According to reviewer’s suggestions, we tested this point in the revised manuscript. DRN^{GAD2}-hM4Di mice were injected with saline or CNO (2mg/kg) 30min before open field test (**Fig. R11a**). Injection of saline or CNO was separated at least five to seven days. Compared saline injection, we found that the CNO-induced chemogenetic inhibition of DRN^{GAD2} neurons decreased the travel distance of mice in the open field (**Fig. R11b, c**, n=6 mice, $P<0.001$, two tailed paired t test), which were consistent with the effect of ablation of DRN^{GAD2} neurons on locomotion. These results have been added in **Supplementary Fig.3d to f**, and described in the text, please see in line 167-170.

Fig. R11 | Chemogenetic inhibition of DRN^{GAD2} neurons increases locomotor activity in the open field. **a**, Schematic diagram of experimental setup for chemogenetic inhibition of DRN^{GAD2} neurons in an open field. **b**, Representative video-tracked paths from a DRN^{GAD2}-hM4Di mouse follow saline or CNO injection. **c**, Distance traveled of DRN^{GAD2}-hM4Di mice in saline and CNO group. *** $P < 0.01$. Data were analyzed by two tailed paired t test and presented as mean \pm s.e.m.

14. Line 444 “the pipette was left in situ for 5min to allow diffusion and slowly withdrawn. For photometry recording, AAV-EF1 α -DIO-GCaMp7b (BrainVTA Technology, China) was injected into the DRN”. Please unify the virus name in Method and figure 3a.

Re: We have checked this point and unified the virus name in the revised manuscript, please see in page12 line 25-26. In addition, we have carefully checked manuscript to avoid similar mistakes.

15. Line 517 “The laser intensity at the tip of optic fibers was adjusted to 10 mW”. Generally, 3-5 mW is enough for photoactivating neurons. Why authors chosed 10 mW? It needs to be explained briefly.

Re: We thank the reviewer for raising this point. The laser intensity was chosen based on previous published studies, in which 10mW laser intensity was used to optogenetically activate DRN neurons (Cho, J. R. et al., *Neuron*, 2017; Gazea, M. et al., *J Neurosci.*, 2021). We thus used 10mW laser intensity to make sure fully activating DRN^{GAD2} neurons transfected with ChR2-mCherry. We have explained this in the method part of revised manuscript, please see line 587-589.

Suggestion:

Authors can attempt to co-label Vgat with GAD1 (or GAD2) in DRN with fluorescence in situ hybridization. The possibility may exist that GAD1 neurons in the DRN promote wakefulness (these neurons may corelease GABA and glutamate), while GAD2 neurons promote sleep and only release GABA. It is quite possible that, in previous studies, the activation of DRN Vgat neurons induced the simultaneous activation of GAD1 neurons and GAD2 neurons. Since the wake-promoting effects of GAD1 neurons may cover the sleep-promoting effects of GAD2 neurons, the activation of Vgat neurons causes wakefulness, which is the opposite effect of activating GAD2 neurons alone in current study. The authors can discuss this possibility in the discussion section. It can well explain the discrepancy between current and previous results, and may further illustrate the specific mechanisms by which DRN GABA neurons regulate sleep-wake behavior.

Re: We thank the reviewer for this constructive suggestion. Dissecting whether *Vgat*, *GAD1*, and *GAD2* target different subtypes of DRN GABAergic neurons will be helpful for explaining the discrepancy between current findings and published results regarding the role of DRN GABAergic neurons in wakefulness regulation (Gazea, M. et al., *J Neurosci.*, 2021; Cai, P. et al., *Sleep*, 2022). Indeed, *Vgat*- and *GAD2* (or *GAD1*)-GABAergic neurons with opposing role in regulating wakefulness in the ventral tegmental area (VTA) (Yu, X. et al., *Nat Neurosci.*, 2019; Chowdhury, S. et al., *Elife*, 2019) or physical activity in the lateral hypothalamus have been reported (Kosse, C. et al., *Proc Natl Acad Sci U S A.* 2017; Rossier, D. et al., *Proc Natl Acad Sci U S A.* 2021). As for the DRN, previous studies evaluated the co-labeling of GABAergic neurons with 5HT or TH neurons (Ren, J. et al., *Elife*, 2019; Cardozo Pinto, D. F. et al., *Nat Commun.*, 2019). However, the expression patterns of different subtypes of GABAergic neurons in the DRN have not been examined.

According to reviewer's suggestion, we have performed FISH experiments to examine the coexistence of different markers for GABAergic neurons in the DRN, including *Vgat* (*Slc32a1*), *GAD1*, and *GAD2* (**Fig. R12a**). We collected at least 4-5 DRN slices each mouse along the anterior to posterior axis and repeated this in 5 mice. Neurons expressing either of the above three markers were considered as GABAergic neurons and a total of 9228 DRN GABAergic neurons were counted from 5 mice (1845.6 ± 243.11 on average). We first calculated the percentage of a specific type of GABAergic neurons (with or without presence of the other two markers) among all DRN GABAergic neurons. *Vgat*-, *GAD1*-, or *GAD2*-positive neurons accounted for $84.84 \pm 2.34\%$, $74.49 \pm 4.18\%$, $95.58 \pm 1.32\%$ of all DRN GABAergic neurons, respectively (**Fig. R12b**). In contrast to our expectations, *GAD2*-positive but not *Vgat*-positive neurons represented the largest proportion of DRN GABAergic neurons. We next calculated the co-labeling between these three GABAergic markers (**Fig. R12c**). Analysis results showed that $70.50 \pm 3.80\%$ DRN GABAergic neurons expressed all these three markers. For co-labeling between *Vgat* and *GAD2*, almost all (99.52%) *Vgat*-positive neurons expressed *GAD2* whereas 79.85% of *GAD2*-positive neurons expressed *Vgat*. Notably, there was a small population (10.71% of all DRN GABAergic neurons) of *GAD2*-positive neurons without *Vgat* expression. For co-labeling between *Vgat* and *GAD1*, 97.26% of *GAD1*-positive neurons expressed *Vgat*. Thus, the present FISH results did not support the speculation that *GAD1* or *GAD2* neurons represents parts of DRN *Vgat* neurons and activation of DRN *Vgat* neurons may induce the simultaneous activation of *GAD1* and *GAD2* neurons, just like *Vgat* and GAD neurons in the VTA (Chowdhury, S. et al., *Elife*, 2019). Actually, these results revealed an underestimated complexity of DRN GABAergic neurons. These results further suggested that targeting DRN GABAergic neurons using *Vgat*-Cre or *GAD2*-Cre mice may result in labelling different populations of GABAergic neurons.

For the reviewer mentioned potential co-released glutamate from DRN GABAergic neurons, we performed whole-cell patch-clamp recording of PVT neurons and optogenetically activated nerve terminals of DRN^{GAD2} neurons in the PVT. We found that activation of DRN^{GAD2} neurons terminals only induced GABA release in postsynaptic PVT neurons (Please refer to **Fig.6h** to **m** in the revised manuscript).

Taken together, the above results provided important clues for explaining the functional difference between DRN *Vgat* and *GAD2* neurons. However, it is still unknown whether the anatomical difference of *Vgat*- or *GAD2*-positive neurons contribute to their different functional roles in regulating wakefulness. In addition, the different output effector targets of these two GABAergic

populations should also be taken into consideration. DRN Vgat neurons promoted wakefulness through direct inhibition of NREM-sleep promoting VTA Vgat neurons or indirectly disinhibition of wake-promoting neurons via LH GABA neurons. However, we found that DRN GAD2 neurons constrained wakefulness through direct inhibition of wakefulness-promoting PVT neurons. Considering that VTA- or LH-projecting DRN Vgat neurons are proved to be non-overlapping, thus future studies are needed to investigate the role of molecular defined- or projection specific-subtypes of DRN GABAergic neurons. We have added these results in **Supplementary Fig.15** and described in the main text. In addition, we discussed the different functional role between DRN^{Vgat} and DRN^{GAD2} neurons regarding their anatomical difference and the different output effector targets, please see in line 464-480.

Fig. R12 | The distribution of *Slc32a1*(Vgat), *GAD1*, and *GAD2* neurons in the DRN. a, Representative images showing the expression of *Slc32a1*-, *GAD1*-, and *GAD2*-positive neurons in the DRN. The zoomed area a1 shows that most DRN GABAergic neurons express all three

markers, whereas the zoomed area a2 shows that DRN GABAergic neurons expressing one or two of GABAergic markers. The white arrowheads indicate *Slc32a1*, *GAD1*, and *GAD2* triple-positive neurons. The purple arrowheads indicate *GAD1*-positive neurons. The green arrowheads indicate *GAD2*-positive neurons. **b**, Percentage of *Slc32a1*-, *GAD1*-, and *GAD2*-positive neurons among all DRN GABAergic neurons. **c**, Percentage of DRN GABAergic neurons expressing either *Slc32a1*, *GAD1*, or *GAD2*.

Reviewer #2 (Remarks to the Author):

In this manuscript, Ren et al. employed viral techniques, fiber photometry and pupil recordings, and sleep analysis to investigate the functions of DR GAD2 neurons in arousal regulation. They find that while the activation of the neurons promotes NREM sleep, the neurons are wake-active and that their manipulation can modulate pupil size during stress.

1. The complex role of DR neurons in regulating wakefulness is well-established, and the DR is highly heterogeneous. DR neurons co-express genes traditionally used as markers for different neuronal cell types, such as GAD2 (Huang et al. 2019; Fu et al., 2010; Okaty et al., 2015). Thus, using conventional genetic driver lines like GAD2, as the authors did, will result in labeling a mixed population that may be functionally irrelevant. Indeed, the discrepancy between the effects of chemogenetic/optogenetic manipulations, ablation and the physiological activity patterns of the neurons could be due to the modulation of a heterogeneous population. DR GABAergic neurons (Vgat-expressing) have already been shown to be wake-active and stress-activated, so studying a "non-functionally significant" population does not significantly contribute to the current understanding of dorsal raphe functions in wake regulation.

Re: We appreciate the reviewer's insightful comments. The reviewer's major concern is that GAD2 may label a mixed population of DRN neurons and thus may not significantly contribute to the understanding of DRN functions in wakefulness regulation, with respect to the reported role of DRN^{Vgat} neurons. As the reviewer mentioned, the DRN is highly heterogeneous. The role of different types of DRN neurons in wakefulness/sleep regulation have been investigated for decades. Though progress have been made in dissecting the roles of specific DRN cell types (especially 5HT and dopamine neurons) in wakefulness/sleep regulation (*Sulaman, B. A. et al., Nat Neurosci., 2022; Oikonomou, G. et al., Neuron, 2019; Cho, J. R. et al., Neuron, 2017*), the role of other DRN cell types (GABAergic neurons for example) is still elusive. Based on these considerations, we investigated the role of DRN GABAergic neurons by using GAD2-Cre driver lines. While our project was almost finished, two recent published studies (*Gazea, M. et al., J Neurosci., 2021; Cai, P. et al., Sleep, 2022*) reported a wakefulness-promoting role of DRN Vgat-positive GABAergic neurons. Since Vgat-Cre and GAD2-Cre mice have been reported to target different subtypes of GABAergic neurons in the lateral hypothalamus (*Kosse, C. et al., Proc Natl Acad Sci U S A. 2017; Rossier, D. et al., Proc Natl Acad Sci U S A. 2021*), our present study and the two published studies targeted may thus label different subtypes of DRN GABAergic neurons. Considering that different subtypes of GABAergic neurons in a same brain region have been reported to play different roles in wakefulness/sleep regulation (*Yu, X. et al., Nat Neurosci., 2019; Chowdhury, S. et al., Elife, 2019; Liu, D. et al., Science, 2020*), our study together with published studies could further significantly improve the understanding of DRN GABAergic neurons' functions in wakefulness regulation.

As for the anatomical identities of DRN GABAergic neurons, makers for GABAergic neurons have been shown to co-expressed in other cell types in the DRN, such as GAD1 (*Fu, W. et al., J Comp Neurol., 2010; Okaty, B. W. et al., Neuron, 2015*) and GAD2 in serotonin (5HT) neurons (*Huang, K. W. et al., Elife, 2019*), Vgat in glutamatergic neurons (*Huang, K. W. et al., Elife, 2019*). For GAD2, published studies using single-cell RNA sequencing method have reported that GAD2 expressed in a subset of DRN 5HT neurons (*Ren, J. et al., Elife, 2019; Huang, K. W. et al., Elife, 2019; Okaty, B. W. et al., Elife, 2020*). In addition, *Sert-Flp* with the *IS* reporter mice crossed with

GAD2-Cre mice labeled double GAD2 and 5HT positive neurons, which were mainly located in dorsal and lateral parts of the DRN (Ren, J. et al., *Elife*, 2019). However, labeling GAD2 neurons by injecting AAV-DIO-GFP into the DRN of GAD2-Cre mice showed that only 0.1% DRN^{GAD2} neurons co-expressed with Tph2 (5HT neurons marker), as well as 1.4% DRN^{GAD2} neurons co-expressed with TH (dopamine neurons marker) (Cardozo Pinto, D. F. et al., *Nat Commun.*, 2019). Thus, it seems like that results from different methods are controversial. To gain more insights about the specificity of DRN^{GAD2} neurons, we performed the following experiments.

1. We first examined the expression patterns of DRN^{GAD2} neurons in conjunction with 5HT and dopamine neurons, which were involved in wakefulness/sleep regulation (Oikonomou, G. et al., *Neuron*, 2019; Cho, J. R. et al., *Neuron*, 2017). DRN^{GAD2} neurons were identified by FISH method while 5HT and dopamine neurons were identified by immunostaining of Tph2 or TH, respectively. Results showed that GAD2⁺ neurons and 5HT neurons exhibited a complementary pattern in the DRN, with GAD2⁺ neurons mostly located in the lateral DRN while 5HT neurons distributed throughout the DRN. Through close examination of the stained slices, we found that there were a population of 5HT neurons in the dorsal part of DRN expressed GAD2 mRNA (**Fig. R13a**), which was consistent with previous studies (Ren, J. et al., *Elife*, 2019). Quantification results revealed that GAD2- and 5HT- double positive neurons account for 17.25% of total 5HT neuron and 8.99% of GAD2⁺ neurons (**Fig. R13b** and **c**). In addition, we analyzed the relative expression level of GAD2 mRNA in Tph2 neurons and non-Tph2 neurons. Notably, we found that GAD2 mRNA was much more enriched in non-Tph2 neurons compared with Tph2 neurons (**Fig. R13d**). In contrast, morphological results showed that only 2.4% TH neurons expressed GAD2 mRNA and only 0.3% GAD2⁺ neurons expressed TH (**Fig. R13d** to **f**). These results suggested that GAD2 was weakly expressed in a small proportion of DRN 5HT neurons, with barely expression in DRN dopamine neurons.

Fig. R13 | Characterization of the expression pattern of GAD2, Tph2, and TH neurons in the DRN. Representative images showing the expression of GAD2⁺ and Tph2⁺ neurons in the DRN. Dashed boxes in the top panels are enlarged at the bottom panels. White arrowheads mark examples of GAD2- and Tph2-double positive neurons. **b** and **c**, Pie charts showing the percentage of GAD2- and Tph2-double positive neurons (yellow) in all Tph2 neurons (**b**, green) or GAD2 neurons (**c**, red). n=4 mice. **d**, Quantification of immunofluorescence intensity of GAD2 mRNA in Tph2 or non-Tph2 neurons. n=4 mice, $P=0.00618$. **e**, Images showing the expression of GAD2 and TH neurons in the DRN. **f**, **g**, Pie charts showing the percentage of GAD2- and TH-double positive neurons (yellow) in all TH neurons (**f**, green) or GAD2 neurons (**g**, red). n=4 mice. ** $P<0.01$. Data were analyzed by two tailed unpaired t test (**d**) and presented as mean \pm s.e.m.

2. Considering that we injected viral vectors to label DRN^{GAD2} neurons by using GAD2-Cre mice, we then examined the expression of Tph2 or TH in GFP-labeled DRN^{GAD2} neurons. We injected AAV-ef1 α -DIO-GFP into the DRN of GAD2-Cre mice and performed immunostaining of Tph2

and TH in DRN slice. GFP-labeled GAD2 neurons were mainly located in the lateral wings of DRN (**Fig. R14a**). In contrast to FISH results, there were rarely GFP-labeled GAD2 neurons containing TPH2. The GFP- and Tph2- double positive neurons constituted only 0.43% of GFP neurons and 0.4% of 5HT neurons (**Fig. R14b** and **c**). Similarly, nearly no GFP-labeled GAD2 neurons expressed dopamine neurons marker TH (**Fig. R14d** and **f**). These findings were in concert with a published report showing that detectable levels of TPH or TH was only found in a small proportion of viral vectors labeled GAD2 neuron in GAD2-Cre mice (*Cardozo Pinto, D. F. et al., Nat Commun., 2019*). These data from GAD2-Cre mice suggested that viral vectors-labeled GAD2 neurons represent an independent cell population in the DRN. However, we still not known the why the above two strategies resulted in different co-labeling ratio between GAD2 and Tph2.

Fig. R14 | Characterization of the expression pattern of GFP-labeled GAD2 neurons, Tph2, and TH neurons in the DRN. **a**, Representative images showing the expression of GFP-labeled GAD2 neurons and Tph2 neurons in the DRN. GAD2 neurons was labeled with GFP by injecting AAV-ef1 α -DIO-GFP into the DRN of GAD2-Cre mice. Dashed boxes in the top panels are enlarged at the bottom panels. **b** and **c**, Pie charts showing the percentage of GFP- and Tph2-double positive neurons (yellow) in all Tph2 neurons (**b**, green) or GFP neurons (**c**, red). n=4 mice. **d**, Images showing the expression of GFP-labeled GAD2 neurons and TH neurons in the DRN. **e** and **f**, Pie charts showing the percentage of GFP- and TH-double positive neurons (yellow) in all

TH neurons (**e**, green) or GFP⁺ neurons (**f**, red). n=4 mice.

3. To further verify the specificity of viral vectors-labeled DRN^{GAD2} neurons in GAD2-Cre mice, we performed *in vitro* patch-clamp recording combined with optogenetics to examine the neurotransmitter release from DRN^{GAD2} neurons terminals in downstream targets. We injected AAV-ef1 α -DIO-ChR2-mCherry into the DRN of GAD2-Cre mice. 4 to 6 weeks later, we collected PVT brain slice and performed whole-cell patch-clamp recording from PVT neurons (**Fig. R15a** and **b**). Activation of nerve terminals of DRN^{GAD2} neurons in the PVT using single 488 light pulses (5ms duration) induced post-synaptic currents (PSC) in recorded neurons (10/14). The average synaptic delay from stimulation onset was 5.15 \pm 0.56ms and average amplitude was 155.6 \pm 58.83pA (**Fig. R15c** and **d**), indicating monosynaptic connections between DRN^{GAD2} neurons and PVT. Consistently, the PSCs were sensitive to Na⁺ channel blocker tetrodotoxin (TTX, 1 μ M) and were rescued by the K⁺ channel blocker 4-aminopyridine (4AP, 2mM). Furthermore, the PSCs were blocked by GABA_A receptor antagonist bicuculine (10 μ M) (**Fig. R15c** and **e**, n =10 cells, ACSF vs TTX: *P*=0.01; TTX+4AP vs TTX+4AP+ bicuculine: *P*<0.001; TTX vs TTX+4AP: *P*=0.9, Friedman repeated measures ANOVA on ranks), suggesting that the monosynaptic connections between DRN^{GAD2} neurons and PVT neuron were GABAergic. To investigate whether optogenetic activation of DRN^{GAD2} neuron terminals induces possible releases of slow reaction neuromodulators, we also used 10Hz or 20Hz prolonged (2s) light stimulation. We found that 10Hz or 20Hz stimulation induced long lasting PSCs in the presence of TTX and 4AP, which was again blocked by bicuculine (**Fig. R15f**, n=6 cells). These results suggested that DRN^{GAD2} neurons only mediated direct inhibitory effects on PVT neurons.

Fig. R15 | DRN^{GAD2} neurons directly inhibit PVT neurons. **a**, Schematic diagram of whole-cell recording in the PVT. AAV-DIO-EF1 α -ChR2-mCherry was injected into the DRN of GAD2-Cre mice. Recordings were made from PVT neurons with simultaneous optogenetic stimulation of terminals of DRN^{GAD2} neurons. **b**, Light-evoked action potentials of DRN^{GAD2} neurons verifies the expression efficacy of ChR2-mCherry. **c**, 5ms light pulses induced post-synaptic currents in the recorded PVT neuron and the TTX, 4AP, and bicuculine on light-evoked currents. The dark lines indicate average traces and gray lines indicate responses of individual cells (n=10 cells). **d**, Latency (left) and amplitude (right) of light-evoked postsynaptic currents in ACSF conditions. **e**, Summary of the amplitude of light-evoked currents normalized to that in ACSF. **f**, Bicuculine

blocks the 10Hz and 20Hz light pulses-evoked postsynaptic currents.

Taken together, the above results indicated that targeting DRN^{GAD2} neurons using GAD2-Cre mice labeled an independent but not a mixed population of neurons, which only made inhibitory connections with downstream targets (or at least the PVT). To explore the possible reasons responsible for the functional difference of DRN GAD2-neurons and Vgat-neurons, we also performed *Slc32a1(Vgat)* and *GAD2* FISH experiments. FISH results revealed that *Vgat*- and *GAD2*-neurons were largely overlapping but *GAD2*-neurons represent a larger population of DRN GABAergic neurons than *Vgat*-neurons (please refer to **Supplementary Fig.15** in the revised manuscript), highlighting an underestimated complexity of DRN GABAergic neurons. These results further prompted a necessity of dissecting the functional roles of different subtypes of DRN GABAergic neurons in control wakefulness/sleep and other behaviors. The related results have been added in **Fig.6h to m**, **Supplementary Fig.13**, and **Supplementary Fig.14** and discussed in the revised manuscript, please refer to page line 338-348, line 452-463.

2. The authors assert that the function of these neurons is to constrain arousal but do not provide adequate justification for this claim. Activating a functionally insignificant subset of neurons may cause opposing effects on arousal. Reducing pupil size is not convincing evidence that the neurons' function is to constrain arousal. The magnitude of response to the manipulation of the neurons is similar in the stress condition (Fig. 4) to the undisturbed conditions (Fig 1 and 2), and there is no evidence in the activity pattern of the neurons to support their claim.

Re: Thanks for the reviewer's insightful comments. In the present study, we used chemogenetic, optogenetic, and genetic ablation manipulations demonstrated the activation/inhibition of DRN^{GAD2} neurons decreased/increased wakefulness while increased/decreased NREM sleep, respectively. These results suggested a possible NREM sleep-promoting role of DRN^{GAD2} neurons. Surprisingly, we found that the activity of DRN^{GAD2} neurons were wakefulness-active and correlated with arousal level, indicating these neurons may not promote physiological NREM sleep. After carefully reviewing the published reference, we found there are indeed neuronal populations engaged in wakefulness/sleep regulation showing inconsistent activity pattern and physiological function (*Oikonomou, G. et al., Neuron, 2019; Yu, X. et al., Nat Neurosci., 2019*). Similar to DRN^{GAD2} neurons, GABAergic neurons the VTA displayed higher activity during wakefulness or active phase of mice, but activation/inhibition of these neurons increased/decreased NREM sleep (*Yu, X. et al., Nat Neurosci., 2019; Takata, Y. et al. J Neurosci., 2018*). Based on the functional effects and activity pattern of DRN^{GAD2} neurons, we hypothesized that DRN^{GAD2} neurons may constrain or limit wakefulness to avoid hyperarousal, just like the role of VTA Vgat⁺ GABAergic neurons (*Yu, X. et al., Nat Neurosci., 2019; Yu, X. et al., Mol Psychiatry., 2020*). To test this hypothesis, we investigated the role of DRN^{GAD2} neurons in constraining wakefulness time and arousal level in acute restraint stress and tail shock conditions, respectively. Regarding the reviewer's concerns that present data may not fully adequate for justifying our claim, we have performed new experiments and made revisions in the following aspects.

1. Pupil size has been widely used as a readout of arousal level and pupil dilation was regarded as a physiological arousal response (*Bradley, M. M. et al., Psychophysiology, 2008; Breton-*

Provencher, V. et al., *Nat Neurosci.*, 2019). We thus investigated the role of DRN^{GAD2} neurons in arousal control by measuring pupil size. In the present study, the role DRN^{GAD2} neurons in constraining arousal was drawn based on results came from recording activity pattern of DRN^{GAD2} neurons in response to pupil as well as functional manipulation of DRN^{GAD2} neurons during tail shock. We found that the activity of DRN^{GAD2} neurons were positively correlated with pupil size. However, activating DRN^{GAD2} neurons during tail shock attenuated the increased pupil size, whereas inhibition of DRN^{GAD2} neurons further enlarged the increased pupil size. Thus, we spatulated that DRN^{GAD2} neurons may constrain arousal. To make more clearly descriptions about this point, we have revised the related descriptions in the revised manuscript, please see in line 312-315.

2. For the magnitude of response to the manipulation of DRN^{GAD2} neurons in stress and undisturbed conditions, we analyzed the EEG/EMG in 3h after injection saline or CNO in undisturbed conditions (Fig 1 and 2) and in 1h after 10 min restraint in stress conditions (Fig. 4). Considering that different time courses of EMG/EMG in stress and undisturbed conditions were analyzed, it may not approximate to directly compare these results. To directly compare the effects of manipulation of DRN^{GAD2} neurons in stress and undisturbed conditions, we evaluated and compared the pupil size in these two conditions when optogenetically activating DRN^{GAD2} neurons. After mice were habituated to the head-fixed apparatus, we delivered 10s 20Hz stimulation to activate DRN^{GAD2} neurons and recorded the pupil size. Compared with pre-stimulation periods, optogenetic activation of DRN^{GAD2} neurons significantly decreased pupil size (**FigR16a** and **b**, n=6 mice, $P=0.00344$, two tailed paired t test). Correlation analysis results showed that optogenetic activation-induced pupil size change was negatively correlated with baseline pupil size, as activating DRN^{GAD2} neurons in periods with larger baseline pupil size produced more obvious constriction (**FigR16c**). Moreover, activation of DRN^{GAD2} neurons-induced pupil size decrease was larger in tail shock conditions that baseline conditions (**FigR16d**, n=6 mice for each group, $P=0.0203$, two tailed unpaired t test). These results indicated that the magnitude of response to the manipulation of DRN^{GAD2} neurons was more pronounced in stress conditions than in undisturbed conditions.

Fig. R16 | Pupil size decrease following DRN^{GAD2} neurons activation depends on the baseline level of arousal. **a**, Heatmap of all pupil trials from DRN^{GAD2}-ChR2 mice around 20Hz,10s optogenetic stimulation. Pupil_{Norm.}, normalized pupil size. **b**, Effects of optogenetic activation of DRN^{GAD2} neurons on pupil size. **c**, Scatter plot showing the influence of baseline pupil size on activation of DRN^{GAD2} neurons-induced decrease of pupil size. Pupil_{Max}, max pupil size. **d**, The change of pupil size following optogenetic activation of DRN^{GAD2} neurons in baseline conditions and in tail shock conditions. * $P < 0.05$, ** $P < 0.01$. Data were analyzed by two tailed paired or unpaired t test and presented as mean \pm s.e.m.

3. For the activity pattern of DRN^{GAD2} neurons in stress conditions, we simultaneously recorded the pupil size and Ca²⁺ activity of DRN^{GAD2} neurons in response to different amplitude of tail shock. A group of head-fixed DRN^{GAD2}-jGCaMP7b mice received 0.025mA, 0.05mA, 0.1mA, and 0.2mA tail shock. We found that tail shock current-dependently increased pupil size (**FigR16a** and **c**, $P=0.003$, $n=6$ mice, one way repeated measures ANOVA), indicating elevations of arousal level. Moreover, tail shock induced enlargement of pupil size was consistently associated with increased Ca²⁺ activity of DRN^{GAD2} neurons (**FigR16b** and **d**, $P=0.007$, $n=6$ mice, one way repeated measures ANOVA), suggesting potential functional connections between these two events in stress conditions. In conjunction with the positive correlation between Ca²⁺ activity of DRN^{GAD2} neurons and pupil size, we speculated that the activity pattern of DRN^{GAD2} neurons may not only reflected arousal level in undisturbed conditions but also indexed the degree of stress in stressful conditions.

Fig. R17 | Tail shock current-dependently increased pupil size and Ca^{2+} activity of DRN^{GAD2} neurons. **a, b**, Heatmaps (top) and average traces (bottom) of pupil size (**a**) or Ca^{2+} activity of DRN^{GAD2} neurons (**b**) aligned to different amplitude of tail shock onset (vertical red line). Pupil_{Norm.}, normalized pupil size. **c, d**, Effects of different amplitude of tail shock on pupil size (**c**) and Ca^{2+} activity (**d**) of DRN^{GAD2} neurons. ****** $P < 0.01$. Data were analyzed by one way repeated measures ANOVA and presented as mean \pm s.e.m.

Together, these results indicated that DRN^{GAD2} neurons exhibited graded responses to different level of acute stress stimuli. In addition, activation of DRN^{GAD2} neurons produced more pronounced effects on arousal level in acute stress conditions than undisturbed conditions. These results added further evidence supporting a role of DRN^{GAD2} neurons in constraining arousal. These results have been added into **Fig.4a** to **d**, **Supplementary Fig.8**, **Supplementary Fig.10 c** to **f** and described in the main text, please seen in line 285-292, line 305-312.

3. The term "allostatic challenges" is misused. Allostasis refers to chronic stress, not the

response to an acute stressor, as used in this study.

Re: We thank the reviewer for pointing out this and we agree with reviewer's concern. The term "allostatic challenges" was referred from previous published studies (*Saper, C. B. et al., Neuron, 2010; Machado, N. L. S. et al., Curr Biol., 2022*) to indicate conditions requiring animals to shift their behavioral states, such as coping with stressors, encountering predators, or adapting to novel situations. These situations have been called as allostatic loads by McEwen and colleagues (*McEwen, B.S. et al., Neuropsychopharmacology, 2000*), whereas allostasis is the active process that leads to chronic adaptation to a stressor. Considering that the present study only evaluating the role of DRN^{GAD2} neurons in acute stress conditions, we thus corrected "allostatic challenges" to "stressful challenges" when needed to avoid misleading. Moreover, the title of present study was correspondingly revised, please see in line 1.

4. Moreover, the paper lacks proper anatomical characterization of the targeted regions, making it impossible to identify the region targeted or recorded. It is evident from the limited representative images that areas outside the DR were substantially transduced. Similarly, the ablated region has not been characterized. A single 'representative' image is not sufficient to establish colocalization.

Re: We agree with the reviewer's concerns. To clearly show the anatomical characterization of targeted brain regions, we have made improvements in the following aspects.

Firstly, we drawn the detailed infected area of viral constructs in the DRN and adjacent brain regions, including hM3Dq-mCherry, hM4Di-mCherry, ChR2-mCherry, jGCaMP7b and GtACR-GFP. Results showed that the infected areas were largely restricted in the DRN region. Please see in **Supplementary Fig.1a** (hM3Dq-mCherry), **Supplementary Fig.2a** (hM4Di-mCherry), **Supplementary Fig.4a** (ChR2-mCherry), **Supplementary Fig.6c** (jGCaMP7b), and **Supplementary Fig.9c** (GtACR-GFP).

Secondly, we characterized the DTA-ablated brain regions and calculated the number of ablated neurons in these regions using *GAD2* FISH staining. Quantification results revealed that DTA ablation significantly decreased the number of neurons in the DRN (from 100% to 23.04%, $P=0.004$, $n=5$ mice for mCherry group and $n=6$ mice for DTA group, Mann-Whitney Rank Sum Test), but without obvious effects on neurons in adjacent brain regions. Please see in **Supplementary Fig.5a to d**.

Thirdly, we verified the colocalization of transfected viral constructs with DRN^{GAD2} neurons using FISH method. FISH results revealed that these viral constructs (hM3Dq, hM4Di, ChR2, jGCaMP7b and GtACR) have >90% specificity in transfecting DRN^{GAD2} neurons. The above results have been added in **Fig.1d**, **Fig.1j**, **Fig.2d**, **Fig.3d**, and **Supplementary Fig.9c**.

These results suggested that the present study could efficiently and specifically manipulate DRN^{GAD2} neurons.

5. Lack of physiological relevance of manipulations: The authors used 5 mg/kg CNO in some experiments, but this dose has been demonstrated to affect sleep in mice and should not be used; the optogenetic stimulation of neurons continuously for 2 minutes at 20 Hz is not physiologically relevant.

Re: We thank the reviewer for raising these two points. The chosen of CNO dosage and optogenetic parameters were referred to previous studies using chemogenetic and optogenetic

method to investigate neural circuitries for wakefulness/sleep regulation (Weber, F. *et al.*, *Nature*, 2015; Zhang, Z. *et al.*, *Cell*, 2019). For chemogenetic inhibition experiments, we used 5mg/kg CNO to inhibit hM4Di-expressing neurons. This dosage of CNO itself has been reported to affect sleep architecture, particular suppression of REM sleep (Traut, J. *et al.*, *Elife*, 2019). Considering that main effects of chemogenetic inhibition of DRN^{GAD2} neurons were increased wakefulness and decreased NREM sleep, it seems unlikely that the present results came from CNO itself. However, to exclude the potential effects that CNO itself on wakefulness and sleep, we repeated the chemogenetic inhibition experiments using 2mg/kg CNO. Similarly, 2mg/kg CNO injection at the beginning of light on phase consistently increased wakefulness and decreased NREM sleep when compared with saline injection (**Fig. R18**, n=7 mice, $P_{\text{wake}}=0.0011$, $P_{\text{NREM}}=0.00099$, $P_{\text{REM}}=0.313$). In addition, we tested the effects of chemogenetic inhibition of DRN^{GAD2} neurons at the beginning of light off phase and found that 2mg/kg CNO injection also increased wakefulness during the light off phase (please refer to **Supplementary Fig.4j to n**). These results have been added **Fig.2e, f** and **Supplementary Fig.4 b to d** and described in the main text, please see in line 161-170.

Fig. R18 | Chemogenetic inhibition DRN^{GAD2} neurons during the light increase wakefulness. **a**, EEG power spectrogram, EMG trances, and hypnograms from a DRN^{GAD2}-hM4Di mouse during 3h post saline (left) or CNO (right) injection at the beginning of light phase. **b** to **d**, Time course of wakefulness (**b**), NREM sleep (**c**), and REM sleep(**d**) following injection of saline or CNO into DRN^{GAD2}-hM4Di mice at the beginning of light phase. Arrow heads indicate the time point of saline or CNO injection. **e**, Time spent in each state during the first 3h after saline or CNO injection. ** $P<0.01$, *** $P<0.001$, n.s., not significant. Data were analyzed by two way repeated measure ANOVA following Bonferroni's multiple comparisons test (**b** to **d**) or two tailed paired *t* test (**e**) and presented as mean \pm s.e.m.

For optogenetic experiments, we used 2 minutes at 20 Hz optogenetic stimulation to activate DRN^{GAD2} neurons. This parameter was chosen based on previous studies using optogenetic method to explore neural substrates for regulating sleep (Zhang, Z. *et al.*, *Cell*, 2019; Zhong, P. *et al.*, *Neuron*, 2019). As the spontaneous single unit firing activity of DRN^{GAD2} neurons was not recorded in present study, we agree that that our optogenetic stimulation parameter may not mimic

the physiological activity patterns of DRN^{GAD2} neurons. In the revised manuscript, we have discussed this limitation, please see in line 442-446.

6. The statistical approach is not rigorous. For example - Extended Data Fig.5d: Fisher LSD post hoc test does not correct for multiple comparisons and should not be employed. In Fig 3n, the authors should clarify whether all instances of pupil dilation during the recording were collected.

Re: We thank the reviewer for these professional comments. We have reanalyzed the data in Extended Data Fig.5d using one way ANOVA following Tukey's post hoc test. The statistical analysis results showed that DRN^{GAD2} neurons exhibited different *Fos* expression across the light/dark cycles, with highest *Fos* expression in sleep deprivation (SD ZT0-6) conditions. In addition, we carefully rechecked the analysis methods used in the present study to avoid similar mistakes. Please see in **Supplementary Fig.6a and b**, line 199-202.

For pupil analysis in Fig.3n, we randomly collected instances with obvious pupil dilatation in recorded mice. 4 to 6 trials were collected in each mouse and a total of 30 trials were collected from 6 mice (**FigR19 a and b**). According to reviewer's suggestion, we realized that these instances could not adequately reflect the relationship between the activity of DRN^{GAD2} neurons and pupil size. Thus, we reanalyzed the obtained fiber photometry and pupil data according to published methods (*Breton-Provencher, V. et al., Nat Neurosci., 2019; Reitman, M. E. et al., Nat Neurosci., 2023*). To accurately evaluate the activity of DRN^{GAD2} neurons around pupil dilation, we first resampled the Ca²⁺ and pupil data to 20Hz. We then low-pass filtered the pupil data to 0.5Hz using a second-order Butterworth filter. The pupil data was normalized to the maximum diameter and presented in a range between 0 and 1. Candidate pupil dilation events were identified by the sign of the calculated pupil derivative. Only pupil derivative >0 and the changes in pupil diameter >10% were used for subsequent analysis. Candidate pupil dilation events shorter than 1s duration from onset to offset were removed from further analysis. The pupil dilation onset time points were then used to extract corresponding Ca²⁺ data from each aligned time series data. We collected a total of 1509 pupil dilation events and analyzed simultaneously recorded pupil and Ca²⁺ data 3s before and 5s after pupil dilation (**FigR19c**). We found that the Ca²⁺ activity of DRN^{GAD2} neurons were positively correlated with pupil size and significantly increased during pupil dilation (**FigR19c to e**), indicating that the activity of DRN^{GAD2} neurons indexed ongoing arousal state. The reanalyzed data was presented in the **Fig.3n to p** and the related analysis methods were described in the corresponding result and method part, please see in line 232-235 and line 640-654.

Fig. R19 | Spontaneous activity of DRN^{GAD2} neurons correlates with pupil size. **a**, Schematic setup for simultaneous photometry and pupil size recording. **b**, Representative pupil (top) and Ca²⁺ (bottom) traces during spontaneous pupil size recording. **c**, Heatmaps showing all pupil dilation bouts (top) and corresponding Ca²⁺ signals (bottom) extracted from recorded mice (n=6). The vertical red lines indicate the onset of pupil dilation. **d**, Average traces of pupil size (red) and Ca²⁺ signals (dark) aligned to onset of pupil dilation. **e**, Cumulative probability distribution of all the Pearson correlation coefficients of Ca²⁺ signals of DRN^{GAD2} neurons with pupil size. The gray area indicates the non-significant Pearson correlation coefficients. n.s., not significant. Data are presented as mean ± s.e.m.

Minor comments

7. This sentence is misleading and incorrect: “DRN vesicular GABA transporter (Vgat)-positive neurons (DRNVgat) and glutamic acid decarboxylase 2 (GAD2)-positive neurons (DRNGAD2) may represent two functional subgroups of GABAergic neurons, regarding their biased modulation of locomotor activity^{33,38,39}.” Both ref 38 and 39 demonstrated that DRN-GABA neurons are wake-active and wake-promoting. Ref 33 studied the function of the neurons in the context of feeding and is unrelated.

Re: We have deleted this sentence in the revised manuscript.

8. Correlation between delta and population activity during wake: an r-value of 0.07 is very low (and thus provides very low explanatory power), and the fact that there is a correlation with theta doesn't mean it is associated with electrocortical arousal. It can be a behavioral-related modulation.

Re: We have made cautious conclusion in the results part in the revised manuscript, please see in line 221-223.

9. At what circadian time was the acute stress done? Did you handle the control mice at the same time – to exclude the possible effects of being awake?

Re: We apologize for missing this information. The acute restraint stress experiments were performed at the beginning of dark phase (ZT12). The tail shock experiments were during the light on phase (ZT3-ZT9). Mice were handled for 3 to 5 days during the same time before formal experiments. We have added these details in the corresponding method part, please see in line 664-665.

Reviewer #3 (Remarks to the Author):

Ren and Zhang et al. employed a multidisciplinary approach, utilizing various techniques such as opto-/chemogenetics, fiber photometry, polysomnographic recordings, pupil size recording, and circuit tracing, to investigate a novel circuit involved in regulating wakefulness and sleep. Their study specifically focused on the GAD2-positive dorsal raphe nucleus (DRN), which was found to modulate arousal levels in response to allostatic ‘stressful’ events that can potentially limit hypersomnia. The inclusion of pupil analysis in this study to assess arousal levels within the context of allostasis is commendable. By identifying a new pathway that connects physiological conditions with vigilance states, this research not only broadens our understanding of sleep but also has the potential to impact other fields, as the circuit they discovered may contribute to other emotion-related behaviors. The manuscript is well-written and provides a thorough description of the methodology employed. The results presented adequately support the authors' claims. However, to further strengthen the study, additional experiments are required.

Firstly, it is essential to provide a statistical quantification of the percentage of GAD2-positive cells that were transduced with DREADD. This information will contribute to a better understanding of the efficiency and specificity of DREADD-mediated transduction.

Re: We thank the reviewer’s suggestions. We have reanalyzed the FISH results and statistically quantified the percentage hM3Dq or hM4Di transfected GAD2-positive neurons in all DRN^{GAD2} neurons as well as in all DREADD virus transfected neurons. As inspired by the reviewer’s suggestions, we also analyzed the specificity and efficiency of viral vectors encoding Chr2, jGCaMP7b, or GtACR in transfecting DRN^{GAD2} neurons. The specificity and efficiency of these viral vectors were listed in below table.

Table. Efficiency and specificity of viral vectors in transfecting DRN^{GAD2} neurons

Viral vectors	Specificity	Efficiency	n
AAV-EF1 α -DIO-hM3Dq-mCherry	96.60 \pm 1.49%	89.89 \pm 2.06%	4
AAV-EF1 α -DIO-hM4Di-mCherry	90.00 \pm 3.04%	99.71 \pm 0.28%	4
AAV-EF1 α -DIO-ChR2-mCherry	96.06 \pm 2.87%	92.60 \pm 2.90%	3
AAV-EF1 α -DIO-jGCaMP7b	86.75 \pm 0.76%	98.41 \pm 0.83%	3
AAV-EF1 α -DIO-GtACR-GFP	89.07 \pm 4.00%	97.73 \pm 1.16%	3

These results showed that viral vectors used in present study could efficiently and specifically transfected DRN^{GAD2} neurons. The results have been added into the corresponding figures and described in the main text, please see in in **Fig.1d, Fig.1j, Fig.2d, Fig.3d, and Supplementary Fig.9c.**

Similarly, to confirm the specificity and effectiveness of ablation using diphtheria toxin subunit A (DTA), it is necessary to statistically quantify the number of Gad2-positive cells that were deleted in the DRN and adjacent areas. This quantitative analysis will support the reliability of the ablation method.

Re: According the reviewer’s suggestions, we have quantified the number of DTA-ablated

neurons in DRN and adjacent brain regions. These regions include the caudal linear nucleus of the raphe (CLi) locating in the rostral ventral part to DRN, ventrolateral periaqueductal gray (vlPAG) locating in the lateral part to DRN, and dorsal tegmental nucleus (DTg) and laterodorsal tegmental nucleus (LDTg) in the caudal lateral part to DRN. Compared with mCherry controls, DTA-induced lesion significantly decreased the number of GAD2 neurons in the DRN ($P=0.004$, $n=5$ mice for mCherry group and $n=6$ mice for DTA group, Mann-Whitney Rank Sum Test), whereas the number of GAD2 neurons in the CLi, vlPAG, DTg/ LDTg was not obviously affected. We have added these results in **Supplementary Fig.5a to d** and described in the text, please see line174-176.

Fig. R20 | The efficacy of DTA-induced genetic ablation of DRN^{GAD2} neurons. a to c, Representative images showing GAD2-positive neurons in the DRN and adjacent brain regions in mCherry control or DTA mice. d, Quantifications of GAD2-positive neurons in mCherry and DTA group. ** $P<0.01$. Data were analyzed by unpaired t test or Mann-Whitney Rank Sum test and presented as mean \pm s.e.m.

Additionally, I note discrepancies in the activity patterns of GABA cells reported in various studies. It would be worthwhile to investigate whether there are any PV+ (parvalbumin-positive) neurons within the Gad2 cell population or if they are distinct. Previous studies have suggested that PV+ cells exhibit wake-active properties, while Gad2 or other subtypes are primarily active during sleep. Dissecting PV+ cells and examining their population activity during different brain states would provide valuable information on the spontaneous activity of subsets of GABAergic neurons and their role in sleep-wake regulation across different brain regions.

Re: We appreciate the reviewer's constructive suggestions! According to reviewer's suggestions, we examined the expression pattern of PV neurons in the DRN and adjacent brain regions using immunostaining. We first validated the specificity of PV primary antibody (Abcam, Ab181086) in a PV neuron-rich brain region, the thalamic reticular nucleus (TRN). Indeed, we found robust PV expression in the TRN (**Fig.R21a**), validating the specificity of PV antibody and immunostaining method. We then performed PV immunostaining in a consecutive series of brain slices along the anterior to posterior axis of the DRN. In contrast to the TRN, we found that PV-positive neurons were rarely expressed in the DRN. Only a very small population of PV-positive neurons were observed beneath the ventral-lateral part of DRN and near the paratrochlear nucleus (pa4, **Fig.R21**

b to h). To further explore the expression pattern of PV in the DRN, we searched the *PV* mRNA expression in the Allen Brain Atlas (<https://mouse.brain-map.org/experiment/show/868>). Similar to our immunostaining results, *PV* mRNA was observed in the adjacent brain regions but weakly expressed in the DRN (**Fig.R21i** and **j**).

These results were in accordance with previous findings that *PV*-positive neurons were sparse in the DRN (Tan, S. K. et al., *J Psychiatr Res*, 2011; Li, X. T. et al., *Neuroscience*, 2020). Although *PV*-positive GABAergic in certain brain regions (basal forebrain and substantia nigra for example) exhibited wake-active properties (Xu, M. et al., *Nat Neurosci.*, 2015; Liu, D. et al., *Science*, 2020), we did not investigate the spontaneous activity and functional role of these neurons when considering their weak expression in the DRN. These results have been added in **Supplementary Fig.16** and discussed in the revised manuscript, please see in line 180-483.

Fig. R21 | The expression of PV-positive neurons in the DRN and adjacent brain regions. a, images showing the expression of PV-positive neurons in the TRN. Right image is an enlarged view of zoomed area in the left image. **b to h**, PV-positive neurons in the DRN and adjacent brain regions. PV-positive neurons were rarely expressed in the DRN. Results were repeated in 3 mice. **i** and **j**, *in situ* hybridization with *PV* in the DRN from Allen Brain Atlas (<https://mouse.brain-map.org/experiment/show/868>).

In terms of minor comments, it is recommended to provide more precise labeling for AAV-ef1 α -DIO-hM3D-mCherry as hM3Dq, as well as for other hM3D and hM4D constructs mentioned in the manuscript (line 117).

Re: Thanks for the reviewer's suggestions. In the revised manuscript, we have corrected this in the revised manuscript.

Furthermore, the description of the stress protocol appears inconsistent. While it is

mentioned that the animals were restrained for 10 minutes, line 246 states that mice were subjected to acute restraint for 1 hour. Clarifying this inconsistency is necessary for accurate interpretation.

Re: We have corrected this inconsistent description in the revised manuscript, please see in line 255-257. In addition, we carefully checked the manuscript to avoid similar mistakes.

REVIEWER COMMENTS

Reviewer #1 (Remarks to the Author):

It is great that the authors specifically manipulated stress-related DRN GAD2 neurons using c-Fos-based activity tagging method (TetTagging). Relevant results further strengthen the role of DRN GAD2 neurons in constraining wakefulness in acute stress condition. I am happy that the authors carefully examined the infected area of AAV in the DRN and adjacent brain regions, which excludes the effect of PAG on experimental results. I am also pleased that functional connections between DRN GAD2 neurons and the PVT were further confirmed with in vitro electrophysiological experiment and behavioral experiment in acute stress conditions.

The revision authors made and the additional data authors provided significantly improved the manuscript, and have addressed my key concerns with the original submission. I have only some minor issues as below:

1. We are pleased that the authors have uploaded the video showing the sleep-promoting effect of optogenetic activation of DRN GAD2 neurons, but only video of the experimental mice with blue light stimulation. For readers to understand the sleep-promoting effect more clearly, the authors also need to provide the video of control mice or experimental mice with no light stimulation.
2. To better illustrate that DRN GAD2 neurons constrain arousal responses under stress conditions, it is hoped that authors provide typical videos or figures of experiments of Figure 5f, if it is possible.
3. In figure 3, the authors showed that population activity of DRN GAD2 neurons across different wakefulness/sleep states. It seems that activity of DRN GAD2 neurons during NREM sleep is a little higher than that during REM sleep, which is quite different from previous results of DRN Vgat neurons (Sleep. 2022 Dec 12;45(12):zsac235. and J Neurosci. 2021 Jun 2;41(22):4840-4849). It is good that the authors discussed this difference, which may help explain the different results of activating DRN GAD2 neurons and Vgat neurons, and may help illustrate the complex mechanisms underlying DRN GABA neurons regulate sleep-wake behavior.
4. The authors have mis-labelled the serial number of figure 7 as figure 6.

Reviewer #1 (Remarks on code availability):

The code is available, which would be used to analyze open field data with MATLAB software.

But the authors did not provide a README file.

A file is recommended to be provided for readers to run the code easier.

Reviewer #3 (Remarks to the Author):The authors have effectively addressed my concerns, and the revised version demonstrates significant improvement. In addition to my concerns, they characterized the molecular identities of targeted cell types, revealing an intermingled heterogeneity. Additionally,

they utilized activity-tagging to explicitly identify ensembles linking stressful stimuli and arousal. I have no substantial concerns, and I now recommend the publication of this study.

REVIEWER COMMENTS

Reviewer #1 (Remarks to the Author):

It is great that the authors specifically manipulated stress-related DRN GAD2 neurons using c-Fos-based activity tagging method (TetTagging). Relevant results further strengthen the role of DRN GAD2 neurons in constraining wakefulness in acute stress condition. I am happy that the authors carefully examined the infected area of AAV in the DRN and adjacent brain regions, which excludes the effect of PAG on experimental results. I am also pleased that functional connections between DRN GAD2 neurons and the PVT were further confirmed with *in vitro* electrophysiological experiment and behavioral experiment in acute stress conditions.

The revision authors made and the additional data authors provided significantly improved the manuscript, and have addressed my key concerns with the original submission. I have only some minor issues as below:

1. We are pleased that the authors have uploaded the video showing the sleep-promoting effect of optogenetic activation of DRN GAD2 neurons, but only video of the experimental mice with blue light stimulation. For readers to understand the sleep-promoting effect more clearly, the authors also need to provide the video of control mice or experimental mice with no light stimulation.

Re: We thank the reviewer for this suggestion. In the revised manuscript, we have provided an additional video showing the effects on optogenetic stimulation on wakefulness/sleep of DRN^{GAD2}-mCherry mice. This video file shows that optogenetic stimulation itself has no obvious effects on the behavioral states of DRN^{GAD2}-mCherry mice. Please see in Supplementary Video 1, page 4 line 13-15, and page 52 line 10-16.

2. To better illustrate that DRN GAD2 neurons constrain arousal responses under stress conditions, it is hoped that authors provide typical videos or figures of experiments of Figure 5f, if it is possible.

Re: According to reviewer's suggestion, we have provided a video file showing the effects of optogenetic inhibition of DRN^{GAD2} neurons on pupil size related to Figure 5f. Compared with light off condition, inhibition of DRN^{GAD2} neurons with light on induces pronounced enlargement of tail shock-induced pupil size. Please see in Supplementary Video 2, page 7 line 38-40 and page 52 line 18-23.

3. In figure 3, the authors showed that population activity of DRN GAD2 neurons across different wakefulness/sleep states. It seems that activity of DRN GAD2 neurons during NREM sleep is a little higher than that during REM sleep, which is quite different from previous results of DRN Vgat neurons (Sleep. 2022 Dec 12;45(12):zsac235. and J Neurosci. 2021 Jun 2;41(22):4840-4849). It is good that the authors discussed this difference, which may help explain the different results of activating DRN GAD2 neurons and Vgat neurons, and may help illustrate the complex mechanisms underlying DRN GABA neurons regulate sleep-wake behavior.

Re: We appreciate the reviewer's constructive suggestions. Gazea et al. (J Neurosci. 2021 Jun 2;41(22):4840-4849) and Cai et al. (Sleep. 2022 Dec 12;45(12):zsac235.) found that DRN^{Vgat} neurons were wakefulness-active and wakefulness-promoting. The present study found that

DRN^{GAD2} neurons were also wakefulness-active, but their activation decreased wakefulness. Though both DRN^{GAD2} neurons and DRN^{Vgat} neurons were wakefulness-active, these neurons exhibited biased activity pattern during sleep. DRN^{Vgat} neurons were more active during REM sleep than that during NREM sleep and slowly increased their activity during transitions from NREM sleep to REM sleep, whereas no obvious activity changes of DRN^{GAD2} neurons had been observed during NREM sleep and REM sleep. The difference of activity pattern and functional outcomes of DRN^{GAD2} neurons and DRN^{Vgat} neurons suggested that these two DRN neuronal populations have different roles in wakefulness/sleep regulation. We have added these discussions in the revised manuscript, please see in page 11 line 15-23.

4. The authors have mis-labelled the serial number of figure 7 as figure 6.

Re: We thank the reviewer for pointing out this mistake. We have corrected this in the revised manuscript. Please see in page 32 line 2.

Reviewer #1 (Remarks on code availability):

The code is available, which would be used to analyze open field data with MATLAB software. But the authors did not provide a README file. A file is recommended to be provided for readers to run the code easier.

Re: We thank the reviewer for raising this point. As the reviewer suggested, we have added a README file for running the open field data analysis code. Please see in <https://github.com/shuanchengren/open-filed-analysis>.

Reviewer #3 (Remarks to the Author):

The authors have effectively addressed my concerns, and the revised version demonstrates significant improvement. In addition to my concerns, they characterized the molecular identities of targeted cell types, revealing an intermingled heterogeneity. Additionally, they utilized activity-tagging to explicitly identify ensembles linking stressful stimuli and arousal. I have no substantial concerns, and I now recommend the publication of this study.

Re: We thank the reviewer for these encouraging comments and for the suggestions improving our manuscript.

REVIEWERS' COMMENTS

Reviewer #1 (Remarks to the Author):

The authors have addressed all of my concerns with the original submission. I believe the authors have done a good job in the revision and by doing so the manuscript has been improved significantly. Now, I recommend the paper for publication without any reservation. Congratulate the authors on an interesting manuscript.

Reviewer #1 (Remarks on code availability):

The authors have added a README file to help readers run the code.

Reviewer #1 (Remarks to the Author):

The authors have addressed all of my concerns with the original submission. I believe the authors have done a good job in the revision and by doing so the manuscript has been improved significantly. Now, I recommend the paper for publication without any reservation. Congratulate the authors on an interesting manuscript.

Reviewer #1 (Remarks on code availability):

The authors have added a README file to help readers run the code.

Re: We thank the reviewer for these encouraging comments and the his/her constructive suggestions to improve the overall quality of our work.